# Harnessing landrace diversity empowers wheat breeding

Shifeng Cheng[1,32✉], Cong Feng[1,32], Luzie U. Wingen[2,32], Hong Cheng[1], Andrew B. Riche[3], Mei Jiang[1], Michelle Leverington-Waite[2], Zejian Huang[1], Sarah Collier[2], Simon Orford[2], Xiaoming Wang[2,4], Rajani Awal[2], Gary Barker[5], Tom O'Hara[2], Clare Lister[2], Ajay Siluveru[2], Jesús Quiroz-Chávez[2], Ricardo H. Ramírez-González[2], Ruth Bryant[6], Simon Berry[7], Urmil Bansal[8], Harbans S. Bariana[8,9], Malcolm J. Bennett[10], Breno Bicego[11], Lorelei Bilham[2], James K. M. Brown[2], Amanda Burridge[5], Chris Burt[6], Milika Buurman[12], March Castle[3], Laetitia Chartrain[2], Baizhi Chen[1], Worku Denbel[13], Ahmed F. Elkot[14], Paul Fenwick[15], David Feuerhelm[16], John Foulkes[10], Oorbessy Gaju[10], Adam Gauley[17,18], Kumar Gaurav[2], Amber N. Hafeez[2], Ruirui Han[1,19], Richard Horler[2], Junliang Hou[1], Muhammad S. Iqbal[1], Matthew Kerton[20], Ankica Kondic-Spica[21], Ania Kowalski[2], Jacob Lage[22], Xiaolong Li[23], Hongbing Liu[1], Shiyan Liu[1], Alison Lovegrove[3], Lingling Ma[1], Cathy Mumford[2], Saroj Parmar[3], Charlie Philp[2], Darryl Playford[2], Alexandra M. Przewieslik-Allen[5], Zareen Sarfraz[1], David Schafer[6], Peter R. Shewry[3], Yan Shi[1], Gustavo A. Slafer[11,24], Baoxing Song[25], Bo Song[1], David Steele[3], Burkhard Steuernagel[2], Phillip Tailby[7], Simon Tyrrell[26], Abdul Waheed[1], Mercy N. Wamalwa[27], Xingwei Wang[1], Yanping Wei[1], Mark Winfield[5], Shishi Wu[1], Yubing Wu[1,28], Brande B. H. Wulff[2,29], Wenfei Xian[1,30], Yawen Xu[1,28], Yunfeng Xu[1], Quan Yuan[1], Xin Zhang[1,28], Keith J. Edwards[5], Laura Dixon[17], Paul Nicholson[2], Noam Chayut[2], Malcolm J. Hawkesford[3], Cristobal Uauy[2], Dale Sanders[2], Sanwen Huang[1,31] & Simon Griffiths[2✉]

Harnessing genetic diversity in major staple crops through the development of new breeding capabilities is essential to ensure food security[1]. Here we examined the genetic and phenotypic diversity of the A. E. Watkins landrace collection[2] of bread wheat (*Triticum aestivum*), a major global cereal, by whole-genome re-sequencing of 827 Watkins landraces and 208 modern cultivars and in-depth field evaluation spanning a decade. We found that modern cultivars are derived from two of the seven ancestral groups of wheat and maintain very long-range haplotype integrity. The remaining five groups represent untapped genetic sources, providing access to landrace-specific alleles and haplotypes for breeding. Linkage disequilibrium-based haplotypes and association genetics analyses link Watkins genomes to the thousands of identified high-resolution quantitative trait loci and significant marker–trait associations. Using these structured germplasm, genotyping and informatics resources, we revealed many Watkins-unique beneficial haplotypes that can confer superior traits in modern wheat. Furthermore, we assessed the phenotypic effects of 44,338 Watkins-unique haplotypes, introgressed from 143 prioritized quantitative trait loci in the context of modern cultivars, bridging the gap between landrace diversity and current breeding. This study establishes a framework for systematically utilizing genetic diversity in crop improvement to achieve sustainable food security.

The world population is projected to increase by 2 billion people over the next 30 years, placing greater demands on wheat (which currently accounts for 20% of human caloric intake) as a vital source of calories, protein, minerals and fibre[3]. Our ability to meet demand is threatened by climate change and geopolitical instability, which have multiplying effects when they disrupt the narrow global wheat export base[4]. A few countries (Russia, USA, Canada, France, Ukraine, Australia and Argentina) dominate exports, and major population centres such as China require adequate imports to satisfy internal demand[5]. Compounding these challenges, yield gains in wheat have slowed, in part owing to the narrowing genetic diversity of modern cultivars.

Historically, farmers relied on locally adapted crop cultivars known as landraces, which could withstand environmental hardships, but compared with modern cultivars, were low yielding[6]. Compared with modern cultivars, landraces have been less exposed to historical and

geographical founder effects, making them a rich, albeit underutilized, source of diversity. There is great potential to enhance the traits of interest in modern cultivars by cross-pollinating with landraces, aiming for superior progeny. However, such efforts are often restricted to major effect traits (for example, disease resistance genes) as the integration of diversity for complex, quantitative traits faces technical, scientific and economic challenges[2]. Among these challenges is the lack of genetic resources and appropriate phenotypic datasets underpinned by sequence information necessary for identifying alleles present in the landraces but absent in modern cultivars. This results in the infrequent use of landrace diversity in modern breeding. Thus, despite the potential for developing more resilient and nutritious crops, the full extent of landrace value is limited by the availability of sufficient genomic, genetic and phenotypic resources for the discovery and deployment landrace diversity which constrains their utilization in breeding.

Here we present the collaborative work of an international consortium that has overcome these obstacles and harnessed untapped landrace diversity (Extended Data Fig. 1). Our strategy capitalized on the rich genetic, geographic and phenotypic diversity within the A. E. Watkins landrace collection of bread wheat (hereafter 'Watkins'), comprising 827 accessions collected from 32 countries in the 1920s and 1930s[2] (Supplementary Note 1). We implemented a pre-breeding strategy[1] to decode, discover, design and deliver progress in breeding. We combined Watkins gene discovery populations[7] with a genomic variation matrix, haplotype map and field phenotyping for quantitative traits. This approach generated an integrated set of tools that provides the research and breeding communities with access to new beneficial diversity.

## Untapped diversity in Watkins landraces

To identify novel genetic variation in Watkins, we conducted 12.73× whole-genome re-sequencing of its 827 accessions (Supplementary Table 1). We aligned these sequences to the IWGSC RefSeq v1.0 bread wheat reference genome[8] and used the SNPs identified to infer population structures within Watkins, visualized by *t*-distributed stochastic neighbour embedding (*t*-SNE), and utilized the maximum likelihood method implemented in ADMIXTURE[9] (Supplementary Tables 2 and 3 and Supplementary Fig. 1). Watkins was categorized into seven ancestral groups designated AG1 to AG7 (Fig. 1a,b, Extended Data Fig. 2, Supplementary Figs. 2 and 3 and Supplementary Table 4). The collection sites of the landrace accessions represent all the major wheat-growing areas of the world in the 1920s (Fig. 1a and Supplementary Table 1). To explore the relationship between modern wheat and the ancestral groups, we selected 208 independent wheat cultivars for whole-genome sequencing, as well as 15 previously described accessions[10], which maximize diversity among 1,169 cultivars from 25 countries (Extended Data Fig. 3) that were genotyped using the Wheat Breeders' Array[11] (hereafter 'modern wheat') (Supplementary Table 1). Modern wheat comprises registered crop varieties within systematic wheat breeding programmes, in contrast to Watkins, which comprises landrace cultivars, which are not products of systematic breeding. Taking Watkins and modern wheat together, we identified around 262 million high-quality single nucleotide polymorphisms (SNPs) (Supplementary Tables 2 and 3). The SNP composition of modern wheat largely overlaps with that of AG2 and AG5 (AG2/5) (Fig. 1b), which have Western and Central European origins, respectively, suggesting that these ancestral groups supplied the founder lines of modern wheat. The AG2/5 hypothesis for the origins of contemporary wheat is supported by independent wheat genomics datasets[12] (Extended Data Fig. 2c). Watkins contains variants that are absent in modern wheat. We identified 162 million SNPs (62%), 9.7 million insertions or deletions (57%) and 57,000 copy number variants (CNVs, 53%) unique to Watkins. These are predominantly carried by five ancestral groups (AG1, AG3, AG4, AG6 and AG7)

(Fig. 1c, Supplementary Fig. 4 and Supplementary Tables 5 and 6), showing almost no overlap with modern wheat. These data indicate that the five phylogenetically isolated ancestral groups are highly diverse and represent a reservoir of previously untapped diversity for wheat breeding.

To further explore the landrace origins of modern wheat, we used long-range haplotypes to visualize the mosaic of identity by state (IBS) regions across their genomes (Fig. 1d and Supplementary Table 7). These IBS segments are signatures of the relatives of Watkins that were the founder lines of modern wheat cultivars, which have retained high chromosome-level identity with AG2/5 landraces, often across multi-megabase tracts extending across the majority of a chromosome's length (Supplementary Fig. 5). On average, IBS segments remained intact along a length of 159.78 Mb in centromeric regions[13]. They were shorter in distal regions. The IBS analysis provided insight into the very small effective population size of modern wheat. As few as 26 Watkins accessions could be modelled as virtual donors of IBS segments to reconstitute more than 50% of the modern wheat genomes.

To map variants that are absent from modern wheat, we used linkage disequilibrium (LD)-based haplotype analysis[14]. This identified 71,282 haploblocks, of which 69.6% (49,626) only contain Watkins-unique haplotypes (median 5 and mean 11.85 haplotypes per haploblock; Fig. 1c and Supplementary Tables 8 and 9). We aligned these haploblocks to the IBS chromosome map of modern wheat, revealing the potential for new arrangements of chromosomal segments to enrich the current IBS structure of wheat germplasm with Watkins-unique haplotypes (Fig. 1e). However, in addition to the unique variants identified in Watkins, 2.5% of the unique haplotype variants were found in modern wheat, several of which were associated with introgressions from wheat wild relatives made by breeders (for example, *1BL/1RS*[15], *RHT1*[16] and *Pch1*[17]), after the AG2/5 landrace foundation of modern wheat (Fig. 1e).

To assess potential for Watkins-unique diversity to influence traits, we studied variations occurring in or around genes. Among the Watkins-unique SNPs, 325,915 affect gene function (Supplementary Table 3). Particularly noteworthy among these are the Watkins-unique SNPs (annotated by SnpEff[18]) found in 13,902 genes that are monomorphic in modern wheat, meaning that there is no variation present in elite wheat pedigrees to improve traits associated with these genes (Supplementary Fig. 6). According to ontology term analysis, these genes control diverse biological processes that could affect important agronomic traits such as yield, stress tolerance, nutritional quality and disease resistance. To leverage these genomics resources and to search for useful trait variations associated with these variants, we undertook a programme for high-throughput quantitative trait locus (QTL) discovery.

## Phenotyping to valorize landrace haplotypes

Economically important traits for improvement of grain yield and quality, as well as those required for climate change adaptation and mitigation, are often controlled by multiple genetic loci in a quantitative manner[19,20]. Thus, structured populations combined with specialized association genomics analyses[21] are required for their study. We used the 827 Watkins accessions as a genome-wide association panel for phenotypic datasets recorded in the UK and China (Supplementary Fig. 7 and Supplementary Tables 10 and 11). We also used 73 'Paragon' × Watkins recombinant inbred line (RIL) populations[7] (Fig. 2a, Extended Data Fig. 4 and Supplementary Table 12), resulting in 6,762 RILs for which large-scale field-based phenotyping and whole-genome imputation was performed. We recorded phenotypic data for 137 traits (Supplementary Tables 13 and 14), covering the major categories of grain yield, nutritional quality, adaptation, and abiotic and biotic stress tolerance (Fig. 2b). These extensive

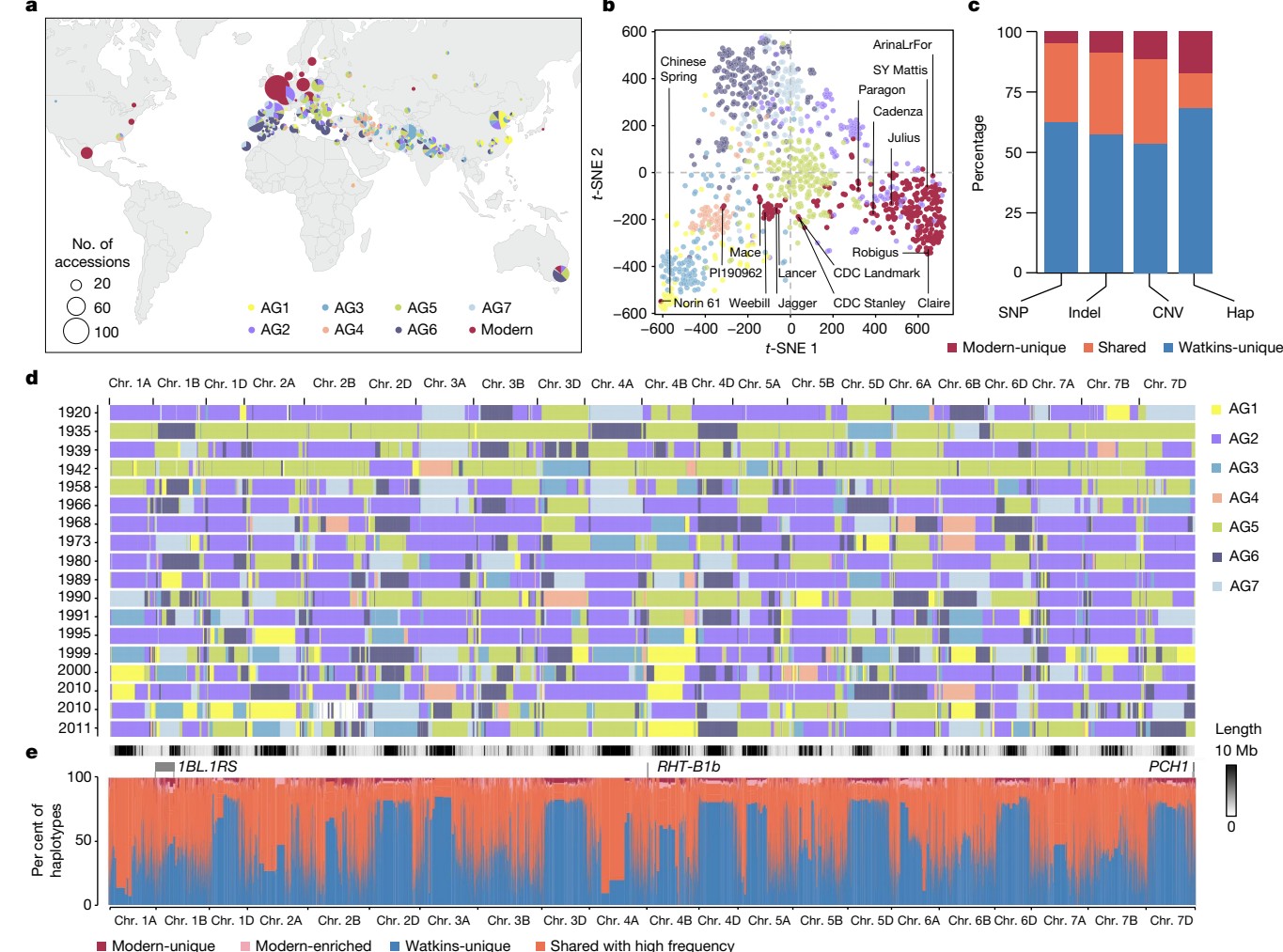

**Fig. 1 | Genomic variants in Watkins landraces compared with modern wheat. a**, Geographical distribution of all accessions, including the entire Watkins collection (*n* = 827) and modern wheat cultivars that are the outputs of breeding programmes (*n* = 224; comprising 208 cultivars sequenced in this study and 15 previously described wheat cultivars from the 10+ Wheat Genomes Project[10] as well as Chinese Spring). The seven ancestral groups (AG1–AG7, derived from Watkins) and modern wheat are colour-coded. **b**, *t*-SNE plot based on the 10 million SNPs shared by different ancestral groups. The SNPs were stringently controlled by LD (see Methods), with AG1–7 and modern wheat colour-coded as in **a**. The distribution of the 15 lines from the 10+ Wheat Genomes Project[10] and Chinese Spring are shown. **c**, Percentage of Watkins-unique, shared and modern-unique variants for SNPs, short (<50 bp) insertion–deletion mutations (indels), gene copy number variations (CNV)

and haplotypes (hap). **d**, *k*-mer based IBSpy long-range haplotype analysis of the selected 18 representative modern wheat cultivars (released from 1920 to 2011). IBS regions shared between these 18 modern wheat lines and the Watkins landraces are shown as coloured blocks according to the source of the detected Watkins accessions from the different ancestral subgroups (top 100 Watkins accessions; Supplementary Table 7; see Methods and ref. 16). **e**, Genomic distribution and comparison of haplotypes between Watkins and modern wheat along the 21 chromosomes, including the proportion of haplotypes that are absent (Watkins-unique), shared with high frequency, modern-enriched or unique to modern wheat. The haplotypes were identified based on LD by PLINK (Methods), with single-base-resolution based on the IWGSC RefSeq v1.0 reference genome.

field-based experiments were conducted over 10 years in 10 environments, resulting in over 717,000 observations and data points (Fig. 2c,d and Supplementary Table 15). The structured populations enabled us to capitalize on the complementary strengths of joint linkage and association studies for complex traits by nested association mapping (NAM) studies, as well as classical QTL analysis in bi-parental populations and gene discovery strategies such as genome-wide association studies (GWAS) (Supplementary Fig. 8 and Supplementary Tables 16–19 and Methods).

Combining the mapping populations with sequence-based haplotypes, NAM–GWAS capture both historical and RIL population recombinations as well as common alleles from the natural population and rare useful alleles with amplified frequency in the segregating populations. Using this approach, we calculated robust QTL effects at

haplotype resolution and determined the distribution of useful QTL alleles between Watkins landraces and modern wheat (Fig. 2e,f and Supplementary Tables 20–22). We defined useful QTL alleles as those that influence a phenotypic value for traits in a direction selected in breeding. For traits subject to directional selection, such as yield and disease resistance, the useful allele was always acting in the same phenotypic direction, but for traits such as heading date, which are subject to stabilizing or disruptive selection, both allelic directions were considered useful (Extended Data Fig. 5).

In total, we identified 8,253 genetic effects (3,280 QTL, 1,428 GWAS and 3,545 NAM–GWAS marker–trait associations (MTAs); Supplementary Tables 17–19). On the basis of the direction of the allelic effects, 1,696 have the potential to improve modern cultivars such as Paragon, and 36% (613) of the most significantly associated SNPs are located

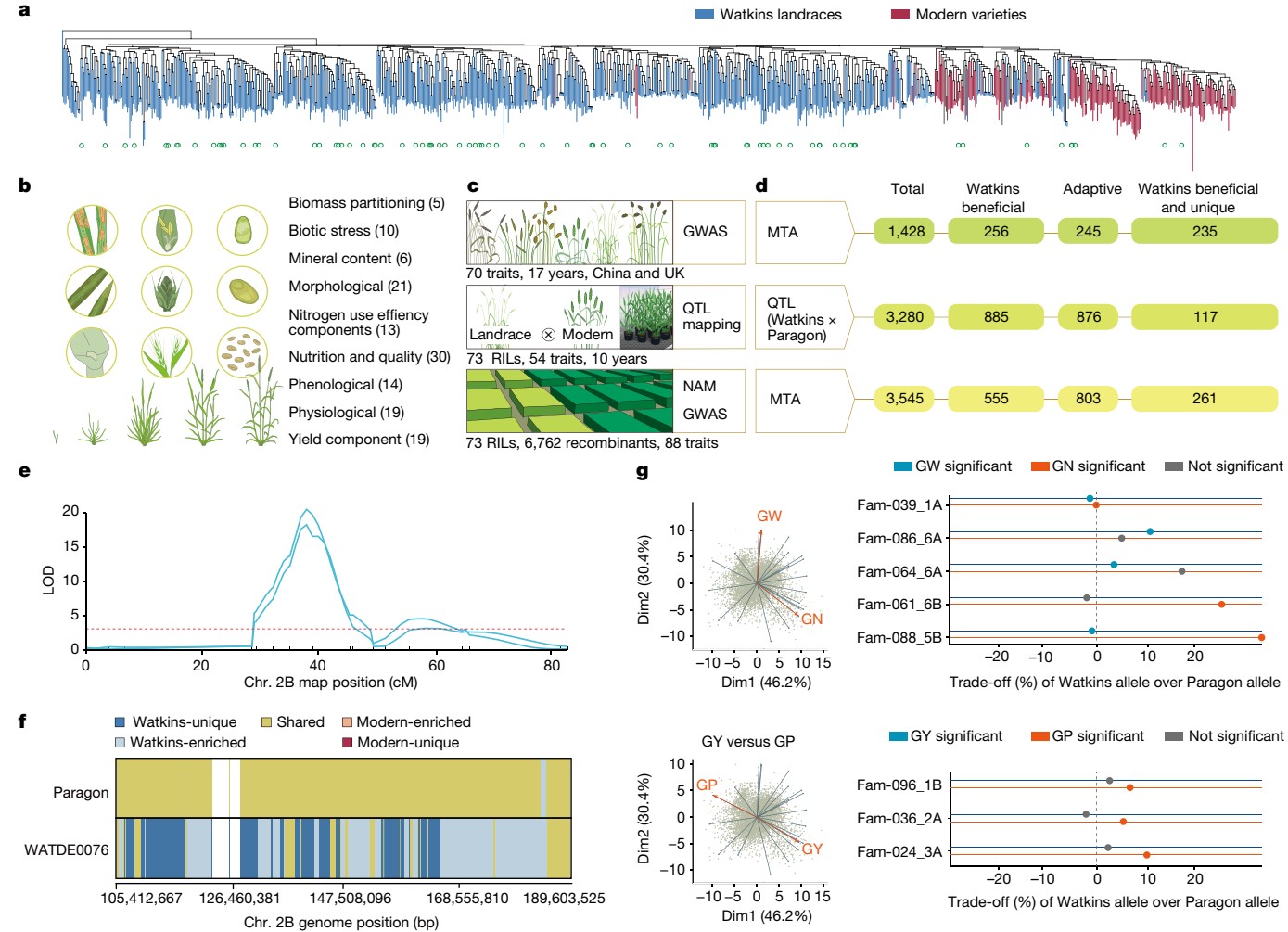

**Fig. 2 | Genetic dissection of useful traits from Watkins. a**, SNP phylogeny of Watkins and modern lines. The tree was built on the core SNP set (Watkins parents of 73 RIL populations are marked with green circles). **b**, Schematic of trait data collected and categorized into nine classes. Total number of sub-traits for each trait is shown in parentheses. **c**, Field experiments and trait data collected from the Watkins natural population for GWAS analysis (top), QTL analysis from individual segregating mapping populations (RIL; middle) and combined analysis of NAM–GWAS with RILs from imputation of the NAM populations (bottom). **d**, Genetic association signals from different methods. Total number of MTAs detected from GWAS (top), NAM (bottom). Middle, summary of total number of QTLs identified using RILs. Watkins beneficial, number of allelic effects in which the Watkins allele exceeded Paragon allele for traits under directional selection; adaptive, effects that can be beneficial

in either direction; Watkins beneficial, adaptive and unique, haplotype under the peak marker or genomic interval was absent or infrequent in modern. **e**, Examples of genetic effect detection as QTL (yellow rust resistance) using data collected from experiments in **c**. LOD, logarithm of the odds. **f**, Haplotype frequency within QTL peak interval blocks in **e** for Paragon and WATDE0076 (Supplementary Table 17), as Watkins-unique, Watkins-enriched, shared, modern-enriched, modern-unique and other (white, no haploblock). **g**, Left, principal component analysis (PCA) of Watkins NIL data, highlighting trait trade-off relationships (grain weight (GW) versus grain number (GN) and grain yield (GY) versus grain protein (GP)). Right, percentage phenotypic differences for the Watkins versus Paragon allele. Each QTL is represented by a NIL pair or family (Fam) (Supplementary Table 31). Significant effects ($P < 0.05$, $F$-statistics) for traits are shown by coloured or grey circles.

within haplotypes that are absent in modern wheat (Fig. 2d and Supplementary Table 16). Despite reduced genetic resolution compared with NAM and GWAS, the use of single bi-parental populations was essential for detecting QTLs from very rare Watkins haplotypes. For example, just 33 Watkins accessions exhibit resistance to the 'Warrior' race of *Puccinia striiformis* (the causal agent of yellow rust disease; Fig. 2e and Supplementary Table 23), a recently emerged, highly aggressive race with increased pathogenicity at elevated temperatures[22]. Iran is the dominant country of origin for these resistant accessions (14 out of 33). GWAS did not identify significant MTAs for these resistance loci, but bi-parental QTL mapping identified 15 new loci conferring yellow rust resistance in the UK and Australia, including to the Warrior race (Supplementary Tables 24 and 25). Twelve of these resistance loci are carried by accessions outside of the modern wheat AG2/5 founder complex, with five originating from Iran. These results highlight the

potential of using the large set of genetic effects identified here to help deliver new traits of agronomic value.

To elucidate the potential utility of these alleles for practical wheat improvement through breeding, we investigated the relationships between multiple key traits (Fig. 2g). This approach is highly relevant, as QTL for one agronomic trait can often be antagonistically coupled to other breeding targets[23]. Thus, uncoupling these relationships, often thought of as trade-offs, is crucial for accelerating the breeding process. We developed near isogenic lines (NILs) to test the extent to which these trait relationships (for example, grain weight versus grain number or grain yield versus grain protein content) were upheld for individual QTL effects (Fig. 2g and Supplementary Table 26). For each locus–trait combination, we found a range of penetrance for these mainly antagonistic relationships, with several positive-effect QTLs for one trait being either neutral or positive for the other trait, reversing

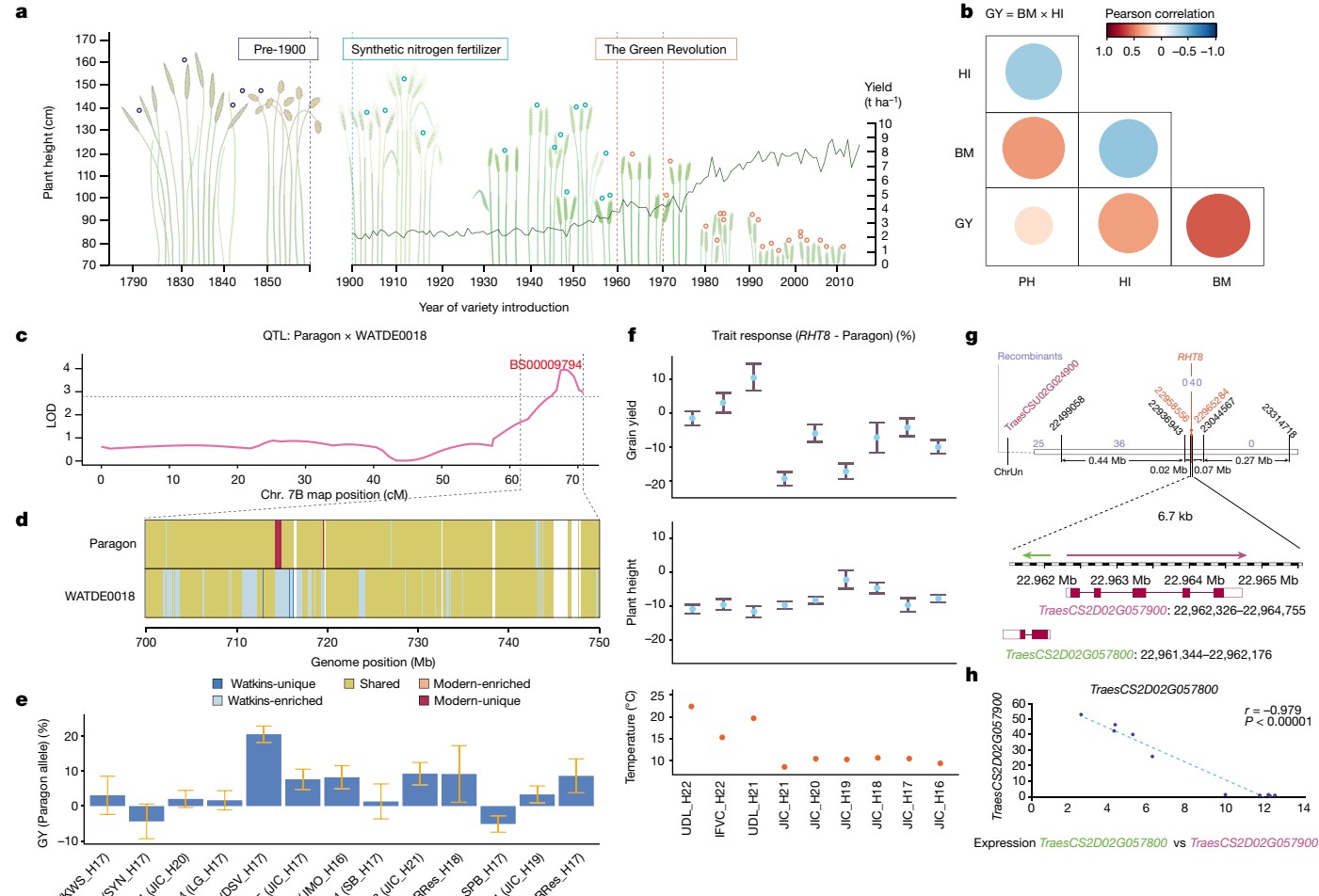

**Fig. 3 | Recovery of useful diversity left behind by the green revolution.**
**a**, Historical reduction in plant height of wheat. Heritage cultivar heights (empty circles) from 1790 to 2010[40] and yields (lines) from 1900 to present. Agricultural milestones are highlighted. **b**, Trait relationships between grain yield, harvest index (HI), plant height (PH) and biomass (BM) of the RILs indicated as a coloured circle, with the colour indicating Pearson correlation. The strength of the significance is represented by circle size. **c**, Plant height QTL (chromosome arm 7BL). The genetic confidence interval ranges from 62.2 to 70.3 cM, corresponding to 701 to 744 Mb, depicted by lines connecting **c** and **d**. **d**, Haplotype blocks within confidence interval from **c** for Paragon and WATDE0018 (Supplementary Table 17). Blocks are coloured by frequency, as Watkins-unique, Watkins-enriched, shared, modern-enriched, modern-unique and other (white, no haploblock). **e**, WATDE0018 segment on grain yield effect. Label prefixes are grain yield in t ha⁻¹. Number after H indicates year harvested; harvest locations include: KWS, KWS Fowlmere; SYN, Syngenta; JIC, John Innes

Centre Church Farm; LG, Limagrain; DSV, DSV, JMO, John Innes-Morley; SB, Sutton Bonington; RRes, Rothamsted; LSPB, LS Plant Breeding (details in Methods). Data are mean ± s.e.m. $n$ = 3 independent field plots per NIL, except $n$ = 4 for LSPB_H17 and $n$ = 5 for JIC_H21. **f**, Mean percentage plant height (middle) and grain yield (top) effects of *RHT8* (Mara) versus Paragon in the UK, Spain and Serbia. H, year harvested; JIC, John Innes Centre (UK); UDL: University of Lleida (Spain); IFVC, Institute of Field and Vegetables Crops (Serbia). Bottom, growing season temperature. Data are mean ± s.e.m. Number of independent field plots per per NIL: $n$ = 5 (H21 and H22), $n$ = 4 (H16–H19) and $n$ = 3 (H20). **g**, Genetic map of the *RHT8* locus delimiting to a 6.7-kb interval. Marker position is shown in black and flanking markers are in red. Independent recombination lines between *RHT8* phenotype are in purple. **h**, RNA-seq expression data for *TraesCS2D02G057800* and *TraesCS2D02G057900* during tiller development reveal a negative regulatory relationship (Pearson correlation $r$ = 0.979; $P$ < 0.0001).

the general relationship and providing a route for selection without the trade-offs that the founding breeders of modern wheat could not avoid (Fig. 2g).

This understanding of trade-offs helped us form a strategy for the recovery of beneficial phenotypes that have been lost in modern breeding owing to preferential selection pressure for one of the paired traits. For example, breeding for reduced crop height to avoid lodging, and increase yield via harvest index[23] was a continuous process since the start of the twentieth century. This culminated in the development of the *RHT1* semi-dwarfing Green Revolution wheat cultivars in the 1960s[24], characterized by low stature and high harvest index and yield (Fig. 3a). However, this coupling of reduced stature and increased harvest index was likely achieved at the expense of crop biomass, which is positively correlated with height[25]. Recognizing the importance

of biomass as a physiological component of grain yield[23], we sought Watkins alleles that increase height/biomass while maintaining or enhancing yield. Among the 291 QTLs identified for plant height, 187 conferring reduced height were derived from the cultivar Paragon (Fig. 3b and Supplementary Table 17), reflecting the selection history for this trait in the modern cultivar and its multi-genic nature. We selected a height-increasing Watkins haplotype on chromosome arm 7BL that had no adverse effect on harvest index, unlike other height-increasing QTL alleles. The local LD block corresponding to the chromosome arm 7BL target region contains contiguous haplotypes that are either absent or present at low frequency in modern wheat (Fig. 3c,d). Testing in NILs in multi-location trials together with breeding companies, we determined that these haplotypes were associated with a height increase of 9.37 cm ($P$ = 0.002) and a grain yield

increase of 0.39 t ha$^{-1}$ ($P < 0.002$) compared with Paragon (Fig. 3e and Supplementary Fig. 9).

Increasing height and biomass to enhance yield may pose the risk of increased lodging. Thus, we leveraged Watkins resources to expand the accessibility of alternative non-*RHT1* semi-dwarfing genes which reduced height, but not yield. We targeted *RHT8* because our NIL data showed the benefits of *RHT8* particularly for Mediterranean environments (Fig. 3f). *RHT8* was introduced into Europe from the Japanese landrace 'Akakomugi'[26] and was genetically defined using the Akakomugi-derived cultivar 'Mara'. Although *RHT8* has been mapped at high genetic resolution[27], breeders do not have access to diagnostic markers for the 'Mara' allele. Although a closely linked gene (*Traes-CSU02G024900*, also known as *RNHL-D1*) also affects plant height[28], this gene is located outside of the genetic interval, and the proposed diagnostic or causative SNP is absent in 'Mara'. We fine mapped *RHT8* to a 0.82-Mb interval containing 23 gene models (Supplementary Tables 27–30). We identified 36 Watkins accessions with the Mara-like haplotype and used haplotype-specific markers derived from novel Watkins SNPs, reducing the *RHT8* mapping interval to a physical distance of 6.7 kb (Fig. 3g). This interval contains two annotated genes in IWGSC RefSeq v1.0, *TraesCS2D02G057800* (unknown function) and *TraesCS2D02G057900* (Photosystem 1 assembly 2), positioned in a head-to-head orientation with only 6 bp between their respective 5′ untranslated regions and 250 bp between their translational start codons. The expression of the two genes is tightly correlated (Fig. 3h): when *TraesCS2D02G057800* expression is high (as it is until mid-stem extension), *TraesCS2D02G057900* expression is low, and vice versa (Fig. 3h; Pearson correlation $r = -0.979$, $P < 0.0001$). These new molecular markers for *RHT8* provide breeders with precise genetic tools to control plant height without negative pleiotropic effects on yield, particularly in Mediterranean environments.

By integrating these findings, we not only discovered modern applications for allelic effects that were left behind in the early stages of wheat breeding (such as chromosome arm 7BL QTLs) but also refined our understanding of established genetic effects such as those of *RHT8*.

## Deployment of landrace variation in breeding

To quantify and accelerate the impact of Watkins diversity on future breeding endeavours, we developed new tools for breeding and systematically introduced potentially beneficial Watkins QTL alleles into the Paragon modern wheat genetic background. This was done through two backcrosses of Paragon with selected Paragon × Watkins RILs carrying the targeted Watkins alleles, resulting in more than 87.5% of isogenic families in which groups of homozygous siblings (2–3 lines for each allele) were used to estimate the Watkins allelic effects. In this way, we successfully introgressed 127 prioritized QTL alleles, represented by a total of 738 NILs, which were used to quantify the breeding value of the introgressed Watkins alleles within the uniform modern genetic background of Paragon. Out of these 127 prioritized QTL targets, 107 originated from AG1, AG3, AG4, AG6 and AG7, which we showed to sample landrace populations that did not contribute to the genomes of modern wheat (Figs. 1c and 4a and Supplementary Table 31). For each chromosome, the enhanced ancestral group diversity beyond AG2/5 is evident (Fig. 4b) and the associated allelic effects are promising for wheat improvement (Fig. 4c). The introgressed segments included 44,338 LD-based haplotypes unique to Watkins, bridging the gap between landrace diversity and modern breeding (Fig. 4d).

To assess the phenotypic value carried by these introgressions, we conducted extensive field-based evaluations of the isogenic families. This was done in at least 3 locations or years (Supplementary Table 15), in replicated field experiments with lines grown in sufficiently large plots (6 m$^2$) for agriculturally realistic yield assessment. We collected detailed phenotyping data for 11 traits from these experiments (encompassing the 127 QTL targets) over a total of 9 years (Fig. 4c). Additive main and multiplicative interactions (AMMI) analysis provided a measure of environmental stability for Watkins allelic effects, so that breeders using these lines as pre-breeding germplasm base their selection on the most environmentally robust effects (AMMI means for each trait are shown in Supplementary Table 26). We found a very high level of phenotypic variation in key traits controlled by the Watkins haplotypes. By comparing the allelic effects between Watkins and Paragon in an isogenic Paragon background, we found variation in heading date ranging from 6 days earlier to 2 days later, height effects varying from 5 cm reduction to 13 cm increase, and yield increases of up to 0.91 t ha$^{-1}$, when considering statistically significant individual allelic effects (Supplementary table 26). Our allele prioritization for breeding included the consideration of deviation from trait antagonism (Fig. 2g, with detail in Supplementary Table 26). An example of the use of this trait prioritization is given by the five haplotypes we found that confer significant increases in grain protein (an important quality trait for bread making), three of which do so without a negative impact for grain yield. This knowledge hugely increases the appeal of these haplotypes as breeding targets, given the often-observed negative correlation between grain yield and protein content. Thus, this study provides a pivotal resource for the selection of Watkins alleles in breeding in which the primary target trait can be considered together with its pleiotropic effects (Supplementary Table 26). This dataset does not provide the information needed to predict the total genetic gains made possible by this work as the genetic additivity of these allelic effects has not been comprehensively tested. However, their arithmetic sums are helpful in conveying the breeding potential of this resource. Looking at the sum of significant allelic effects (see Supplementary Table 26) for yield components in this way reveals a 4.5 t ha$^{-1}$ increase in grain yield, an increase of 11,500 grains per m$^2$, and an increase in thousand grain weight of 55.6 mg. These are valuable breeding targets validated in isogenic backgrounds, but realizing the benefits by combining these alleles within breeding pedigrees will be dependent on genetic interactions, which fall outside the scope of the current study.

Our comprehensive characterization of modern wheat as a mosaic collection of Watkins IBS segments, combined with the identification of LD haplotypes between Watkins and modern wheat, offers a unique framework for designing and developing new wheat varieties. To enable the selection of any haplotype for this purpose, we identified diagnostic 'tag' SNPs along with the corresponding molecular markers for each of the 1.7 million haplotypes (Supplementary Table 32). To demonstrate the efficacy and utility of these markers, we selected tag markers for two QTLs: 5A awn inhibitor (*B1*)[29] and 7A coleoptile colour (*Rc*) (Supplementary Fig. 10). Using KASP assays, we genotyped 382 independent accessions from the Biotechnology and Biological Sciences Research Council small grain cereals collection (Supplementary Table 33). These markers proved to be highly effective for enriching target haplotype selection, resulting in correct predictions of the 2 phenotypes in 87.4% and 93.7% of accessions, respectively.

## Discussion

It is crucial to discover (or re-discover) and characterize genetic resources that improve crop performance under challenging environmental conditions. Arthur Ernest Watkins first described the bread wheat landraces used here in *The Wheat Species: A Critique* in 1930[30,31]. Although nearly a century has passed, we can now use genomics to fully realize the potential of these invaluable genetic resources. Particularly, our whole-genome re-sequencing of the Watkins landrace collection showed that five of the seven Watkins ancestral groups are phylogenetically isolated from modern wheat varieties (Fig. 1b). By combining structured gene discovery populations, genomic data and extensive field experimentation with in-depth phenotyping, we confirmed that

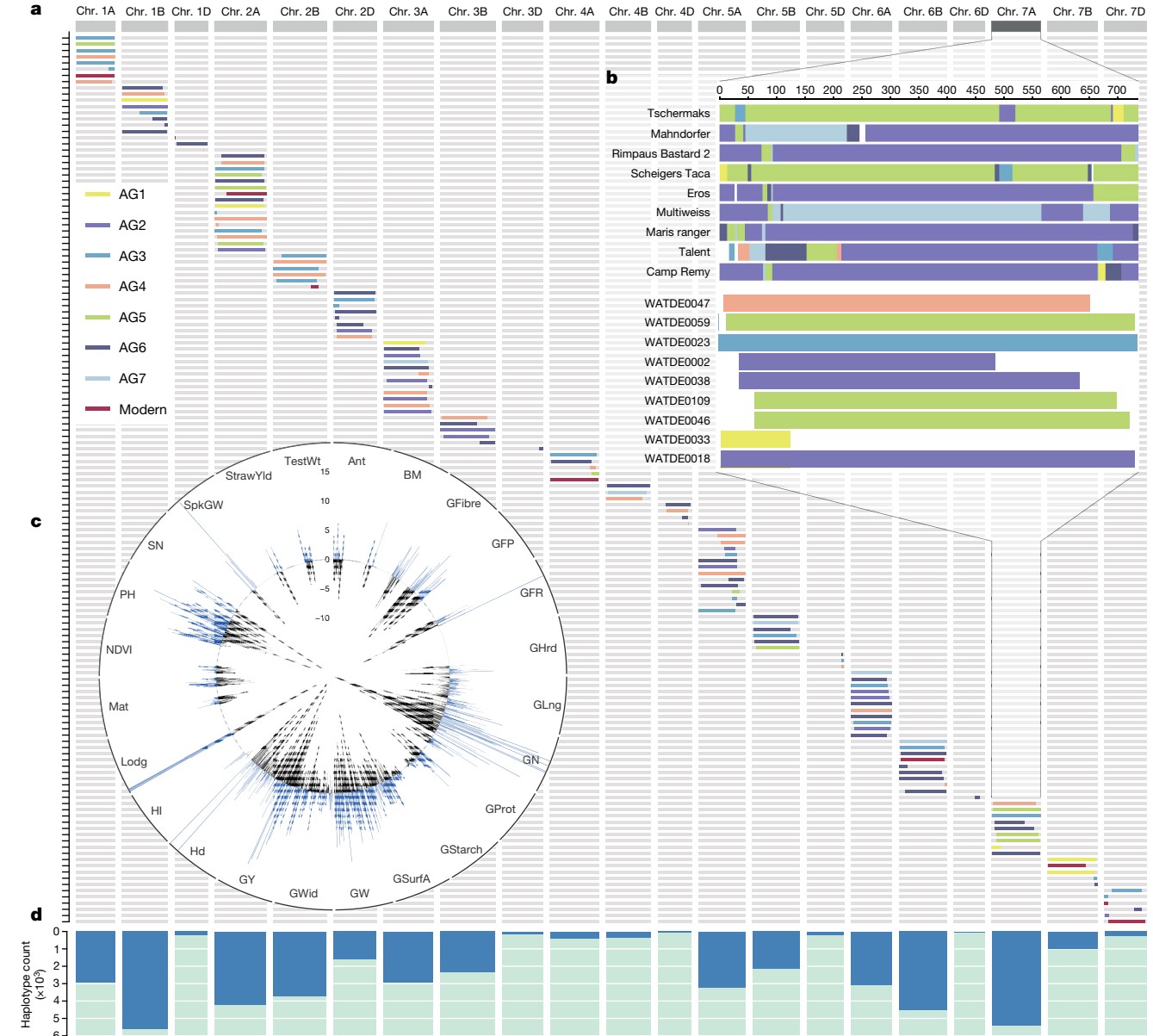

**Fig. 4 | Validation of the breeding value and delivery of target segments.**
**a**, Introgressed target segments of Watkins in 143 Paragon NILs based on QTL prioritization. Colours of segments represent the corresponding ancestral groups (AG1–AG7) of the donor parents. **b**, Inset highlighting one example for chromosome 7A, showing the composition of old breeding cultivars (top, mainly composed of AG2/5) and of NILs (bottom, also containing AG1, AG3 and AG4 representation). Details on each segment are provided in Supplementary Table 31. **c**, Phenotypic effects (AMMI means) of the genetic substitution derived from multi-environment screening for the 127 introgressed loci on 24 traits (marked on the perimeter; for trait abbreviations see Supplementary Table 13). Percentage increase or decrease in Watkins allele effects compared with Paragon are shown according to the vertical scale at 12 o'clock. **d**, Count distribution along each chromosome of the number of Watkins-unique haplotypes (represented by blue bars) that were introgressed into the modern wheat (Paragon) by backcrossing.

many of the haplotypes restricted to phylogenetically isolated Watkins ancestral groups have beneficial effects on yield potential, adaptation, human nutrition and disease resistance.

Our development of a whole-genome sequence wheat haplotype map based on whole-genome sequence, coupled with a haplotype–phenotype association allowed us to estimate frequencies of beneficial alleles and assess their functional significance by association with high-resolution NAM QTL. To introduce novel and useful landrace diversity into the landscape of modern wheat breeding, we transferred 127 prioritized QTL targets into a single modern elite wheat variety. These introgressed segments carry 44,338 Watkins-unique haplotypes which were evaluated within the modern breeding context for the first time.

The breeding value confirmation conducted here is guiding the stacking of beneficial Watkins haplotypes into new cultivars for further evaluation across environments.

The integrated Watkins resources described here represent a major step forward for cereal research and breeding. Even in species such as rice and maize, which have transitioned into the post-genomics era well ahead of wheat, there is currently no publicly accessible resource in a project that encompasses large-scale genomic analysis (which identified 261 million SNPs and 17 million indels) in primary germplasm resources integrated with comprehensive field-based phenotyping. This includes the development of extensive genetic association mapping populations, the construction of both the LD-based haplotypes

and the IBS-based long-range haplotypes, and the validation of QTL effects, tagged with diagnostic haplotypes.

Our companion papers give a first look at the potential application of the Watkins resources beyond the targets chosen by us. This includes the identification of novel genes for *Septoria* resistance[32] and the first wheat gene conferring resistance to the devastating Bangladesh/Zambia MoT isolate[33] for which the key resistance allele *Pm4f* was only detected in Watkins. The *Pm4f* allele is now being deployed in breeding programmes worldwide as a direct result of the Watkins resources presented here. With human nutrition as a target, a mineral atlas for wheat has been developed with new variants for the breeding of more nutrient-dense wheat[34]. In addition to these trait-centred discoveries, the Watkins resources have been used to develop a new generation of high-density wheat genotyping array[35]. These companion studies exemplify some of the ways in which the Watkins resources can be used by wheat researchers.

Consideration of limitations and the next steps that will facilitate the fullest possible utilization of the Watkins genetic resources raises questions of alternative sequencing or bioinformatic approaches and, crucially, the deployment of variation in registered varieties. Large-scale structural and copy number variations are an important component of genetic variation in wheat but they are not detected using the short-read sequencing technologies deployed in this study. Following the model of the 10+ Wheat Genomes Project[10], and incorporating long-read sequencing technologies[36] would provide significant uplift to the value of the Watkins resources. These variations will be elucidated by such an approach. For breeders, there are still significant barriers to combining novel Watkins alleles in a single variety. New innovations in breeding technology are still required to overcome linkage drag, so that new beneficial alleles can be introduced while maintaining optimal combinations already regionally deployed. It is also the case that most of the genetic gains that we have described have been quantified using NILs in the genetic background of the UK spring wheat variety Paragon which, released in 1998, expresses approximately 70% of the yield potential of modern UK winter wheat varieties. Ongoing breeding efforts will show whether Watkins alleles for yield increase will deliver benefits in the next generation of varieties growing in farmer's fields.

To empower the whole of the global community to accelerate breeding in wheat, we have adhered to the collaborative spirit of the Human Genome Project by making our resources, including germplasm, genomic and phenotypic data, publicly available through the Watkins Worldwide Wheat Genomics to Breeding portal (https://wwwg2b.com/). We aim to promote openness and collaboration that will enable the full potential of this work to be realized, providing resources for extending the use and further development of the Watkins resource.

Our analysis of this remarkable genetic resource from the 1900s underscores the enduring value of collecting and preserving genetic diversity ex situ. The Watkins collection, assembled from diverse regions worldwide a century ago, is now reasserting its global significance. This is exemplified by its introduction since 2019 to China, where it is being cultivated across various regions with extensive phenotyping and crossing experiments. Within the Watkins collection are 118 landrace accessions originally collected from China and now repatriated thanks to long-term ex situ conservation efforts. A similar initiative dates back to 1932, when a Chinese colleague, Shen Zonghan, introduced approximately 1,700 wheat accessions from the John Percival Collection[31], significantly contributing to wheat genetic improvement and breeding in China. The profound legacy left by Watkins has inspired our international collaboration and commitment to open data sharing and knowledge exchange, recognizing the collective benefits to the global community[37]. Although recent policies have restricted international germplasm exchanges[38,39], it is crucial to remember that the challenges faced by humankind transcend these artificial boundaries.

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

[1]Shenzhen Branch, Guangdong Laboratory for Lingnan Modern Agriculture, Genome Analysis Laboratory of the Ministry of Agriculture, Agricultural Genomics Institute at Shenzhen, Chinese Academy of Agricultural Sciences, Shenzhen, China. [2]John Innes Centre, Norwich, UK. [3]Rothamsted Research, Harpenden, UK. [4]State Key Laboratory of Crop Stress Biology for Arid Areas, College of Agronomy, Northwest A&F University, Yangling, China. [5]Functional Genomics, School of Biological Sciences, University of Bristol, Bristol, UK. [6]RAGT, Saffron Walden, UK. [7]Limagrain UK, Bury St Edmunds, UK. [8]School of Life and Environmental Sciences, Faculty of Science, The University of Sydney Plant Breeding Institute, Cobbitty, New South Wales, Australia. [9]Western Sydney University, Richmond, New South Wales, Australia. [10]School of Biosciences, University of Nottingham, Sutton Bonington, UK. [11]Department of Agricultural and Forest Sciences and Engineering, University of Lleida–AGROTECNIO-CERCA Center, Lleida, Spain. [12]Elsoms Wheat, Spalding, UK. [13]Debre Zeit Agricultural Research Center, Ethiopian Institute of Agricultural Research, Debre Zeit, Ethiopia. [14]Wheat Research Department, Field Crops Research Institute, Agricultural Research Center, Giza, Egypt. [15]Limagrain UK, Market Rasen, UK. [16]Syngenta Seeds, Cambridge, UK. [17]School of Biology, University of Leeds, Leeds, UK. [18]Agri-Food and Biosciences Institute, Belfast, UK. [19]Qingdao Agricultural University, Qingdao, China. [20]DSV UK, Banbury, UK. [21]Institute of Field and Vegetable Crops, National Institute of the Republic of Serbia, Novi Sad, Republic of Serbia. [22]KWS UK, Royston, UK. [23]Key Laboratory of Quality and Safety Control for Subtropical Fruit and Vegetable, Ministry of Agriculture and Rural Affairs, College of Horticulture Science, Zhejiang A&F University, Hangzhou, China. [24]ICREA, Catalonian Institution for Research and Advanced Studies, Barcelona, Spain. [25]National Key Laboratory of Wheat Improvement, Peking University Institute of Advanced Agricultural Sciences, Shandong Laboratory of Advanced Agriculture Sciences in Weifang, Weifang, China. [26]The Earlham Institute, Norwich, UK. [27]Egerton University, Njoro, Kenya. [28]Huazhong Agricultural University, Wuhan, China. [29]Center for Desert Agriculture, Plant Science Program, Biological and Environmental Science and Engineering Division (BESE), King Abdullah University of Science and Technology (KAUST), Thuwal, Saudi Arabia. [30]Department of Molecular Biology, Max Planck Institute for Biology Tübingen, Tübingen, Germany. [31]State Key Laboratory of Tropical Crop Breeding, Chinese Academy of Tropical Agricultural Sciences, Haikou, China. [32]These authors contributed equally: Shifeng Cheng, Cong Feng, Luzie U. Wingen. ✉e-mail: chengshifeng@caas.cn; simon.griffiths@jic.ac.uk

## Methods

### Statistical analysis

Statistical analyses were conducted in R software suite (version 4.2; https://www.r-project.org/) unless otherwise stated. The LD and haploblocks were calculated by PLINK (version 1.90 beta)[40,41], the haplotype clustering was performed by HAPPE[14]. Where relevant, statistical tests were two-sided, randomized experimental units were used as replications, multiple measurements of single experimental units were treated as subsamples, and all data was tested for assumptions and corrected accordingly, and described. The phenotypic effects observed in the NILs were used to calculate best linear unbiased estimators (BLUEs) for each NIL from a linear model (lm in R) encompassing the whole trial with replication. The phenotypic estimates for each NIL over several environments were received using AMMI by fitting additive main effects for the NILs and environments by an ANOVA procedure using BLUE and then apply PCA using singular value decomposition to the remaining residuals after the fitting of main effects. Statistical differences between the two different NIL alleles were tested for each NIL family as ANOVA using the AMMI-BLUEs.

### Germplasm collection and glasshouse growing conditions

The 827 accessions from the entire A. E. Watkins Collection of landraces were used, alongside 208 modern wheat lines selected to represent worldwide diversity and to include parents of publicly available populations[11]. Seeds were obtained from the John Innes Centre Germplasm Resource Unit (JIC GRU; http://www.jic.ac.uk/germplasm/). Single seeds were sown in 3.8 cm diameter pots containing peat and sand mixture (85% fine grade peat, 15% washed grit, 4 kg m$^{-3}$ Mag Lime (powdered limestone containing 90% $CaCO_3$), 2.7 kg m$^{-3}$ slow release fertilizer (Osmocote, 3–4 months), 1 kg m$^{-3}$ PG Mix 14-16-18 + TE 0.02% and wetting agent) for 3 weeks growth at 20/17.5 °C day/night temperature with 16 h day length, before transfer to 6 °C day and night temperature with 8 h day length for 8 weeks.

Following vernalization in controlled environment rooms, plants were transferred to glasshouse conditions with automated watering and following 2 weeks further growth, transplanted into 2 l pots containing cereals mixture (40% medium grade peat, 40% sterilized loam (soil), 20% washed horticultural grit, 3 kg m$^{-3}$ Mag Lime (powdered limestone containing 90% $CaCO_3$), 1.3 kg m$^{-3}$ PG mix 14%-16%-18%, N-P-K + TE base fertilizer, 1 kg m$^{-3}$ 'Osmocote mini' 16%-8%-11% 2 mg + TE 0.02%, and wetting agent) for continued development. During plant growth, spikes were bagged to prevent cross pollination and plant material dried naturally. Seeds were deposited in JIC GRU and transferred to the Agricultural Genomics Institute at Shenzhen (AGIS), Chinese Academy of Agricultural Sciences.

### DNA extraction

Genomic DNA was extracted from approximately 50 mg wet weight young leaf tissue of 3-week stage seedlings. Extractions for the Watkins collection used the DNeasy 96 Plant Kit protocol (Qiagen) and extractions for remaining lines used the oKtopure automated plant-based system (LGC Biosearch Technology) following tissue desiccation with silica for 48 h. A bespoke maxi prep protocol was used with the following volumes per sample: 250 μl lysis buffer, 170 μl binding buffer, 20 μl sbeadex suspension, 300 μl PN1 wash buffer, 300 μl PN2 wash buffer, 300 μl PN2 wash buffer (x3 wash cycles) using 75 μl final elution buffer.

### Whole-genome re-sequencing and quality control

DNA was used for sequencing library construction following the manufacturer's protocols (Illumina). Libraries were 150 bp paired-end with insert size ~500 bp and were sequenced on an Illumina NovaSeq 6000 at Berry Genomics (954 accessions), Beijing, in 2018, and also on DNB-SEQ Platform at BGI group (90 accessions). A total of over 200 TB raw data was generated, producing, on average, 193 Gb raw reads for each accession (Supplementary Table 1). The raw data was filtered using the following parameters (fastp[42] v0.20.0: -f 9 -F 9 -l 80 -g); adapter sequences were removed; reads of N number ≥5 and reads where the base quality ≤15 exceeds 40% were discarded; 9 bp in the front of reads were trimmed of and reads with a length ≥80 bp were retained; After removing these low-quality and adapter-containing reads, an average of ~185.14 Gb of clean data (~12.73× coverage) was retained for each accession.

### Reads mapping, variant discovery, quality control and SNP annotation

The clean reads were mapped to IWGSC RefSeq v1.0 using BWA-MEM (v0.7.17)[43] with default parameters. Non-unique mapped and duplicated reads were excluded using SAMtools (v1.9)[44] and Picard (v2.20.3-SNAPSHOT; http://picard.sourceforge.net), respectively. SNP and indel calling were performed by GATK (v4.1.2)[45]. A total of 720,048,179 raw variants (SNP or indel) were identified from GATK, including 668,764,660 SNPs and 51,283,519 indels.

Four main steps of variant filtering and quality control for both SNPs and indels were contducted, corresponding to Supplementary Table 2. First, only the bi-allelic variants were retained, including 634,873,707 SNPs and 44,525,746 indels. Second, variants were filtered based on the parameters recommended by GATK (QD < 2.0||FS > 60.0||MQ < 40.0||MQRankSum < −12.5||ReadPosRankSum < −8.0||SOR > 3.0. Indels Filter criteria: QD < 2.0||low_QD||FS > 200.0|| high_FS||ReadPosRankSum < −20.0||low_ReadPosRankSum) after which a total of 411,400,604 SNPs and 42,415,907 indels were retained. Third, variants were filtered by inbreeding coefficient ($F$). $F$ is computed as: $F = 1 − (H_{obs}/H_{exp})$. $H_{obs}$ is the frequency of heterozygous calls, and $H_{exp} = 2p(1 − p)$, in which $p$ is the frequency of non-reference allele (or reference allele). The median value of $F$ ($F_{median}$) for each chromosome is calculated using the SNP site of $F > 0$ and minor allele frequency (MAF) > 0.05. The maximum observed heterozygous frequency ($H_{obs\_max}$) is computed as: $H_{obs\_max} = 10 × (1 − F_{median}) × H_{exp}$. The sites of $H_{obs} > H_{obs\_max}$ were discarded. A total of 261,659,890 high-quality SNPs (dataset 1) and 17,279,131 high-quality indels were retained. Finally, the SNPs with missing rate >20% and MAF < 0.01 were discarded. A total of 90,750,089 common SNPs (high frequency) were retained as high-quality SNP dataset (dataset 2).

SnpEff (v4.3t)[18] was used for annotating and predicting the genome structural position and functional effects of identified SNPs and indels. SNPs were annotated as exonic, intronic, splicing region, upstream, downstream, intergenic and 3' and 5' untranslated region variants. Exonic variants can be further divided into synonymous and nonsynonymous variants, and the latter included missense variants, stop loss, stop gain, start loss, start gain and stop retained. Intron variants can be categorized as splicing donors, splicing acceptors and others.

### Identification of gene CNVs

Considering the limitations of using short reads for CNV identification, only the CNVs of the wheat protein-coding genes were calculated in this study. Five steps were implemented for the identification of gene CNVs. (1) Calculation of read depth for each gene, sequenced in each accession, based on the properly mapped reads. This is referred to as the absolute read depth for each gene. (2) Optimal correction of the absolute value for read depth variation (RDV). To account for highly similar genes in the reference genome, such as paralogues arising from recent duplication events, we performed an all-vs-all coding sequence alignment using BLASTN. Genes that met specific criteria (fewer than five gaps and fewer than five mismatches) were classified as recently duplicated. For depth calculations, these highly similar genes were collapsed into a single representative gene in the reference genome. Specifically, the depth values of duplicated genes were summed up together. This approach aims to minimize the depth bias introduced by recent gene duplications in the reference genome[46]. (3) Normalization

for each accession. Considering the slight variation for the sequencing depth for each accession, we divided the corrected read depth for each gene in step (2) by the average sequencing depth for each accession. This is the relative RDV. (4) GC bias correction. There were GC bias in Illumina short-read sequencing technology despite all libraries being PCR-free. We analysed the read depth distribution for the GC content within the wheat genome (1,000-bp windows for GC content from 0% to 100%) and corresponded the estimated read depth for each level of GC content for each gene to this distribution, which was divided by the overall read depth of the overall GC content. This resulted in the GC content correction value for each gene, to avoid and correct the GC bias for read depth[47]. (5) Correction of RDV for genomic regions with insertions or deletions in the genome reference. To explore the population-unique and shared CNVs, the number of accessions with different copy numbers, such as [0–0.25], [0.25–0.75] or [0.75–1.25], was calculated for each gene in both Watkins and modern populations.

### Ensuring correspondence of sequenced accessions with existing public genotypic data

To ensure consistency of sequenced samples, sequence data were compared to publicly available genotyping data using the Wheat Breeders' Array[48]. The probe set of the Breeders' array (https://www.cerealsdb.uk.net/cerealgenomics/CerealsDB/Excel/35K_array/35k_probe_set_IWGSCv1.xlsx) was used to generate two allele-specific sets of $k$-mers ($k$ = 31) for each SNP of the array. We also built $k$-mer databases from the sequence data of each Watkins line. The allele for each SNP in each Watkins line was determined using presence or absence of allele-specific $k$-mers in the $k$-mer databases. This resulted in a genotype profile for each Watkins line. Next, the new genotype profile was compared to existing data generated using the Wheat Breeders' Array. This analysis resulted in a 100% match between the sequenced accessions and the existing public genotypic data for these accessions. Scripts for the procedure and details on the pipeline are available at https://github.com/JIC-CSB/WatSeqAnalysis/tree/master/qc_vs_iselect.

### LD analysis and construction of the wheat haplotype map

To obtain a core SNP set from the filtered SNP set described above, a two-step LD pruning procedure were conducted as previously done in rice[49]. First, SNPs were removed by LD pruning with a window size of 10 kb, window step of one SNP and $r^2$ threshold of 0.8 using PLINK[40]. Second, another round of LD pruning with a window size of 50 SNPs, window step of one SNP and $r^2$ threshold of 0.8 was performed. About 10 Mb SNPs were retained after the two-step LD pruning (dataset 3). The construction of the wheat haplotype map (HapMap) consists of two parts, the population LD-based haplotype map, and the Identity-by-State in python (IBSpy) $k$-mer based large-scale long-range haplotypes segments[50].

**Population LD-based haplotype analysis.** First, the SNP dataset 2 was phased by Beagle (v 21Apr21.304)[51]. Using this phased dataset, haplotype blocks were identified using PLINK[40] with the parameters (--blocks no-pheno-req --blocks-max-kb 1000 --geno 0.1 --blocks-min-maf 0.05). To merge the adjacent blocks that might still maintain strong LD into larger ones, the $D'$ statistic value were calculated for all of the SNPs (dataset 1) of every two adjacent blocks. If the lower quartile (Q1) was larger than 0.98, the two adjacent blocks were merged. The program HAPPE[14] was used to identify haplotype clusters (haplogroups) for each block based on SNPs from dataset 1.

**k-mer based IBS long-range haplotype analysis.** We devised a systematic approach for conducting $k$-mer based IBS approach for long-range haplotype analysis and reconstructed modern wheat genomes using the Watkins lines. The methodology comprised of the following steps:
(1) Generation of $k$-mer matrices and variation analysis: We initiated the analysis by utilizing the kmerGWAS pipeline[52] (https://github.com/voichek/kmersGWAS) to produce a $k$-mer matrix ($k$ = 31) for our dataset containing 1,051 wheat accessions. Concurrently, we integrated the chromosome-level genome assemblies from the 10+ Wheat Genomes Project, including Arina*LrFor*, Chinese Spring, Jagger, Julius, LongReach Lancer, CDC Landmark, Mace, Norin 61, CDC Stanley, and SY Mattis. Employing the IBSpy pipeline (https://github.com/Uauy-Lab/IBSpy), we computed the $k$-mer variation matrix ($k$ = 31) for each reference assembly vis-à-vis the 1,051 accessions, utilizing non-overlapping 50-kb step windows.
(2) Transformation of $k$-mer variations to IBSpy values and haplotype assignment: Next, we converted the $k$-mer variation matrix into IBSpy variation values. We then conducted haplotype assignment for distinct non-overlapping 1-Mb windows. We applied the affinity propagation clustering technique[53] (with a window size of 1 Mb) based on IBSpy variation values computed from 20 consecutive 50-kb windows. To enrich the IBSpy values utilized in clustering, we incorporated IBSpy values derived from syntenic regions across the 10+ Wheat Genomes Project references.
(3) Pedigree tracking and genome reconstruction: pedigree tracking was employed to track the ancestry of each modern cultivar within the Watkins collection. Building upon the haplotype assignments in non-overlapping 1-Mb windows, we reconstructed each cultivar's genome using extended matched haplotype blocks from the Watkins lines, employing an in silico strategy featuring a minimum tiling path approach.
  a. Haplotype comparison and identification of progenitor: on a per-chromosome basis, we began by comparing the haplotypes of a cultivar's first window with the corresponding chromosome's haplotypes in each Watkins line. The length of matched haplotype windows (MHWs) from each Watkins line was noted. The Watkins line with the longest MHW was identified as the potential progenitor contributing to that genomic region. Subsequent MHWs from other Watkins lines were discarded.
  b. Iterative process and longest MHW recording: the comparison process was iterated from the second window onward. At each step, we identified the longest MHW and its associated Watkins line. This iterative comparison continued window by window until covering the entire chromosome.
  c. Refinement and minimum tiling path construction: we then aligned the physical positions of identified MHWs from each window. Overlapping MHWs were removed, retaining those forming the minimum tiling path for reconstructing the chromosome. The reconstructed path allowed observation of the mosaic composition originating from the Watkins lines. This strategic process delineates the closest Watkins relatives at a 1-Mb resolution for any given genomic region within a modern cultivar. The contribution percentage of each Watkins line to a cultivar was quantified as the ratio of its total MHWs to the entire genome windows.

### Population structure analysis

**Phylogenetic tree and ADMIXTURE.** For the Phylogenetic genetic analysis, neighbour-joining tree and maximum likelihood tree were constructed for the genome-wide 4DTv (fourfold degenerate synonymous site) using rapidnj (version 2.3.2)[54] and iqtree (version 1.6.9)[55], respectively. One thousand bootstrap replications were performed for each tree. Interactive Tree of Life (iTOL) was used to visualize and annotate these trees (https://itol.embl.de/)[56]. To explore and obtain the accurate population structure of the Watkins collection, a pipeline was developed to deal with the common introgressions in the wheat genome. First, the genetic distance between all pairs of accessions was calculated in 5-Mb sliding windows (dataset 2) using vcf2dis software (https://github.com/BGI-shenzhen/VCF2Dis). Moreover, the geographic distance of each pair of accessions was calculated, based on the longitude and latitude information, using the R package geosphere. Then, the correlation between the genetic and geographic distances

was calculated for each 5-Mb window using the R package corr. For 88.97% of introgression blocks with known resource the correlation was under 0.07, indicating that this value has potential to exclude introgressions. Therefore, SNPs located in windows with correlation value less than 0.07 were discarded to reduce background noise. The remaining SNPs were used to quantify the genome-wide population structures using ADMIXTURE[9].

**t-SNE and PCAs.** For the haplotype matrix, we first imported the data into a Python environment and then transformed the matrix into a one-hot encoded format using the OneHotEncoder class from the sklearn.preprocessing module. Subsequently, we used the PCA class and the TSNE class from the sklearn.decomposition and sklearn.manifold modules, respectively, to perform PCA and t-SNE. For the SNP matrix, VCF datasets were converted into a numerical matrix. In this conversion, a value of 0 represents a reference allele, 1 represents heterozygosity, 2 indicates an alternative allele, and −1 is used for missing data. We then applied the PCA and t-SNE methods as described above.

## Genetic diversity and population differentiation

In consideration of the deviating effects of missing rate and MAF on genetic diversity, dataset 1 (without missing rate and MAF filtering) was used to calculate the number of SNPs, number of indels and π of accessions in different populations and countries. These calculations were performed in non-overlapping 2 Mb windows across the whole genome using PLINK[40]. For genic diversity, the number of total SNPs, population-unique SNPs and allele frequency among different populations within each gene were calculated. To further evaluate the populations differentiation among Watkins groups and modern variety, plink--cluster was used to calculate the identity by state distance for each pair of accessions[41]. The allelic diversity, haplotype clustering and cataloguing, and CNV diversity according to the RDVs were analysed and visualized with HAPPE[14].

## Field experiments for Watkins collection

We conducted field experiments to assess the phenotypic diversity of the Watkins collection (the natural population, Supplementary Fig. 8).

**UK field experiments for Watkins collection.** The Watkins collection was grown at the JIC Field Experimental Station (JI) (Bawburgh, Norfolk; 52.628° N, 1.171° E) in 2006 in unreplicated 1 m² plots under low nitrogen input as previously reported[57]. Experiments were repeated at JIC in 2010 and 2014. Phenotypes measured were: 2006: heading date, plant height and growth habit; 2010: presence of awns, heading date, thousand grain weight, grain width and grain length; 2014: heading date, kernel hardness and coleoptile colour. Details on phenotype measurements are given in crop ontology format in Supplementary Table 13.

**Chinese field experiments.** The Watkins collection, alongside 208 contemporary cultivars, were grown and phenotyped in diverse geographic locations throughout China. These sites were: Shenzhen city (22.597° N, 114.504° E, seasons 2021, 2022 and 2023), Guangdong Province, southern China; Ezhou (30.386° N, 114.656° E, season 2023), Hubei Province, central China; Nantong (32.268° N, 120.759° E, season 2023), Jiangsu Province, southeast China; Tai'an (35.987° N, 116.875° E, season 2023), Shandong Province, northern China; Quzhou (36.863° N, 115.016° E, season 2022), Hebei Province, northern China; and Harbing (45.830° N, 126.853°E, season 2023), HeiLongJiang Province, northern China. All trials were hand planted in the autumn (mostly November), with the exception of Harbing, where sowing took place in March. Plants were grown in 1.2 m or 2 m long rows using an augmented plot design with 50 or 100 plants per block with three check varieties, with the exception to Shenzen 2020 and Ezou 2022 where a factorial split-block design with two nitrogen treatments and two replicates were grown. Plants were phenotyped for a broad range of traits spanning lodging,

height, tillering, phenology, disease resistance, and various morphological traits, alongside yield and biomass components. Following harvest, yield component traits including spikelet number and grain morphometric traits were measured.

**Egyptian field experiments for Watkins collection.** A diverse subset of 300 Watkins bread wheat landraces and 20 modern lines were grown at four agricultural research stations in Egypt: Sakha (31.0642° N, 30.5645° E), Nubaria (30.6973° N, 30.66713° E), Gemmiza (30.867° N, 31.028° E) and Side (29.076° N, 31.097° E). Growing season was from November to the end of May with harvest in 2020, 2021 and 2022. The wheat lines were grown in 3.5 m rows and hand harvested. Fertilizer applications were before sowing phosphorus (200 kgP ha⁻¹) and potassium sulfate (50 kgK ha⁻¹); and three doses of urea (in total 300 kgN ha⁻¹) at sowing, 30 days post-sowing, and at the tillering stage. Phenotypes recorded (as the mean of ten plants) were: growth habit, plant height, heading date, number of kernels per spikes, grain weight and maturity date. Rust scores were taken at the early dough stage as host responses and rust severity.

## RIL population development and analysis

**Construction of bi-parental populations.** Bi-parental populations with diverse Watkins landrace parents were developed as described[2]. Initial crosses of Paragon (female) to Watkins landrace (male) plants were advanced to $F_4$ using single seed descent. In total, we developed 109 populations using 107 different Watkins accessions, resulting in 10,259 RILs. Of these, 73 RIL populations have been phenotyped to date (Supplementary Table 12 and https://www.seedstor.ac.uk/search-browseaccessions.php?idCollection=47).

**Genotyping RIL populations.** Early in the project, genotyping was conducted with KASP and SSR markers followed by genetic map construction as described. This was the case for the majority of the populations, see Supplementary Table 12, column 'Genetic map type'. Later, genotyping was done with the Wheat Breeders' Array[11].

**Genetic map construction.** Genetic map construction was carried out using the R package ASMap (version 1.6) as described[58]. The genotype scores can be retrieved from CerealsDB (https://www.cerealsdb.uk.net/cerealgenomics/CerealsDB/array_info.php).

**QTL discovery.** QTL mapping was conducted in R (v3.6.1) using package qtl (v1.5)[59,60] as described, taking cross type (RIL) and generation number ($F_4$ or $F_5$) into account. The QTL model used a significance threshold calculated from the data distribution. A first QTL scan, using Haley−Knott regression, determined co-factors for the second scan. The second scan by composite interval mapping identified QTL at a significance level of 0.05 taking the co-factors into account. The resulting QTL with a LOD score equal or larger than 2.0 are listed in Supplementary Table 17.

## Field experiments using RIL populations

Trials were drilled around mid-October and harvested end of July to late August and grown with standard farm management[61,62]. Seventy-three bi-parental RIL populations were grown in field experimentation trials over ten years between 2011 and 2020 at the JIC Field Experimental Station (JI) (Bawburgh, Norfolk; 52.628° N, 1.171° E) in randomized unreplicated 1 m² multiplication trials with low nitrogen input (approx 50 kgN ha⁻¹). A subset of 18 populations were grown in a randomized block design (3 replicates, 1 m² plots) at Rothamsted Experimental Farm (RH) in southeast England (51.8100° N, −0.3764° E) at 2 different nitrogen levels over 7 years (2012–2018). Nitrogen supply was taken to be the sum of the amount of mineral Nitrogen in the top 90 cm of soil measured late winter each year (before the first fertilizer N application) and the amount of N fertilizer applied (either as ammonium nitrate or

ammonium sulfate). Nitrogen fertilizer applications (50 or 200 kgN ha$^{-1}$) were made late February to May, with a split application to plots receiving 200 kgN ha$^{-1}$. A set of 15 populations were grown at Bunny Farm, University of Nottingham, Nottinghamshire (SB) (52.8607° N, −1.1268° E) in randomized replicated 1 m$^2$ plots under 2 different nitrogen conditions as in RH over 4 years (2012–2015). Details on which population was grown in which season are given in Supplementary Table 15. Seed sources for the JI trials were glasshouse seed, and for the other trials the JIC field multiplied seeds. Field experiments at RH were targeted to specifically assess grain yield and nitrogen use efficiency (NUE). Above-ground nitrogen uptake was calculated from the sum of nitrogen in straw and grain at final harvest. These data were calculated from grain and straw dry matter yield; both recorded by plot combine harvester at Rothamsted, grain by combine and straw from a pre-harvest grab sample at Nottingham. Harvest index (ratio of grain dry matter to above-ground dry matter) was also calculated, and grain and straw nitrogen concentration, measured on samples taken at final harvest and measured by near Infrared spectroscopy using in-house calibrations. NUE was calculated using published methods[62]. Overall, phenotypes recorded were: JI: heading date, plant height, grain yield, grain weight, grain length, grain width, and grain surface area; RH and SB: anthesis date[63], crop height, grain yield, straw yield, grain and straw nitrogen concentration. RH also carried out canopy maturity, grain and straw moisture content and on targeted populations mineral analysis by Inductively coupled plasma optical emission spectrometry. Further phenotypes were calculated from the direct measurements—for example, grain number, harvest index, nitrogen uptake, NUE, grain fill rate, grain protein concentration and grain protein deviation. Details on phenotype measurements are given in crop ontology format in Supplementary Table 13. Four RIL populations targeted for yellow rust resistance were grown in six experiments by commercial partners as winter drilled, unreplicated 1 m$^2$ plots and scored for yellow rust incidence. Locations were: Limagrain UK (Rothwell Cherry Tree Top, 53.4882° N, −0.25437° E), RAGT (Ickleton 52.063481° N, 0.170783° E and Gasworks 52.081978° N, −0.169570° E), Elsoms Wheat (Weston, 52.842426° N, −0.079539° E), KWS (Fowlmere, 52.0533° N, 0.03551° E). Details on seasons and populations in Supplementary Table 15.

## NIL development

**Construction of the NIL library.** QTLs for putative advantageous alleles from landrace parents were targeted for NIL development. For each QTL, a RIL was selected that carried the landrace allele at all markers of the QTL confidence interval and that carried a maximum number of Paragon alleles at the remaining loci. Selected RILs were crossed with cultivar 'Paragon', followed by two backcross steps also to Paragon (that is, > 87.5% Paragon background), ensuring the heterozygous state of the confidence interval region in the selected parent for the next crossing step by marker-assisted selection. All crosses were conducted with Paragon as pollen donor and plants were grown under standard glasshouse conditions. For the final step of the NIL development, BC$_2$F$_1$ lines were self-pollinated and BC$_2$F$_2$ lines homozygous for the confidence interval region for both, either the landrace parent or Paragon, were selected by marker-assisted selection to be used as NIL pairs or families for the QTL validation (an overview over the crossing scheme is given in Supplementary Fig. 11). A more complete genotype of the BC$_2$F$_2$ lines was determined using the 35k Wheat Breeders' Array[58]. NIL development progressed in annual sets, called Toolkit (TK) sets, starting in 2012. We report here on sets developed up to 2017 (TK1 to TK5, details of selected QTL, number of NIL families and individual NILs are given in Supplementary Table 31). BC$_2$F$_2$ NIL seeds are available from https://www.seedstor.ac.uk/search-browseaccessions.php?idCollection=40.

## Field experiments for NIL families

The performance of BC$_2$ NIL pairs (> 87.5% isogenic background), or families of NILs coming from a cross with opposing parental alleles in the targeted QTL region, were compared (Supplementary Fig. 11). After initial multiplication trials at JI (1 m$^2$ plots) further field trials were conducted as yield trials (replicated 6 m$^2$ plots in randomized block design) at JI, RH and SB under the same conditions as the RIL trials. The NIL families were grown in yearly TK sets between 2015 and 2022 (see details in Supplementary Table 15). Phenotypes recorded were similar to the RIL trials, and raw data can be downloaded from https://grassroots.tools/fieldtrial (Search term: "DFW Academic Toolkit Trial"). NIL family performance for measured phenotypes were assessed using simple ANOVA in individual years (Supplementary Table 26). The performance over all seasons and environments was assessed using AMMI (Supplementary Table 26). Specific NIL field experiments to assess the grain yield potential were conducted for NIL lines WL0019 and WL0026 at 15 different locations. Of those trials, six were the general NIL trials reported above; another six trials were part of grain yield germplasm trials by commercial UK breeders, grown in triplicated randomized 6 m$^2$ plots, harvested in 2017 (details and raw data can be downloaded from https://grassroots.tools/fieldtrial, search term: DFW-BTK-H2017) and three similar trials at JI with harvest in 2018, 2019 and 2020.

In total, six trials for *RHT8* NILs were conducted with a NIL carrying *RHT8* and the recurrent NIL parent Paragon, with two seasons at each of the three locations: JI, the University de Lleida, Spain (41.36464° N, 0.48197° E) and IFVC, Novi Sad, Serbia (45.20° N, 19.51° E). All trials were grown in a randomized block design together with other varieties with three replicates (JI) or five replicates (the other sites). All trials were autumn drilled and harvested in July at JI (2020, 2021) and in June at the other sites (2021, 2022). Plot size was 6 m$^2$ at JI and 5 m$^2$ at the other site. The average temperature was recorded at all three sites.

## Quantification of trait relationships

The trait–trait relationship analysis was performed using the phenotypic dataset of the TK trials, collected at multiple locations in multiple years. For individual years and locations, the Pearson correlation coefficient ($r$) was calculated using R package corr. Then, the median and mean of the $r$ values over several trials was used as the final result of correlation coefficient. These results were visualized using R package corrplot.

Trait trade-off plots were created for all traits together as PCA plots and in individual plots and for the trait pairs: grain weight (GW) × grain number (GN); grain yield (GY) × grain protein (GP). The individual plots are based on the effect direction and effect size of the introgressed alleles from the 127 NIL families, where the majority of introgressed alleles come from Watkins and are compared to the Paragon allele. The trait effects were calculated using the AMMI method as described. In the trade-off plots, the traits effects are shown as filled circles on a horizontal bar, which represents a sliding scale of the allele effect as percentage of the mean trait value. A positive allele effect will result in a positive value. For the trade-off plots, two traits are shown together, with the bar for the first trait being on top and for the second trait at the bottom. The circles are shown in colour if the effect was statistically significant, and in grey otherwise (Fig. 2g). The PCA bi-plots on the left of the individual trade-off plots are based on a PCA calculated using the phenotype estimators from the AMMI, employing the package princomp in R. Specific trait pairs are highlighted by bold arrows in the PCA plots to show their overall relationship. This contrasts the individual trade-off plots, which show exception to this overall trend.

## GWAS from Watkins collection

The markers used for GWAS of Watkins collection were ~10 Mb core SNPs in dataset 3. Extreme outlier values of phenotypic data were removed. Based on these, we performed GWAS using GEMMA (v0.98.1)[64] with parameters (gemma-0.98.1-linux-static -miss 0.9 –gk -o kinship.txt and gemma-0.98.1-linux-static -miss 0.9 -lmm -k kinship.txt). In-house scripts programmed in R were used to visualize these results.

## NAM imputation and NAM–GWAS

**Pre-processing for skeleton marker.** The accessions of NAM/RIL populations were genotypes using the 35k Wheat Breeders' Array[48]. To obtain a high-quality SNP dataset, we used marker flanking sequences to align to the reference genome (IWGSC RefSeq v1.0)[8]. Positions and alleles of SNPs that were consistent with the re-sequencing data were considered and of those only markers with polymorphisms between parents were retained.

**NAM Imputation.** The HapMap constructed in this study was the base for NAM imputation. The SNPs of RIL parent were extracted from SNP dataset 1, since the rare alleles in the natural population will not be rare in the RILs. Overall, we used the genotype results of the RILs as the skeleton to predict the genotype of each site in each accession. For each RIL population, the detailed methods are as follows:

Step 1:

(1) Go accross the sequenced SNP sites of each parent.
(2) The SNP locus is in a haploblock: If there is one or more skeleton markers in the block, the RIL genotypes will be filled according to the nearest skeleton marker in the block; If there is no skeleton marker in the block, the RIL genotypes are filled according to the nearest skeleton marker on the chromosome.
(3) The SNP locus is not in haploblock: the RIL genotype is filled according to the nearest skeleton marker on the chromosome.
(4) The method of filling RIL genotypes according to a skeleton marker: for example, marker genotype coding is A, B, H or '-', where A represents the allele from parent 1 and B from parent 2, H represents heterozygosity, and '-' represents a missing allele. Then, for an adjacent SNP site the SNP from the same parent is selected from the SNP matrix.

Step 2: Go through each RIL and follow the procedure from step 1.

Step 3: Merge all RIL groups to generate vcf files using the bcftools merge command.

Step 4: Carry out two rounds of LD pruning (plink --indep-pairwise 10 kb 1 0.8; plink --indep-pairwise 50 1 0.8).

**Percentage of the environmental mean.** To standardize phenotypic values across different environments, we calculated the 'percentage of the environmental mean' for each trait. For each individual trait, the raw mean values were first used to calculate the environmental mean of that trait. Next, the individual trait values were converted into a percentage of this environmental mean. The formula used was: percentage of environmental mean = (individual trait value/environmental mean of the trait) × 100. By doing so, each trait's value was expressed as a percentage relative to the mean trait value in that environment. Importantly, before the calculation, controls and outliers were excluded from the data. This approach allowed us to compare traits on a similar scale, effectively reducing potential bias introduced by the raw numerical values of different traits.

**NAM–GWAS.** Based on the SNP sets generated after NAM imputation, two-step LD filtering was performed: plink --indep-pairwise 10 kb 1 0.8; plink --indep-pairwise 50 1 0.8. Finally, 19,253,188 SNPs were retained. To perform NAM–GWAS the following data was collated for each phenotype:

(1) Phenotypic value of a single year: the phenotypic values of the specific RIL populations measured in that year were combined.
(2) Phenotypic values of years with field experiments with high nitrogen fertilization: the phenotypic values of all RIL populations from those experiments were integrated, using percentage of the environmental mean values.
(3) Phenotypic values of years with field experiments with low nitrogen fertilization: the phenotypic values of all RIL populations in those experiments were integrated, using the percentage of the environmental mean values.

(4) The phenotypes of all years and all environments were combined, using the percentage of the environmental mean values.

## Discovery and functional verification

***RHT8* fine-mapping and gene discovery.** Seventy-two recombinant lines (not to be confused with the 73 Paragon × Watkins RIL populations that make up the NAM used in this study) within the *RHT8* locus (Supplementary Table 28) were used for genetic mapping[27]. These are from the cross: RIL4 (from the 'Cappelle Desprez' × 'Cappelle Desprez (Mara 2D)' population described[65]) with 'Cappelle Desprez'. Seven new KASP markers were designed and used as described[66] based on Watkins SNPs (Supplementary Table 29).

**Genetic dissection for yellow rust.** Seedlings of the Watkins lines were screened with single isolates of *Puccinia striiformis* under cold glasshouse conditions in April 2018. Two single isolates (UK 16/342 and 19/501) belonging to the 'Warrior' race, were tested separately as described[67]. Field screening was conducted in 1 m² untreated plots. Yellow rust scores were: 1, no infection observed; 2, stripe per tiller; 3, two stripes per leaf; 4, most tillers infected but some top leaves uninfected; 5, all leaves infected but leaves appear green overall; 6, leaves appear half infected and half green; 7, Leaves appear more infected than green; 8, Very little green tissue left; 9, leaves dead, no green tissue left. In Egypt, we evaluated a diverse set of 300 Watkins bread wheat landraces and 20 additional control lines for yellow rust disease resistance and agronomic traits under natural field conditions at the Sids, Sakha, Nubaria and Gemmeiza Agricultural Research Center stations (Egypt) during the 2019–2020, 2020–2021 and 2021–2022 growing seasons.

## Development of haplotype-based diagnostic SNPs and design of KASP markers

For each haploblock, we selected SNPs that were able to differentiate all haplotypes within this block, defined as tagSNPs. The detailed process is as follows.

Step 1: Distance calculation of all the SNPs between each pairwise haplogroups. The genotype encoding rules used during the distance calculation are as follows: reference allele (homozygous): −1; alternative allele (homozygous): 1; missing: NA; heterogeneous: 0.

For each SNP site, we first calculated the state of this SNP within the haplogroups. Then the average genotype state for all accessions within its respective haplotype cluster was computed. The state of the SNP in haplotype clusters 1 and 2 were denoted as $s_{h1}$ and $s_{h2}$, respectively:

$$s_{h1} = \frac{1}{n_1} \sum_{j=1}^{n_1} g_{ij}$$

$$s_{h2} = \frac{1}{n_2} \sum_{j=1}^{n_2} g_{ij}$$

Here, $g_{ij}$ was the genotype of the $j$th sample in haplotype cluster $i$, and $n_1$ and $n_2$ were the counts of samples in haplotype clusters 1 and 2, respectively.

Step 2: Calculation of the distance of this SNP position between haplotypes. The Euclidean distance between the average genotype states of two haplotype clusters was then calculated:

$$d_i = (s_{h1} - s_{h2})^2$$

If the SNP fell within a coding region, its weight was quadrupled:

$$d_i = d_i \times 4$$

Step 3: Sort of distances of all the SNPs. The SNPs were sorted based on distances, and the SNP with the maximum distance was chosen as the tagSNP. Within a haploblock, a tagSNP was selected for each pairwise

haplogroup. The union of these selected tagSNPs formed the tagSNP set for the haploblock.

Step 4: This process was repeated across all 71,000 haploblocks in the genome to compile the complete set of tagSNPs.

In summary, for each haploblock, the haplogroups that each accession belonged to were determined in the above HapMap analysis. We further calculated the distances at all SNPs between each pairwise haplogroup. The SNPs with the maximum distance were chosen as the tagSNPs that distinguishes these two haplotypes.

### Germplasm availability

The 827 Watkins single seed derived accessions and their 827 predecessor landrace populations, 208 modern cultivars, 73 RIL mapping populations and 143 NIL families are all available from the John Innes Centre Germplasm Resources Unit (https://www.seedstor.ac.uk/) and the Agricultural Genomics Institute at Shenzhen, Chinese Academy of Agricultural Sciences (https://wwwg2b.com/).

### Reporting summary

Further information on research design is available in the Nature Portfolio Reporting Summary linked to this article.

### Data availability

All whole-genome sequence data has been deposited at the National Genomics Data Center (NGDC) Genome Sequence Archive (GSA) with BioProject accession number PRJCA019636 and GSA accession ID CRA012590. Variation matrix and annotations, wheat HapMap, phenotyping data, genetic maps with genotype scores, association genetics analyses, the developed tagSNPs and KASP markers were deposited in WWWG2B breeding portal (https://wwwg2b.com). IBSpy variations tables, haplotypes, long-range tilling paths, variant files (VCF) and all raw phenotypic data are available online (https://wwwg2b.com/dataAvailable, https://wwwg2b.com/toolIndex/academic and https://opendata.earlham.ac.uk/wheat/under_license/toronto/WatSeq_2023-09-15_landrace_modern_Variation_Data/). Publicly available sequencing data were obtained from SRA accessions SRP114784, PRJNA544491, PRJEB37938, PRJNA492239, PRJNA528431, PRJEB39558 and PRJEB35709 and from the NGDC database project CRA005878. Source data are provided with this paper.

### Code availability

Code associated with this project is available at Github: https://github.com/ShifengCHENG-Laboratory/WWWG2B, https://github.com/Uauy-Lab/IBSpy, https://github.com/JIC-CSB/WatSeqAnalysis/ and https://github.com/pr0kary0te/GenomeWideSNP-development.

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

**Acknowledgements** The authors thank G. Moore and M. Bevan for providing valuable feedback at multiple stages of the project; colleagues for assistance in Watkins field trial and phenotyping work from five experimental stations across China: Z. Zhu, Q. Wang, Y. Song, Y. Zhu and X. Zhang; the John Innes Centre (JIC) NBI Computing Infrastructure for Science group; the JIC Field Trials and Horticultural Services teams for support in field and glasshouse experiments; T. Florio for figure visualization; and the Rothamsted Research farm team and Analytical Chemistry unit for support in field experiments and analytical mineral analyses. This work was supported by the Program for Guangdong "ZhuJiang" Introducing Innovative and Entrepreneurial Teams (2019ZT08N628), the National Natural Science Foundation of China (32022006), the Agricultural Science and Technology Innovation Program (CAAS-ASTIP-2021-AGIS-ZDRW202101), the Shenzhen Science and Technology Program (AGIS-ZDKY202002) to S. Cheng, and the Guangdong Basic and Applied Basic Research Foundation (2020A1515110677) to L.M. The UK work was possible owing to the long-term investment of the UK Biotechnology and Biological Sciences Research Council (BBSRC) in wheat research through Institute Strategic Programme (ISP) grants and longer larger grants: BBSRC LOLA 'Enhancing diversity in UK wheat through a public sector prebreeding programme' (BB/I002545/1); BBSRC ISP 'JIC WISP ISP—Wheat Institute Strategic Programme' (BB/J004596/1); BBSRC ISP 'BBSRC Strategic Programme in Designing Future Wheat (DFW)' (BB/P016855/1); BBSRC ISP 'BBSRC Institute Strategic Programme: Delivering Sustainable Wheat (DSW)' (BB/X011003/1) and for wheat germplasm conservation and global distribution through the Germplasm Resources BBSRC National Capability award (BBS/E/J/000PR8000). S.G. and C.L. also received support from the UK Department for Environment, Food and Rural Affairs (Defra) as part of WGIN phases 3 and 4 (CHO106 and CHO109). This work was also supported by the European Research Council (ERC-2019-COG-866328), the Sustainable Crop Production Research for International Development (SCPRID) programme (BB/J012017/1), the Mexican Consejo Nacional de Ciencia y Tecnología (CONACYT; 2018-000009-01EXTF-00306), the Science, Technology & Innovation Funding Authority (STDF), Egypt-UK Newton-Mosharafa Institutional Links award, project ID 30718 and EG–US cycle 19–project ID 42687.

**Author contributions** S. Cheng and S.G. conceived, designed, coordinated and managed the project. S. Cheng led the genomics, bioinformatics, association genetics analysis, phenotyping activities and crossing experiments in China. C.F. led population genomics, association genetics and bioinformatics analyses under the supervision of S. Cheng, including variants calling and quality control, construction of the LD-based haplotype map, *k*-mer matrix for long-range haplotype analysis, GWAS, NAM imputation and NAM–GWAS, integrated association genetics and quantification of genetic effects and haplotype-phenotypic associations, the development of HAPPE pipelines and the WWWG2B portal, tagSNP and KASP markers development. L.U.W. led quantitative genetics analyses including genetic mapping, QTL mapping, statistical analysis of field trials including AMMI and trait ontology. L.U.W., A.B.R., M.L.-W., M.J.H. and S.G. led the field trial, phenotyping activities and data analyses in the UK. H.C. and M.J. assisted in the bioinformatics analyses including variant calling, LD-based haplotype construction, GWAS and NAM–GWAS. Z.H. built the WWWG2B portal and provided additional computational support. S. Collier, S.O. and R.A. developed

germplasm, performed marker-assisted selection and curated seed stocks. C.L., A.K. and S.G. led the fine-mapping of *RHT8*. B.B., A.K.-S., G.A.S. and S.G. conducted field trials for *RHT8*. A.S., N.C. and L.U.W. led the grain calcium content phenotyping. J.Q.-C., R.H.R.-G. and C.U. developed IBSpy and with Xiaoming Wang performed the long-range haplotype analyses. R.B. and S.B. coordinated field and glasshouse disease phenotyping in commercial settings. L.B. extracted DNA for sequencing. A.B., M.L.-W. and K.J.E. conducted high-density genotyping. K.G. and Xingwei Wang supported GWAS analyses. H.C., X.L. and Bo Song performed population structure analyses. W.X. led CNV identification. Baoxing Song and H.L. provided bioinformatics support. R. Horler and S.T. digitized germplasm, genomic and phenotypic data. B. Steuernagel led bioinformatics work in the UK. U.B., H.S.B., J.K.M.B., C.B., M.B., L.C., A.F.E., P.F., D.F., A.N.H., M.S.I., M.K., W.D., J.L., S.L., D. Schafer, P.T., M.N.W., B.B.H.W. and Q.Y. conducted disease phenotyping and data analyses. M.J.B., M.C., B.C., J.F., O.G., R. Han, J.H., Z.H., L.M., C.M., S.P., C.P., D.P., Z.S., Y.S., D. Steele, A.W., Y. Wei, S.W., Y. Wu, Yawen Xu, Yunfeng Xu and X.Z. conducted phenotyping experiments. A.L. and P.R.S. conducted grain phenotyping experiments. N.C. managed germplasm and coordinated phenotyping of grain traits. D. Sanders and S.H. provided leadership and coordination roles. S. Cheng, C.F., L.U.W., Xiaoming Wang, H.C., M.J., C.U. and S.G. prepared the figures, extended data figures, supplementary figures and supplementary tables. S. Cheng., C.F., L.U.W., C.U. and S.G. wrote the manuscript, with additional help from N.C. and A.B.R. All authors read and approved the final manuscript.

**Competing interests** The following authors are employed in private wheat breeding companies: Limagrain UK (S.B., P.F. and P.T.), KWS (J.L.), DSV (M.K.), RAGT (D. Schafer and C.B.), Syngenta (D.F.) and Elsoms (M.B.). The other authors declare no competing interests.

**Additional information**
**Correspondence and requests for materials** should be addressed to Shifeng Cheng or Simon Griffiths.

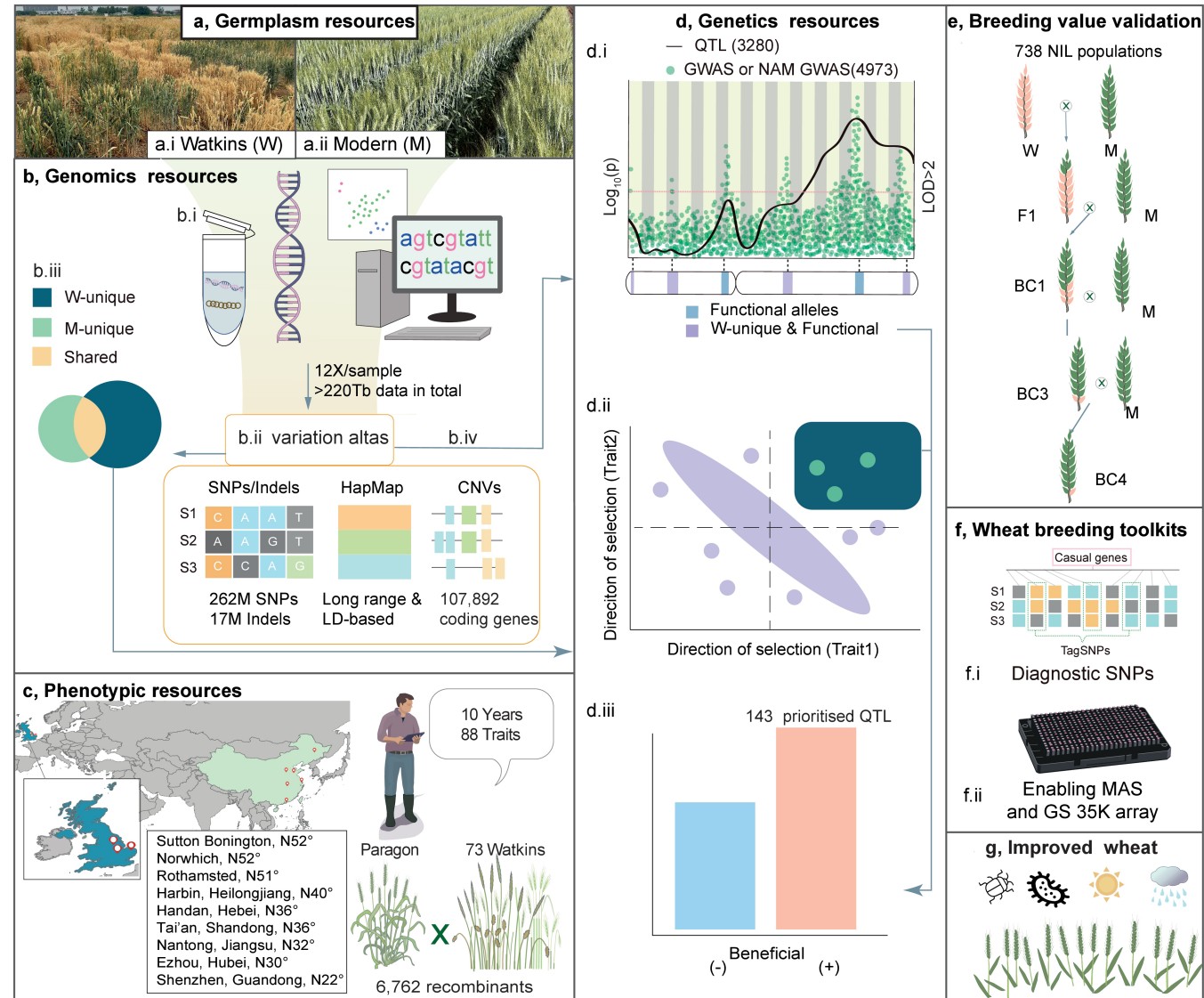

**Extended Data Fig. 1 | Graphical abstract and conceptual strategies. a**, The WWWG2B strategy started with the comparison of variations between the Watkins landrace collection (W, a.i) and the modern wheats (M, a.ii). **b**, By developing an extensive genomics resource, we determined the extent to which landraces carry variants that are not present in modern wheats. **c-d**, Natural and structured populations were then combined in multi-site field-based experiments to identify novel and useful genetic variations not yet deployed in modern wheat. This required genetic dissection by a combination of whole-genome resequencing (b.i), construction of variant atlas and haplotype map (b.ii) and extensive field-based phenotyping (**c**) of next-generation-gene-discovery populations, the NAM RIL segregating populations combined with GWAS (d.i), bi-parental QTL mapping, and haplotype analysis enabled by the development of the advanced genomics and genetics resources shown in b and d. **e**, Alleles with high breeding values and their phenotypic effects (determined by calculating the AMMI means for their selection) were validated and delivered

for use in breeding. **f**, Diagnostic SNPs and KASP markers were designed for assisted molecular breeding. **g**, The overall objective of the Watkins Worldwide Wheat Genomics to Breeding (WWWG2B, http://wwwg2b.com/) consortium is to enable the development of a new generation of modern elite wheat cultivars that are climate resilient. The flow of useful genes and alleles through this process was gated by three prioritisation criteria: 1) the alleles or haplotypes are novel or unique to Watkins landraces or are at very low frequency in modern wheat (b.iii); 2) the Watkins-unique alleles or haplotypes are associated with significant genetic effects (target QTL and MTA) (d.i); 3) haplotype analysis showed that the Watkins-unique trait associated alleles or haplotypes are beneficial with increasing effects based on our understanding of physiological trait relationships (d.ii and d.iii). These prioritisation selection criteria are used to choose alleles to build new modern wheat cultivars while avoiding negative trade-offs.

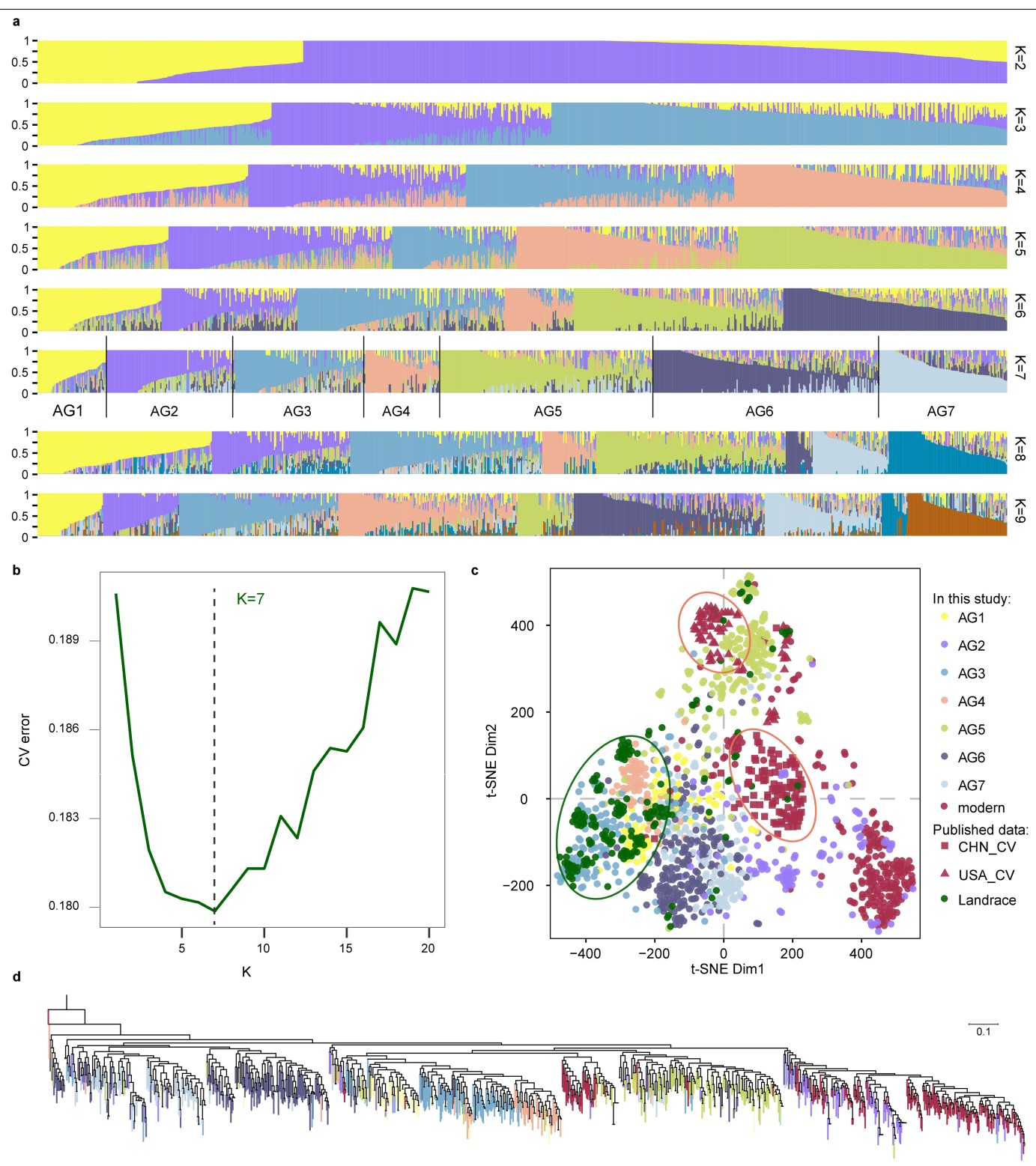

**Extended Data Fig. 2** | See next page for caption.

**Extended Data Fig. 2 | Population structure and phylogenetic analysis of Watkins landraces and modern wheat cultivars. a**, Genome-wide ADMIXTURE (Maximum Likelihood Estimation) results for 827 Watkins landraces from K = 2 to K = 9. Each colour represents an ancestral population. The length of each segment in each vertical bar represents the proportion contributed by ancestral populations. **b**, Estimated CV error for different K values (from K = 2 to K = 20) in the ADMIXTURE analysis with a K of 7 designating the ancestral groups (AGs) identified here. **c**, dim1 and dim2 plots of t-SNE results using PLINK haplotypes, for the merged variation matrix (4 M shared SNPs) between the SNP matrix built in this study (the 10 M core SNPs, see Methods) and the published dataset (76 M SNPs) from Niu et al.[13]. **d**, Phylogenetic reconstruction of a set of 133,222 4DTv sites using Maximum Likelihood Estimation with 1000 bootstrap replicates, corresponding to Fig. 2a with the ancestral groups (AG1-7) color-coded. It is worth noting that a subjective observation of the phylogenetic analysis of Watkins based on the maximum likelihood method (Extended Data Fig. 2d), largely places individual accessions of each ancestral group beneath common branchpoints of the phylogenetic tree. Instances where this is not observed reveal extensive admixture (Extended Data Fig. 2a, Supplementary Table 4). The highest level of admixture for AG2 is from AG5 (approximately 11.2%), for AG5 it is from AG7 (approximately 7.5%). This reflects the reticulate relationship among these ancestral groups and further demonstrates the need for alternative methods, particularly sophisticated models reliant on haplotype-based clustering for accurate inference of ancient wheat populations[10].

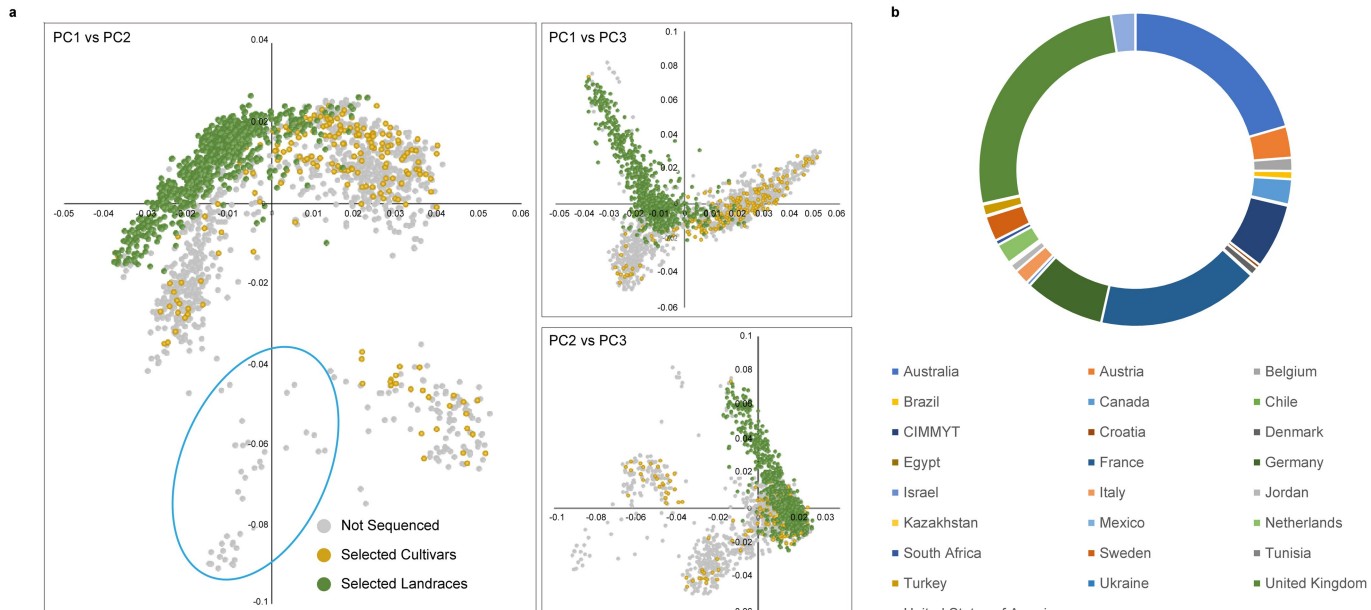

**Extended Data Fig. 3 | Choice of modern wheat cultivars and comparison with the Watkins collection. a**, Principal component analysis based on 35k Wheat Breeders' Array[6] genotypic data for 1169 modern cultivars and Watkins landraces hosted by CerealsDB (https://www.cerealsdb.uk.net/cerealgenomics/CerealsDB/indexNEW.php) when selections for sequencing in this project were made. Selected cultivars are shown in gold, Watkins in green and cultivars not selected for sequencing in grey. Representative cultivars from the blue circled group in PC1 vs. PC2 were not selected because synthetic wheat derivatives are highly represented in this group. **b**, Countries of origin of the 1169 modern cultivars genotyped.

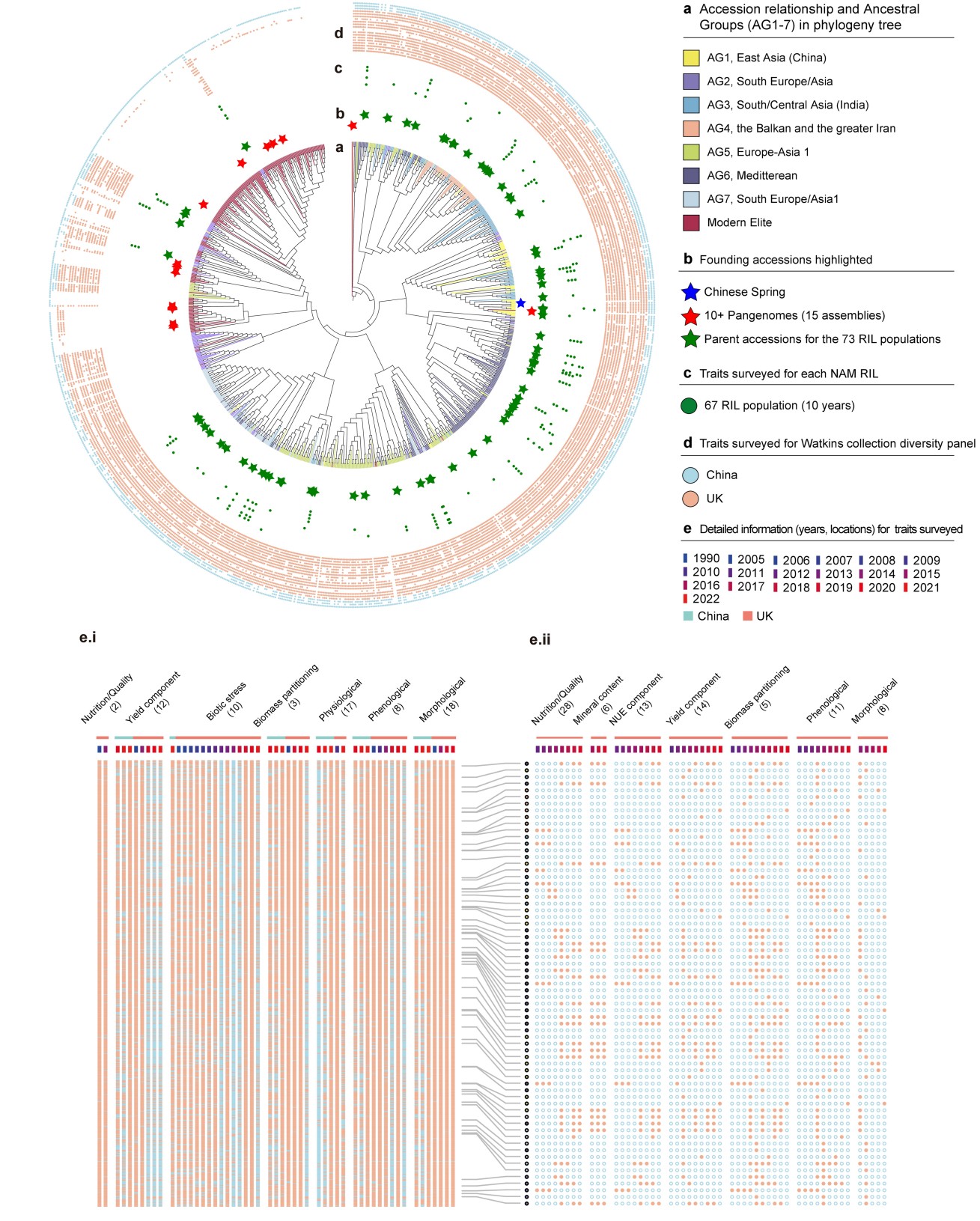

**Extended Data Fig. 4** | See next page for caption.

**Extended Data Fig. 4 | Watkins collection and mapping populations, phenotypic resources and traits surveyed summarised in a phylogeny-based circular diagram. a**, Phylogenetic tree of the wheat accessions examined in this study. The phylogenetic tree was constructed using a set of 133,222 four-fold degenerate sites using rapidNJ with 1000 bootstrap replicates. The seven ancestral groups (AG1–7) and modern wheats are colour-coded as in Fig. 1a. **b**, The founder parents (green stars) of 73 Watkins x Paragon RIL populations are marked, the 15 pan-genome lines (red stars) and Chinese Spring (blue star) are indicated on the phylogenetic tree. **c**, Traits surveyed in multiple environments and multiple years for each of the NAM RIL populations, in which the corresponding Watkins line was used as the non-common parent.

Each track represents a year (total of 10 years: 2011, 2012, 2013, 2014, 2015, 2016, 2017, 2018, 2019 and 2020). **d**, Traits surveyed in multiple environments and multiple years for the Watkins collection diversity panel (natural populations) grown in five geographic locations across China (blue circle, four years: 2020, 2021, 2022, 2023) and the UK (orange circle, 16 years: 1990, 2005, 2006, 2007, 2008, 2009, 2010, 2011, 2012, 2013, 2014, 2015, 2018, 2019, 2020 and 2021). **e**, Magnified view of data from the detailed field experiments, traits and phenotyping datasets as indicated in track d, including the traits surveyed in the Watkins collection diversity panel in multiple environments (e.i) and in multiple years (left) and in the RIL populations (NAM RILs) in multiple environments and in multiple years (e.ii).

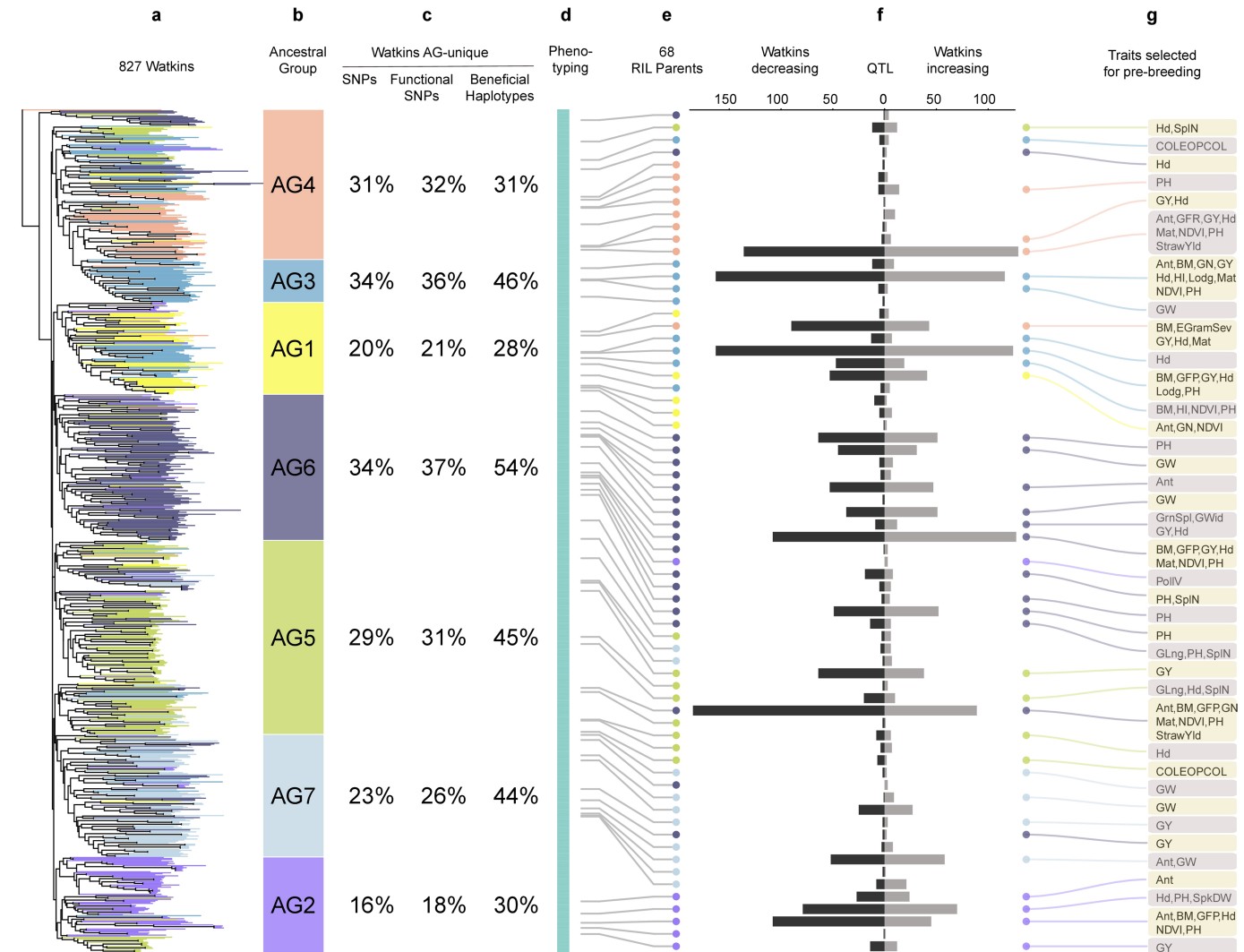

**Extended Data Fig. 5 | Information flow from novel and functional genetic diversity derived from Watkins landraces to the quantification of the beneficial increasing QTL allele associated with target traits. a**, Phylogenetic tree of the 827 Watkins accessions, colour-coded by ancestral groups; the branches were roughly classified into the seven ancestral groups in panel **b**. **c**, Percentage of AG-unique genetic diversity for SNPs, functional SNPs (defined by SnpEff (v4.3t)[20]) and beneficial haplotypes. **d**, Phenotyping of the Watkins collection including data collected in China and the UK and data for the

68 of 73 total RIL populations which exhibit significant QTLs in panel **e**. The Watkins parental lines in the RIL population, with colours representing their ancestral groups. **f**, distribution of the number statistic of QTLs detected from the biparental QTL mapping populations, comparison was made for the beneficial QTL with increasing effects (right) and decreasing (left) in Watkins. **g**, Prioritised QTL and the major traits selected for introgression into Paragon via backcrossing to test their phenotypic effects for pre-breeding.

Simon Griffths

# Reporting Summary

## Statistics

For all statistical analyses, confirm that the following items are present in the figure legend, table legend, main text, or Methods section.

| n/a | Confirmed | |
|---|---|---|
| ☐ | ☒ | The exact sample size (*n*) for each experimental group/condition, given as a discrete number and unit of measurement |
| ☐ | ☒ | A statement on whether measurements were taken from distinct samples or whether the same sample was measured repeatedly |
| ☐ | ☒ | The statistical test(s) used AND whether they are one- or two-sided *Only common tests should be described solely by name; describe more complex techniques in the Methods section.* |
| ☒ | ☐ | A description of all covariates tested |
| ☐ | ☒ | A description of any assumptions or corrections, such as tests of normality and adjustment for multiple comparisons |
| ☐ | ☒ | A full description of the statistical parameters including central tendency (e.g. means) or other basic estimates (e.g. regression coefficient) AND variation (e.g. standard deviation) or associated estimates of uncertainty (e.g. confidence intervals) |
| ☐ | ☒ | For null hypothesis testing, the test statistic (e.g. $F$, $t$, $r$) with confidence intervals, effect sizes, degrees of freedom and $P$ value noted *Give P values as exact values whenever suitable.* |
| ☒ | ☐ | For Bayesian analysis, information on the choice of priors and Markov chain Monte Carlo settings |
| ☒ | ☐ | For hierarchical and complex designs, identification of the appropriate level for tests and full reporting of outcomes |
| ☐ | ☒ | Estimates of effect sizes (e.g. Cohen's *d*, Pearson's *r*), indicating how they were calculated |

*Our web collection on statistics for biologists contains articles on many of the points above.*

## Software and code

Policy information about availability of computer code

| Data collection | No software used for data collection. |
|---|---|
| Data analysis | Alignment and SNP calling: |

Alignment and SNP calling:
Alignment: BWA-MEM (v0.7.17)
Mark duplicate: picard (v 2.20.3-SNAPSHOT)
Sort Alignment file: SAMtools (v1.9)
SNP/Indel Calling: GATK (v4.1.2)
SNP/Indel filtering: (https://github.com/ShifengCHENG-Laboratory/WWWG2B/tree/main/00.SNP_calling_and_QC )
LD pruning: PLINK (v1.90b6.7)

Population structure analysis:
Phylogenetic tree: iqtree (v1.6.9); rapidnj(Version 2.3.2)
Tree visualisation: iTOL (https://itol.embl.de/)
Structure: ADMIXTURE (v1.3.0)
Genetic diversity (π): vcftools (v0.1.13)
Population differentiation (FST): vcftools (v0.1.13)
identity-by-state matrix: PLINK (v1.90b6.7)
PCA: Python-sklearn (v0.24.2)
t-SNE: Python-sklearn (v0.24.2)

Wheat HapMaP construction:

Block identification: PLINK (v1.90b6.7)
Block connection: (https://github.com/ShifengCHENG-Laboratory/WWWG2B/tree/main/01.HAPMAP_pipeline )
Haplotype assignment: HAPPE (v0.1.4), https://github.com/fengcong3/HAPPE

GWAS study:
Calculate kinship and perform GWAS: GEMMA (v0.98.1)
Manhattan plot and QQ plot: (https://github.com/ShifengCHENG-Laboratory/WWWG2B/tree/main/04.GWAS_pipeline )
Local details and LD: LDBlockShow (v1.40)

NAM imputation:
Imputation: (https://github.com/ShifengCHENG-Laboratory/WWWG2B/tree/main/03.NAM_imputation_pipeline)
Merge VCF: bcftools (v1.9)

CNV identification:
Blastn (v 2.8.1+)
In-house pipeline(https://github.com/ShifengCHENG-Laboratory/WWWG2B/tree/main/07.CNV_pipeline )

k-mer databases count from raw reads and genome assemblies:
kmerGWAS pipeline https://github.com/voichek/kmersGWAS
Jellyfish v.2.2.6
KMC v3.0.1

IBSpy variations and Long Range haplotypes:
The implementation of the complete pipeline is in the final version of IBSpy (v.0.4.6) in https://github.com/Uauy-Lab/IBSpy
Affinity Propagation (AP) methods and the API: https://scikit-learn.org/stable/modules/generated/sklearn.cluster.AffinityPropagation.html
AP clusters were scored by Silhouette Coefficient (SC) score and the API: https://scikit-learn.org/stable/modules/generated/
sklearn.metrics.silhouette_score.htm
Syntenic windows among wheat genome references were developed in (Brinton et al. Communications Biology, 3, 712)
Minimum landrace path: https://github.com/Uauy-Lab/MLP_finding.

Sequencing data quality control:
Comparison of 35k Axiom array with sequence data: https://github.com/JIC-CSB/WatSeqAnalysis/tree/master/qc_vs_iselect

Statistical analyses
R software suite (version 4.2)
Pearson correlation coefficient (r) was calculated using R package corr v0.4.4  and corrplot v0.92.
ANOVA has been calculated as a linear model with the genotype as a factor, using base function lm.
AMMI was calculated using base functions aov and svd following the code from https://journal.r-project.org/articles/RN-2007-003/
RN-2007-003.pdf

Genetic map construction:
Mapdisto (version 1.7) for KASP genotypes and R package ASMap (version 1.6) for 35k Wheat Breeders' array genotyping.

QTL analysis
R software suite (v3.6.1) package qtl (v1.5)

Triticum aestivum Next Generation array
Scripts for the design of the TaNG genotyping array: https://github.com/pr0kary0te/GenomeWideSNP-development

Standardization of phenotypic data:
Crop Ontology Curation tool (https://cropontology.org/)

Geographic distance:
R package geosphere (v1.5-18)

For manuscripts utilizing custom algorithms or software that are central to the research but not yet described in published literature, software must be made available to editors and reviewers. We strongly encourage code deposition in a community repository (e.g. GitHub). See the Nature Portfolio guidelines for submitting code & software for further information.

# Data

Policy information about availability of data

All manuscripts must include a data availability statement. This statement should provide the following information, where applicable:

- Accession codes, unique identifiers, or web links for publicly available datasets
- A description of any restrictions on data availability
- For clinical datasets or third party data, please ensure that the statement adheres to our policy

All whole-genome sequence data has been deposited at the National Genomics Data Center (NGDC) Genome Sequence Archive (GSA) (https://ngdc.cncb.ac.cn/gsa/search?searchTerm=CRA012590), with BioProject accession number PRJCA019636, and with GSA Accession ID: CRA012590. Variation matrix and annotations, wheat HapMap, phenotyping data, genetic maps with genotype scores, association genetics analyses, the developed tagSNPs and KASP markers were deposited in WWWG2B breeding portal (https://wwwg2b.com). IBSpy variations tables, haplotypes, long-range tilling paths, variant files (VCF) and all raw phenotypic data are available online (https://wwwg2b.com/dataAvailable, https://wwwg2b.com/toolIndex/academic and https://opendata.earlham.ac.uk/wheat/under_license/toronto/WatSeq_2023-09-15_landrace_modern_Variation_Data/). Publicly available sequencing data were obtained from SRA accessions SRP114784, PRJNA544491, PRJEB37938, PRJNA492239, PRJNA528431, PRJEB39558, PRJEB35709 and from the NGDC database project CRA005878.

The IWGSC RefSeq is download from https://urgi.versailles.inra.fr/download/iwgsc/IWGSC_RefSeq_Assemblies/v1.0/ .

Raw phenotypic data is available at https://wwwg2b.com/dataAvailable/Watseq_phenotype_data/, https://grassroots.tools/fieldtrial/ and https://opendata.earlham.ac.uk/wheat/under_license/toronto/WatSeq_2023-09-15_landrace_modern_Variation_Data/WatSeq_phenotypic_data/ (field and glasshouse data). Yellow rust resistance scores for Watkins natural populations in Ethiopia and Kenya field trials are available here: https://doi.org/10.5281/zenodo.8349020.

35k Wheat Breeder's array genotype data of RILs and NILs is available at: https://www.cerealsdb.uk.net/cerealgenomics/CerealsDB/array_info.php

Additional datasets are provided in the Supplementary Tables.

# Research involving human participants, their data, or biological material

Policy information about studies with human participants or human data. See also policy information about sex, gender (identity/presentation), and sexual orientation and race, ethnicity and racism.

| | |
|---|---|
| Reporting on sex and gender | not applicable |
| Reporting on race, ethnicity, or other socially relevant groupings | not applicable |
| Population characteristics | not applicable |
| Recruitment | not applicable |
| Ethics oversight | not applicable |

Note that full information on the approval of the study protocol must also be provided in the manuscript.

# Field-specific reporting

Please select the one below that is the best fit for your research. If you are not sure, read the appropriate sections before making your selection.

☒ Life sciences    ☐ Behavioural & social sciences    ☐ Ecological, evolutionary & environmental sciences

For a reference copy of the document with all sections, see nature.com/documents/nr-reporting-summary-flat.pdf

# Life sciences study design

All studies must disclose on these points even when the disclosure is negative.

| | |
|---|---|
| Sample size | The Watkins collection consists of 827 individuals. Unreplicated field trial data was used for GWAS. The diversity subset of the Watkins collection, grown in Egypt, contained 300 Watkins lines and was used for GWAS. The RIL population size was near 94 RILs for most of the 73 populations and was used for QTL mapping. In a very few extreme cases, QTL mapping was conducted with as low as 88 RILs. We assume that a small population size will lead to a higher false positive rate in QTL mapping. We counteract this problem in our research strategy by either a repetition of the QTL experiment in another season or by checking QTL effects in NIL validation trials. Furthermore, we use several populations grown at the same site and the same season, for NAM-GWAS analysis. Sample size in the NAM GWAS was between several hundred to several thousand RILs. |
| Data exclusions | The genotyping of three RIL populations (ParW444, ParW264, ParW313) out of 112 populations, revealed unexpected genotypes, which could only be explained by a mistake in the crossing programme. The populations and collected data were discarded, resulting in 109 RIL populations in the study. |

NIL families Fam-013 to Fam-016 and Fam-023 lost all members with a second allele due to a drilling error in the multiplication trial. These families were removed from the data analysis.

Data from three NIL lines of family Fam-008 was excluded from the analysis because they showed a striking dwarf growth habit in the field that was very different to the rest of the germplasm.

Data from two lines WL0019 and WL0026 from the SB_H18 TK trial was excluded from the analysis of the Q7B-PH grain yield effect, as the plant height effect for all three replicates suggested that the samples had been swapped and it was not possible to go back to the original records.

**Replication**

Field trials with the 827 Watkins accessions were grown in a randomised, unreplicated augmented block design with 10% 'check' varieties. Similarly, glasshouse trials of the 827 Watkins acessions, to confirm specific effects like yellow rust resistance or glume colour, were conducted in randomised but unreplicated trials, due to the size of the collection. Two Watkins trials in China were grown in a randomised split-plot design with two Nitrogen fertiliser treatments and two replicates each. Initial seed multiplication exercises of the 127 RIL populations, each with 88-102 individuals, were unreplicated and were grown in augmented block design with 20% check varieties. TK trials at JIC, consisting of NIL families of two to 8 NILs with two contrasting alleles, were grown in augmented block design with 10% check varieties in the multiplication trial. After multiplication, TK trials were grown in three replicates and in randomised block design at JIC, RH and SB. All RIL trials at RH and SB, were grown in randomised block design with three replicates. Specific NIL trials, comparing an individual NIL pair, were grown in randomised block design with three replicates, or even five replicates for the RHT8 trials in Spain and Serbia. Yellow rust trials of RIL populations (90-94 RILs each) at commercial breeding stations were grown in randomised unreplicated trials, with two time points of disease scoring.

**Randomization**

All field trial design included randomisation of the genotypes.

**Blinding**

Phenotypic measurements on the Watkins, RILs, and NILs were taken independently by different scientists, across locations, and without knowledge of the underlying identifier of each accession. Hence all phenotypic measurements were blinded.

# Reporting for specific materials, systems and methods

We require information from authors about some types of materials, experimental systems and methods used in many studies. Here, indicate whether each material, system or method listed is relevant to your study. If you are not sure if a list item applies to your research, read the appropriate section before selecting a response.

## Materials & experimental systems

| n/a | Involved in the study |
|---|---|
| ☒ | Antibodies |
| ☒ | Eukaryotic cell lines |
| ☒ | Palaeontology and archaeology |
| ☒ | Animals and other organisms |
| ☒ | Clinical data |
| ☒ | Dual use research of concern |
| ☐ | ☒ Plants |

## Methods

| n/a | Involved in the study |
|---|---|
| ☐ | ☒ ChIP-seq |
| ☒ | Flow cytometry |
| ☒ | MRI-based neuroimaging |

## Plants

**Seed stocks**

All the germplasm used in this study is conserved in the UKRI-BBSRC Germplasm Resources Unit (GRU) National Capability at JIC. The seed and passport data is available on https://www.seedstor.ac.uk/.

The progenitor landrace populations (Watkins Historic Collection) are available on https://www.seedstor.ac.uk/search-browseaccessions.php?idCollection=4

The derived 827 sequenced Watkins landraces seed stocks are available on https://www.seedstor.ac.uk/search-browseaccessions.php?idCollection=39

The 208 modern elite wheats used were sourced from various SeedStor collections and can be viewed and ordered collectively as a Compiled Accession List at https://www.seedstor.ac.uk/search-custom.php

Pure stocks of the seed harvested from a single DNA sequenced plant (gold standard stocks) are kept for reference in the GRU and were shared with AGIS. Progeny of these were multiplied in greenhouse with bagged ears for preventing cross pollination and handled following international Genebank Standards (Rome 2014)

**Novel plant genotypes**

The BBSRC Designing Future Wheat - Recombinant Inbred Lines (RILs) Nested Association Mapping panel (DFW - NAM) comprising 8,359 greenhouse grown cellophane bagged hand threshed seed stocks, deposited in the GRU as part of this work. It includes the

reported 73 RIL populations. Seed stocks are available on  https://www.seedstor.ac.uk/search-browseaccessions.php?idCollection=47

The DFW Wheat Academic Toolkit pre breeding germplasm collection of 1,845 lines that were deposited in the GRU as part of this work include all the Near Isogenic Lines (NILs), reported in this study. SeedStock are available on  https://www.seedstor.ac.uk/search-browseaccessions.php?idCollection=40

**Authentication**

High heritability traits (height, heading date, glume colour, seed appearance, presence of awns, length of awns) are confirmed as a standard procedure for all glasshouses grown stocks that were available on SeedStor collections for this study. Seed and ear morphology are compared by the genebank curator to the previously grown seed lot and sampled ear. Plant morphology traits are compared to the previous regeneration data records, dating back to the early 1970s for modern varieties and to the original century old grouping in case of Watkins landrace collection. The purified Watkins landrace accessions and the modern wheat panel were genotyped with Axiom 35K high density genotyping and SNP calls checked against the sequence generated in this study (see Methods). NILs were also subjected to Axiom 35K genotyping. RILs are genotypes with Axiom 35K Breeder array or KASP markers.

To ensure that field plots are composed of the intended lines, they were scored for morphological and high heritability traits such as height, heading, glume colour, and the presence of awns. Awns were particularly useful as Paragon (common NAM and Recurrent backcross parent) carried the B1 awn inhibiting allele on chromosome 5A whereas the majority of Watkins accessions are awned. Trait values were correlated between experiments and QTL mapped (e.g., Awns to B1 on chromosome 5A, height to major QTL on chromosome 6A, glume colour to chromosome 1B). In the event of any doubt, small sets of diagnostic KASP markers were used to genotype ~8 seeds from each field plot. We estimate 2-5% cross contamination of field plots harvested by plot combine harvesters (This was tested and confirmed in 2021 in five breeder and academic institute sites evaluating the DFW stocks by counting off-types in 6-metre plots of F5 NILs). Seed was not multiplied from field plots for more than two generations; new "pure" stocks were taken from glasshouse regenerated plants (bags placed over spike before anthesis to prevent cross pollination, see Methods).

# ChIP-seq

## Data deposition

☐ Confirm that both raw and final processed data have been deposited in a public database such as GEO.

☐ Confirm that you have deposited or provided access to graph files (e.g. BED files) for the called peaks.

**Data access links**
*May remain private before publication.*
*For "Initial submission" or "Revised version" documents, provide reviewer access links.  For your "Final submission" document, provide a link to the deposited data.*

**Files in database submission**
*Provide a list of all files available in the database submission.*

**Genome browser session**
(e.g. UCSC)
*Provide a link to an anonymized genome browser session for "Initial submission" and "Revised version" documents only, to enable peer review.  Write "no longer applicable" for "Final submission" documents.*

## Methodology

**Replicates**
*Describe the experimental replicates, specifying number, type and replicate agreement.*

**Sequencing depth**
*Describe the sequencing depth for each experiment, providing the total number of reads, uniquely mapped reads, length of reads and whether they were paired- or single-end.*

**Antibodies**
*Describe the antibodies used for the ChIP-seq experiments; as applicable, provide supplier name, catalog number, clone name, and lot number.*

**Peak calling parameters**
*Specify the command line program and parameters used for read mapping and peak calling, including the ChIP, control and index files used.*

**Data quality**
*Describe the methods used to ensure data quality in full detail, including how many peaks are at FDR 5% and above 5-fold enrichment.*

**Software**
*Describe the software used to collect and analyze the ChIP-seq data. For custom code that has been deposited into a community repository, provide accession details.*

