## [Peer Review File · Nature]

Manuscript Title: Harnessing Landrace Diversity Empowers Wheat Breeding

Reviewer Comments & Author Rebuttals

Reviewer Reports on the Initial Version:

Referees' comments:

Referee #1 (Remarks to the Author):

Harnessing Landrace Diversity Empowers Wheat Breeding for Climate Change

General Comments

Firstly, I commend the authors for their significant work on this manuscript - this was not small task. The collective work is substantial, delivering a broad range of practical tools to support wheat improvement. I generally support the publication of the article, given its impact on the large wheat and breeding community. However, in its current state, I feel the manuscript lacks focus, and, as a consequence, details and relevance are lost to the reader. Indeed, presenting a range of topics, including variation in diversity, nitrogen use efficiency, height/flowering nutritional quality, inter-trait correlations, cloning of several useful genes (disease and otherwise), development of a new genotyping system, strategies for introgressing diversity in breeding, and QTL analysis (all useful and practical), seems overly ambitious. I recommend that the authors reevaluate their presentation style and prioritize three to four main areas of novelty, while removing parts (that could be published elsewhere and not critical to this manuscript) and/or relegating more to supplementary information.

From my perspective, I found the sections on the characterization of diversity of haplotypes (and associated useful software packages developed for their utility), strategies for introgression (to capitalize on diversity) very informative and useful. Perhaps bolstering text around these and one/two examples of its utility for climate-resilient would allow more opportunity to expand on each in the main text to stress the value and would appeal to a broader audience. For example, the marker trait association studies (GWAS, QTL, NAM) are substantial, but stressing novel findings (blast resistance for example) with those haplotypes not available to modern breeding would reveal the impact/utility for improving climate resilience. The nutritional angle of increasing calcium content does have support from functional validation of a candidate gene and seems to this reviewer to be novel - although this trait is not tied to climate resiliency which seems to be the target focus of the manuscript.

The haplotype variation in and around Cbf genes (cold tolerance) and Ppd genes (flowering time), while interesting, is not particularly novel (it is well known for example that variation in CNV of Cbf genes is associated with cold tolerance) and could be removed.

I found the section on developing a new 40K SNP high-throughput genotyping array lacking in the main text. I am not suggesting this work is not important, but perhaps this could be better integrated within the main text/supplemental materials or considered in another manuscript that can adequately deal with its description and comparative utility. Were any of the haplotype-based

diagnostic KASP markers developed to detect the haplotype blocks included in the design?

Lastly, I am pleased the authors acknowledged the limitations of short-read whole-genome sequencing and mapping to a reference genome that itself was assembled using predominantly short reads. Indeed, several long-read generated-based reference sequences have recently become available for wheat, which have been shown to be better ordered and more continuous. I was hopeful these too would have been included in various analyses given this advantage. I think it is worthwhile in the discussion to stress the current limitations (especially CNV-related analysis) and that more can be captured. This does not negate the positive results presented or the significant genomic resources presented in this paper. I was glad to see the 10+ Genome Project assemblies included in the K-mer based haplotype analysis, which provides a link to additional pan-genome assemblies.

Specific Comments:

The abstract suggests that variation in modern cultivars can trace to two ancestral landrace groups. This is supported by the data for those modern cultivars assessed in the study (vs. all modern cultivars, like those with introgressions from synthetic wheat or wild relatives).

Paragraph three of the main text discusses several obstacles in the introgression of novel alleles into wheat breeding, and that the work presented "overcomes these obstacles." Indeed, the results presented provide tools and resources to efficiently use the diversity that has been characterized, but many of the obstacles remain a challenge (e.g., dominant gain-of-function mutations, complex genetic architecture). What has been overcome is the availability of large, useful datasets and well-characterized resources that allow targeted introgression of diversity. This, in my opinion, represents the true value and overcomes the obstacle of "random" in breeding. There is a new strategy to target diversity. Perhaps this should be the focus of the "obstacles."

It took a fair bit of cross-referencing various methods and extended data/supplemental tables to really understand the "maximal diversity in the 1169 cultivars from 25 countries." How does this relate to the 3500 mentioned in extended Figure 4? I may simply have missed it.

The paragraph starting at line 171 references functional traits, functional variations, functional SNPs, functional significance. It is not clear to me what is meant by functional. Please reword to be clear. Also, there is mention that there is no "Standing variation [in those regions IBS in all modern cultivars] to improve through traditional breeding." Of course, these regions can be improved through traditional breeding (intercrossing with landraces is one strategy of breeding). Perhaps rewording this to be more explicit that a wide-crossing strategy, coupled with molecular selection (as suggested), would be effective.

Line 205 mentions useful QTL alleles that increase phenotypic value when selected in the same direction. It is not clear to this reviewer what is meant by selection in the same direction. Please clarify.

Line 221 of the main text, 1,696 genetic effects were identified as high potential for wheat improvement "based on the direction of allelic effects." Please clarify – do you mean positive effect, that is, those QTL that give substantial improvement based on QTL effect size? Line 261 – beneficial traits? Do you mean beneficial phenotypes?

I was particularly excited to see the results showing Pm4 is the casual gene underlying blast resistance in wheat. There is little variation available for wheat improvement to this devastating disease. The authors refer to a companion paper, with some results presented in a composite figure (Figure 4). The online methods section, however, did not provide this reviewer with sufficient information to really appreciate these results and to assess the robustness of the validation experiments. While I am not privy to the companion paper, it would seem this is a novel aspect of this manuscript that could be bolstered.

The paragraph starting at L261 – This section notes the positive effect of Rht1 on reducing plant height (which seems a positive phenotype) but at the expense of plant biomass. The authors then identify a height-increasing QTL on 7BL that together increased plant height (by 10 cm) and grain yield (0.39 t ha⁻¹). This section seemed awkward to this reviewer -- if the intent is to capture reduced height but higher yield (and thus biomass), why not present a QTL where short stature is maintained, but also has a positive effect on yield (and/or associated yield components). This would be quite valuable for breeding and climate resiliency (maintaining short stature but elevated yield). In the next section of the manuscript, Rht8 mapping and gene discovery was facilitated by the Watkins collection. Indeed, two candidate genes were identified where their expression is associated with phenotype. No validation of these candidates was performed to confirm their functional involvement in variable trait expression. Perhaps the authors can combine the Rht8 with the height QTL dissection characterized on 7BL into one section related to height in a more thoughtful way.

Methods:

The statistical analysis methods presented were very generalized. Given the large amount of phenotypic data conducted in various replicated experiments at multiple environments and over time, I would like to see more information on how the ANOVA was conducted (mixed models, BLUPS?). This would be important to appreciate the robustness of the analysis (especially since such large datasets can be unbalanced) and the final set of means that were used in marker trait association studies.

Referee #2 (Remarks to the Author):

The greatest strength of this manuscript is the incredible amount of data that was collected and used to produce it. The greatest weakness of this manuscript is that there is simply too much “stuff” for the broad readership of Nature to walk away with a take home message(s) (I suggest a couple of the big ones below and later in my summary). I struggled to follow what the main takeaways were supposed to be and there was so much supplemental information at some point I had to give up on evaluating these figures and tables where they were called out in the manuscript. I believe the overarching theme of this paper is that there is a large amount of genetic diversity in untapped germplasm that breeders can use for wheat improvement (demonstrated) and especially improvement for climate change (I don’t think well demonstrated). This is mostly a pretty conventional theme for crop CNS publications, but this article has an unprecedented scale with a tremendous amount of data collection supporting it. I don’t mean this theme is conventional to

denigrate the article, but to imply that it fits the journal and readerships expectations well in subject matter. There are some especially novel components in this article. I appreciated all of the phenotypes taken that led to a discussion and case study on tradeoffs in height (line 261 to 275). The NILs / HIFs developed and tested are an especially strong component and resource, given wheat does not cross easily, this also represents a tremendous effort (although I argue later that a BC2F2 is a HIF not a NIL, so some tempering of wording is needed). A major problem this article addresses is THE major limitation to breeders and crop scientists in using germplasm collections, the lack of relevant field phenotypic characterization; the major strength of this article is the unprecedented scale of characterization that was undertaken; if anything the authors could better emphasize this point for the broad readership of Nature by comparing it to the “next largest” study in terms of accessions or RILs or genotyping etc. Overall, the authors should, in my opinion, better emphasize their impressive field phenotypic work and findings and remove much of the excess genomics work that did not result in broadly interesting discoveries – that would greatly simplify and shorten the article.

More philosophically, the potential impact of this work for wheat improvement rests on two hypotheses that I take issue with as presented and deserve some thought by the authors: 1) the implicit assumption that the loci discovered are purely additive without epistasis, genetic background effects, or substantial GxE interactions, ready to be used by breeders; 2) that breeders will trust and use these loci in their own breeding efforts. In my experience, wheat breeders are among the most conservative in their belief of crossing elite by elite and not “wanting to break up those good linkage blocks” as a few breeders have told me. In other words, a century of wheat breeding purged deleterious alleles and pyramided elite ones, why go back and bring in the “junk”? I don’t think commercial wheat breeders will want to use these resources, except for a few economical simply inherited traits (yield, protein). Overall, it is also my belief that the reason exotic germplasm is rarely used in elite “practical wheat improvement through breeding” (line 250) is not simply linkage drag (linked deleterious alleles brought into the genetic background) but also because many times alleles identified in genetic studies fail to work the same way in elite genetic backgrounds (epistasis and genetic backgrounds) which is not addressed. Though this is a hypothesis, it is one of the few explanations and is supported by the authors own findings in NILS (“range of penetrance” line 257 and possibly lines 370 to 384). So, if the authors goal of the paper is to convince wheat breeders to use this pre-breeding material, the focus should be on the economical traits (protein, yield, disease, NUE) and demonstrate no linkage drag and consistent allele effects across genetic backgrounds. If the goal is to present other aspects to the general readership of Nature and policy makers, the focus should remain on much of the current presentation, but the presentation needs to be simplified and honest that this is defensive research and clear that current commercial breeders will be reluctant to use these resources for the reasons above.

One of the line by line comments I am moving up here for broader discussion, is some of the individual “NIL” phenotypes (again, not really NILs, more like HIFs; lines 369 to 374), a very exciting and important component of the paper, but which remains lost. However, the specific claimed yield gains of the NILS “4.5 t ha⁻¹” are especially incredible, to the point that I struggle to believe them, especially from a single NIL/HIF. Given the extensive field testing reported I don’t have concerns on if it was tested in the field (although I could not find specific evidence of where this number came

from in the mountain of supplemental information). If this NIL/HIF is real I think that would deserve to be its own article, as more detailed examination of the materials and methods is needed to understand how these results were obtained. I would especially like to see if there was a decrease in protein. There are often pleiotropic tradeoffs between yield and protein, and although a 3.2% increase in protein was also noted, I don't believe these were in the same NIL/HIF or same genes as the yield increase? Not clear. More detail on the NIL/HIF evaluation, a confirmation of QTL/ GWAS hits discovered, would improve this paper greatly. A lot of the genomics could be deleted to make space, as there is less novelty or interesting discovery in these sections.

In general, all the main figures are nice, useful and important, though simplification would help. The other figures are also nice, and represent a lot of work, but many are not needed for broad readership. Extended Data Figure 1 and 2 summarize the incredible amount of work done and also summarize my major issue with the paper: there is so much "stuff" that this group did, that I and any reader will struggle to follow it; the legends for these figures are arguably longer than some manuscripts! On the plus side, whatever methods a reader could want to find (except phenomics – a drone and spectra are shown but no evidence of this – too bad) is in this paper as something they did at very large scale, but with the combined massive volume of work it is hard to find where it was discussed, and with what resource was tested, and critically what important was found from all of this work. At the same time, while Extended Data Figure 1 and 2 focuses on what the authors did and how they did it, I am not sure what the main message is beyond that they were able to do it. Extended figure 7 also is helpful to show scale of phenotyping. Among the extended data figures, Extended figure 4 is one of a few that shows discoveries I think the broader readership of Nature would be interested in, demonstrating Watkins diversity compared with other modern wheat cultivars. The others are either methods or intermediate findings of the paper. Extended data figure 8 also shows some QTL results, but I am not sure what the reader is supposed to take away. Of course, we expect a massive number of QTL from such a large study, so I hope that is not the point the authors are trying to make. Yet, any wheat researcher interested in a specific genomic location or trait will need to go to the tables to learn exact effects and genomic positions. I think the only useful takeaway for readers of studies showing most or all QTL in one figure like this is to present pleiotropic tradeoffs of major loci, but that was not highlighted in a visible way here. Extended data figure 9 is also interesting, but I think part 'g' needs to be presented not in the same line rows, as it looks like 'Yield components' were only measured in AG4 derived RILS? Actually, I am not even sure which parts do connect across the figure parts. I most like part 'f', this is great to show the positive and negative QTL effects from Watkins landraces and could be a main figure and a main point (demonstrating the benefit of diversity + linkage drag), however does part 'f' connect with 'e'? with 'b'?, with 'g'? Maybe rotating some parts 90 degrees to columns could help? Extended data figure 10 may be the best for the readership of Nature, it shows a trait of broad interest, its genetic dissection and how it is unique distributed in the landraces – this alone could be a singular focus of the article. However, I scarcely remember the discussion of this nitrogen trait in the article – it would need to be made more memorable. Also I am unclear if there were NILS or other specific resources to validate it? If it was simply an identified QTL that was not validated by NILS in a separate relevant experiment with appropriate field data, this would not be reliable and not a major interest of the paper.

I feel fatigued after reading this massive, and very impressive, tour de force manuscript (as the authors probably do by this review). I do not envy the writers putting this all together. I apologize for

the delay in my review but in my hundreds of reviews and papers I have edited I have never seen anything like this, it seems fair to say this is about 10 manuscripts in one, and quickly I found myself taking shortcuts from my normal reviewing process of skipping supplemental work as it was called out. There was just too much supplemental information and too many complicated figures to easily follow. I fear readers will feel the same way, that is why some revision of presentation is need to focus on what was discovered and message around that and not on the massive amounts of work conducted.

In summary, there is something interesting and very important here for the readership of Nature. I think it is 1) putting into perspective the unprecedented scale needed for characterization of genetic resources, and a few case study discoveries that this study allowed that past studies have not, AND discussing what this characterization requires (\$, hours, etc.) and why it has not been routinely done in the past. If people want to use genetic dissection for to improve wheat and wheat resilience, make them understand how underinvested this aim is and that studies like this are needed (if the authors think they are, I do). This must be followed up with 2) the validation through NILS/HIFS and field trials, without this effort I would suggest publishing in a specialty journal as another massive unconfirmed QTL / GWAS study (although with the amount of data, I wonder if cross validation could also be used for validation in this case?). I also think the main point that 3) there is much diversity that is missing from modern wheat, existing in landraces and germplasm banks and this diversity won't get used unless it is phenotypically characterized, and a few case studies of that. For the broad readership of Nature, everything else seems like showing data for the sake of showing data and is just too much for a reader to digest. Targeting 10 supplemental tables and 10 supplemental figures, and 5 extended data figures seems a good goal compromise. Everything else could be split into different manuscripts. As a quantitative biologist, I am used to dealing with lots of complexity, but this is so complex and so method / data-collection heavy, I think that the main points will be lost to many readers as they were to me.

Other line by line comments by line order below – some of these I believe are very important while others are minor.

Is the “Warrior” race (line 225) and “yellow rust” (line 230) the same? Are we to assume that a resistance to yellow rust is the same as a resistance to the warrior race? (minor wordsmith)

On RHT8 (line 287 to 310) the authors conclude “These findings demonstrate the potential of using sequence variations in landraces to identify candidate genes for key agronomic traits.” However, they do not demonstrate / validate that they actually identified the functional polymorphism so this point is not made. Furthermore, I am not convinced that other lines would not be able to map this loci the same or better. (moderate point about study – could be minor wordsmith)

Nutritional content and composition can be altered by both increasing the nutrient and by decreasing the starch and/or protein (yield) of the crop (lines 335 to 354). When the authors say that they identified a QTL for calcium which upon induced mutation “...resulted in a greater than 10% increase in grain calcium content in five mutants...” were these tested for grain yield as well? Is it possible or likely that the starch or protein was reduced or had other pleiotropic tradeoffs? Any indication from the QTL of co-localization / pleiotropy with any other traits? From the M&M it says

“...no significant change in GW (Supplementary Table 47), demonstrating that the GrnCaCnc increase is not simply a result of a reduced grain weight.” However, I wonder if this is field / farmer relevant as it appears that these were only greenhouse grown (line 1079) from relatively few plants? This seems a little problematic. (moderate point)

The ‘tag’ SNP section (lines 370 to 384), mentions two traits with what I assume is Mendelian inheritance 5A awn inhibitor and 7A coleoptile color. Therefore, it seems a little odd that these markers were not fully penetrant in explaining the trait “...resulting in correct predictions of the two phenotypes in 87.4% and 93.7% of accessions, respectively.” Do the authors have any understanding as to why they were not 100%? (minor point that relates to my major concern around context dependency and epistasis for use of any of these loci – especially quantitative ones - in breeding.)

Lines 385 to 394 – this is only supported by supplementary figures and tables. If the authors or editors are looking for a place to cut, I am not sure this section adds much value compared to the tremendous value of some of the other components of the paper. (minor change)

The beginning of the discussion (line 391) the authors state the importance of breeding climate-resilient crops, but I remain unconvinced that the work conducted here supports this goal. Placing the lines in a free CO₂ enrichment (FACE) might? Placing them in a rainout shelter that increases heat might? Or detailed physiological studies might, even if they would have limits to extending these to field studies. The authors demonstrate that there is genetic diversity in these populations, some of this diversity could address climate-resilience. However, it seems a little disingenuous to imply these lines will directly address climate change, unless they are further tested and used in elite breeding programs (beyond the papers scope). I think this can be modified or improved with a little wordsmithing that these resources would be a great resource for defensive climate-resilience even if the authors did not directly test that here. (moderate point resulting in wordsmithing throughout, no additional fieldwork is suggested)

Line 410 – “... we transferred 44,338 Watkins-unique haplotypes from Watkins into a single elite wheat variety.” There were not this many NILS? Were the NILS not NILS, but biparental crosses or HIFs? Are they counting biparental linkage progeny? This is not clear. (very critical point that needs clarification and hard to find, could be minor wordsmithing – or these are definitely not NILS)

Line 426 – “...nutritious wheat varieties will facilitate dietary shifts towards plant-based diets, with the associated reduction in livestock greenhouse gas emissions (Fig. 4h).” I think this is a sociological question, I don’t believe the authors provide any support or evidence of this claim in their paper. Is there evidence that higher protein wheat (which is deeply constrained by culture and food processing constraints, and public skepticism on gluten) will transition people away from eating animals? Did the authors survey food consumers if there is no such reference? (minor wordsmithing – delete)

The data is not currently available at “Genome Sequence Archive (GSA) (<https://ngdc.cncb.ac.cn/gsa/>), with BioProject accession number PRJCA019636.” (line 450) , it says “CRA012590 will be available on 2025-09-12” (major problem if some data is not available, guessing minor issue to fix?)

Line 576 – “Adaptive are effects ...” does not sound right to me, but might be right. (minor)

Line 722 – what are the “limitations of using short reads for CNV operation?” (minor)

Line 816 – “was” to “were” (minor)

Line 888 – The manuscript says 9,771 RILs, however the link for availability says 8259 RILs.
(moderate area of concern)

Line 941 – A BC2F2 is not a NIL, and calling so misleads in my mind that there is a single QTL or gene isolated in a parental background. Based on line 960: “one cross with opposing parental alleles in the targeted QTL region” it seems more like a heterogeneous inbred family (HIF). (moderate, needs to be better explained in main document and wordsmithing throughout)

Line 1384 – Extend figure 1 shows “high-throughput field-based phenotyping (c.i-c.viii) of next-generation-gene-discovery populations” – I do not recall seeing this discussed or presented.
(moderate)

A. Summary of the key results – see above

B. Originality and significance: if not novel, please include reference – novel

C. Data & methodology: validity of approach, quality of data, quality of presentation – validity good from what I can tell, too much to get into details. Quality of presentation is moderate as explained above.

D. Appropriate use of statistics and treatment of uncertainties – uncertain.

E. Conclusions: robustness, validity, reliability – mostly good from what I can tell, except as discussed above.

F. Suggested improvements: experiments, data for possible revision – no additional data needed but revised presentation needed.

G. References: appropriate credit to previous work? – none that I know of, but I am not a wheat breeder / geneticist. I suggest a wheat researcher to complement my review for this component.

H. Clarity and context: lucidity of abstract/summary, appropriateness of abstract, introduction and conclusions – discussed above

Referee #3 (Remarks to the Author):

The authors provide an impressive/resource intense document detailing the novel genome level variation that remains untapped as judged from the study of released cultivars. The manuscript is undisciplined in trying to illustrate "everything" and fails to really focus on what are the key results. The case for the AG groupings in the Watkins collection, for example, and the fact that only two of the ancestral groups contributed to modern varieties is lost in Figures that are a collection of facts that are poorly described in the respective legends and sometimes not even referred to in the main text.

The significance of the manuscript is clear, as indicated at the outset. Also as indicated above the methodology is valid but presentation is undisciplined. Some examples include Fig 1d where the 1BS region representation is illustrated but does not comment on the apparent excess of AG2 lines. In Figure 2a the phylogeny of the lines in the Watkins collection does not indicate where the AG groups fit-in and, in fact, a much clearer presentation is provided in the Extended data Figure 3d. In the main text no comment is made on the fact that the AG groupings do not correspond the grouping evident in the phylogenetic representation. Figure 3e (main text) is not mentioned in the main text. In Figure 3f (main text) no scale is indicated for the Y axis.

The use of statistics and treatment of uncertainties are appropriate and conclusions are OK.

As indicated above a suggested improvement is to get a coherent linkage between the main text, Figures and Figure legends. There are too many content-free parts within the Figures and the different colors deployed are not explained in the legends.

Within the above reservations the clarity and context of the manuscript is "OK". The authors should note that TraesCS5A02G543300 (cited in the text) is transcribed at a relatively low level whereas another copy of a related gene is TraesCS5A02G004300 and is much more highly transcribed. The occurrence of multiple copies of genes needs to be considered in interpreting genome information.

Minor edits

* line 90: effect should read effects

* line 172: basis for functional significance prediction is not clear

* line 175: "no standing variation" please provide an alternative to "standing"

Responses to Editor and Referees

In this report, we have addressed all the questions and comments from the editor and the three referees (marked in *italics cyan-blue*), by our point by point responses (marked in black).

Editor

Significantly improve on the accessibility of figures and supplementary materials (everything should be readable; figures should not be crammed full with information - it is better to focus on some key data items and relegate supporting information to the SI files), and that you streamline the main article file.

Main Figures:

We very much embrace the guidance of the Editor and review team. To simplify and streamline the article we have removed **Figure 4** which summarised the use of the Watkins resource for gene identification. We now highlight these outputs through our sister manuscripts:

Blast resistance	bioRxiv 2023.09.26.559489; doi: https://doi.org/10.1101/2023.09.26.559489	
Seed calcium	Sister paper entitled "Improving wheat grain composition for human health: an atlas of QTLs for essential minerals" submitted in Nature Food with reference number "NATFOOD-23121815"	

Meanwhile, we have briefly described the key discoveries (e.g. blast resistance, seed calcium content) in the Discussion section to exemplify the value of the resource.

We retain *Rht8* but, as suggested by Referee #1, this is now integrated with our chromosome arm 7BL height/yield example shown in a new version of **Figure 3**. The addition of *Rht8* in Figure 3 has been offset by the removal of the illustration of the NAM GWAS (panel c, the data is still in the supplementary tables) as we think that the QTL plot (panel d) is enough to show the position of the locus. The QTL-haplotype analyses, a key element in our 'Haplotypes-QTL-RIL/NIL-Breeding' story.

In **Figure 2**, the specific examples of GWAS, and NAM GWAS (panel e) are removed as we think they contributed to the feeling of "clutter", but these results are still maintained in the supporting resources in a form of supplementary tables. Similarly, we retain the yellow rust resistance, another QTL-haplotype example to demonstrate our 'Haplotypes-QTL-RIL/NIL-Breeding' story. In this same figure, we removed the example of physiological trade off between heading date and yield in panel f to simplify the figure while retaining the core message.

Figure 1 and **Figure 5** (new Figure 4) is unchanged.

Extended and Supplementary Figures:

We reduced the Extended Data Figures from 10 into 5, suggested by Referee #2, to simplify the presentation in a more focused way with key novelties highlighted but without sacrificing the unprecedented scale of resources presented in our study. For example, Extended Data Figure 2 was totally removed because it is indeed crammed full of information. Because of the simplification work from the Main Text, the Main Figures and Extended Data Figures, some supplementary figures do not match any more, so the number of supplementary figures were reduced into 22.

Supplementary Tables:

Accordingly, we have reduced the number of Supplementary Tables from 52 to 37, but have made the data from the removed tables available through WWWG2B (https://wwwg2b.com/dataAvailable/Other_Supplementary_files/). We have completely removed Supplementary Tables 3, 13, 14, 17, 28, 31, 32, 33, 37, 38, 39, 40, 46, 47, 51, and 52. Additionally, we have updated the table that describes the haplotype frequency and distribution arounding the QTL peak corresponding haploblocks carried by two parents in subgroups (current Supplementary Table 24). This table provides details on the QTL haplotype analyses.

For all the changes in Figures/Tables, a summary was given in the Annex at the bottom of this document.

The narrative should surround the presentation of the resource, with some key examples of how it can be used. Please focus on the most novel aspects (see also comments made by reviewers #1 and #2). Please also take good care

not to present results here that are going to be published in separate companion papers (concern of dual publication), e.g. the wheat blast resistance story.

To allow clear distinction, and address this point and referee comments, we have removed the “Gene Discovery” section and the description of the genotyping array from the manuscript. We have now linked to three additional manuscripts which are deposited in bioRxiv. In these we describe the cloning of *Pm4*, the grain mineral atlas alongside the cloning of the calcium content gene and, the development of the 40k *TaNG* array. We have, however, added additional statements in the Discussion to highlight the immediate impact of the Watkins collection on wheat blast, as this was highlighted by Referee 1. We have now added to the Discussion that the Watkins-specific allele (*Pm4f*) which confers resistance to the Bangladesh/Zambia isolate, was absent in world modern cultivars, and is now being actively deployed in CIMMYT cultivars, which are the founder germplasm for 70% of the world’s wheat.

Referee #1

Firstly, I commend the authors for their significant work on this manuscript - this was not small task. The collective work is substantial, delivering a broad range of practical tools to support wheat improvement. I generally support the publication of the article, given its impact on the large wheat and breeding community. However, in its current state, I feel the manuscript lacks focus, and, as a consequence, details and relevance are lost to the reader. Indeed, presenting a range of topics, including variation in diversity, nitrogen use efficiency, height/flowering nutritional quality, inter-trait correlations, cloning of several useful genes (disease and otherwise), development of a new genotyping system, strategies for introgressing diversity in breeding, and QTL analysis (all useful and practical), seems overly ambitious.

I recommend that the authors reevaluate their presentation style and prioritize three to four main areas of novelty, while removing parts (that could be published elsewhere and not critical to this manuscript) and/or relegating more to supplementary information.

We agree with the comments of both referees and the Editor which point us towards our three to four main areas of novelty. We agree that these revolve around the unprecedented scale of this resource and that there is a need to accentuate our broad impact on the crop and wheat breeding community. To achieve this, we now focus on the message on the untapped genetic variation carried by wheat landraces and then connects ‘genomics’ to ‘breeding’ so that this variation can be used. We have highlighted the key novelties **and streamlined** the manuscript by removing specific topics highlighted across reviewers’ comments, but without sacrificing the unprecedented scale of resources presented in our study.

As explained above, these include:

- Removed *CBF*, *PPD1*, and *FT1* sections (224 words)
- Removed the “Watkins genomic resources accelerate gene discovery” section (1074 words) *
- Removed NUE paragraph (149)
- Removed *TaNG* array paragraph (137 words) *

Although we removed these sections, the data resource remains intact as readers have access to the underlying phenotypic data, QTL, NAM-GWAS, and GWAS via the relevant supplementary tables or associated data sets (WWWG2B, https://wwwg2b.com/dataAvailable/Other_Supplementary_files/). For the two sections indicated by asterisks, the detail of these discoveries are expanded in corresponding sister papers. For the gene cloning these are already given in a previous response for *TaNG*: doi: <https://doi.org/10.1101/2023.11.27.568448>.

In addition, we have **expanded on our strategies for introgressing diversity in breeding**. See specifically:

- **Lines 94-109** of Introduction which focuses more directly on the problem of limited genetic and genomic resources to exploit landrace diversity.
- **Line 186**, we change the title of the section to “IN-DEPTH PHENOTYPING TO VALORISE LANDRACE HAPLOTYPES” which we hope directs the reader more clearly to our focus. Previously it was “TARGETING LANDRACE QTL ALLELES FOR BREEDING”.
- **Line 337-500** We made several additions in the section “TARGETED DEPLOYMENT OF WATKINS LANDRACE VARIATION IN BREEDING”. We give the reader much more background on the backcrossing strategy and the nature of the multi environment experiments emphasising important elements of scale such as the use of 6 m² plots which are large enough to give reliable yield estimates and the use of AMMI to show how we account for GxE interactions. Instead of summarising NIL additive effects in one sentence (which we agree was confusing on its own) we now give specific examples for individual QTL/NIL families which exhibit large

and highly significant effects for traits which are important in breeding and, of course, for which the Watkins allele is beneficial compared to Paragon.

- In the DISCUSSION **lines 575-585** we draw comparison with resources in other species and specifically use landmark studies in rice as a comparator. This was a very helpful suggestion of referee 2 and we feel it helps to address the above comment of referee 1. It shows that the resource developed here is indeed unprecedented and a treasure trove for the crop and wheat community.
- **Lines 586-626**, we add some detail from our identification of *Pm4* for blast resistance and the fact that the key resistance allele *Pm4f* is only present in the Watkins Collection.
- New extended Figure 5 and **Lines 209-213** are added to make it as clear as possible to the reader that we have developed an unprecedented and fully integrated resource that they can use immediately to make new discoveries in wheat in the context of a huge existing dataset that already holds the answers to many of their questions.

Taken together we think that these changes achieve the direction of the Editor and review team, increasing the focus on our core story while maintain the impact and links to gene cloning vignettes which serve as valuable case studies for others who use the resource.

From my perspective, I found the sections on the characterization of diversity of haplotypes (and associated useful software packages developed for their utility), strategies for introgression (to capitalize on diversity) very informative and useful. Perhaps bolstering text around these and one/two examples of its utility for climate-resilient would allow more opportunity to expand on each in the main text to stress the value and would appeal to a broader audience. For example, the marker trait association studies (GWAS, QTL, NAM) are substantial, but stressing novel findings (blast resistance for example) with those haplotypes not available to modern breeding would reveal the impact/utility for improving climate resilience.

These points are addressed in our previous response.

The nutritional angle of increasing calcium content does have support from functional validation of a candidate gene and seems to this reviewer to be novel - although this trait is not tied to climate resiliency which seems to be the target focus of the manuscript.

To address the more general comments from the Editor, we have removed the “Gene Discovery” section which included the calcium content gene. The cloning of this gene is now prepared into a sister paper and already submitted to Nature Food (Nature Food with reference number “NATFOOD-23121815”, entitled “Improving wheat grain composition for human health: an atlas of QTLs for essential minerals”). We have, however, maintained the trait within the Discussion section to highlight the role that the new resource could have in cloning of genes.

The haplotype variation in and around Cbf genes (cold tolerance) and Ppd genes (flowering time), while interesting, is not particularly novel (it is well known for example that variation in CNV of Cbf genes is associated with cold tolerance) and could be removed.

In an effort to streamline the manuscript we have removed this section from the manuscript.

I found the section on developing a new 40K SNP high-throughput genotyping array lacking in the main text. I am not suggesting this work is not important, but perhaps this could be better integrated within the main text/supplemental materials or considered in another manuscript that can adequately deal with its description and comparative utility. Were any of the haplotype-based diagnostic KASP markers developed to detect the haplotype blocks included in the design?

Again, as above, we have removed this paragraph from the manuscript. However, we have maintained the concept within the Discussion section and refer readers to the more in-depth publication on the genotyping array which is now on *bioRxiv*.

Lastly, I am pleased the authors acknowledged the limitations of short-read whole-genome sequencing and mapping to a reference genome that itself was assembled using predominantly short reads. Indeed, several long-read generated-based reference sequences have recently become available for wheat, which have been shown to be better ordered and more continuous. I was hopeful these too would have been included in various analyses given this advantage. I think it is worthwhile in the discussion to stress the current limitations (especially CNV-related analysis) and that more can be captured.

We wholeheartedly agree with the thoughts of the referee on these points. In the Discussion from **lines 575 to 585** we stress exactly these limitations and our future aspirations to overcome them.

Specific Comments:

The abstract suggests that variation in modern cultivars can trace to two ancestral landrace groups. This is supported by the data for those modern cultivars assessed in the study (vs. all modern cultivars, like those with introgressions from synthetic wheat or wild relatives).

The germplasm used included many lines with wild relative introgressions such as 1BL/1RS (rye), Pch1 (*Aegilops ventricosa*), 2NS/2AS (*A. ventricosa*), *Lr19* (*Thinopyrum ponticum*) and probably many more than less well known. CIMMYT germplasm was well represented in our modern panel some of which have synthetic parents (D genome from *Aegilops tauschii*). However, we were concerned that our selection criteria might have introduced some bias (eg poor representation of Chinese germplasm). We were pleased to be able to include the data of Niu et al (see **line 144**). When we incorporated variants from that study (which included much more germplasm from the PRC and USA) we saw that the AG2/5 relationship held up as can be seen in Extended Data Fig. 2c. We show the panel below for convenience. We interpret this result as US cultivars (triangles) sitting within AG5 which is the Eastern European ancestral group. This concurs very well with historical accounts of the contribution of Ukrainian, Polish and Russian wheat to the founding lines of the US. The Chinese cultivars (squares) overlap with AG2. Most strikingly AG1,3,4,6 and 7 remain isolated.

Paragraph three of the main text discusses several obstacles in the introgression of novel alleles into wheat breeding, and that the work presented "overcomes these obstacles." Indeed, the results presented provide tools and resources to efficiently use the diversity that has been characterized, but many of the obstacles remain a challenge (e.g., dominant gain-of-function mutations, complex genetic architecture). What has been overcome is the availability of large, useful datasets and well-characterized resources that allow targeted introgression of diversity. This, in my opinion, represents the true value and overcomes the obstacle of "random" in breeding. There is a new strategy to target diversity. Perhaps this should be the focus of the "obstacles."

Thank you very much for these suggestions. Along with the comments from Referee 2, we have rewritten these paragraphs to focus the obstacles more closely with the suggestions from the reviewers.

It took a fair bit of cross-referencing various methods and extended data/supplemental tables to really understand the "maximal diversity in the 1169 cultivars from 25 countries." How does this relate to the 3500 mentioned in extended Figure 4? I may simply have missed it.

We apologise as the figure 3500 was added in error to the previous Extended Figure 4 (now it is Extended Figure 3). This value includes all of the genotypes in the original CerealsDB Axiom file. However, that included Watkins genotypes, duplicate genotypes, and RILs that are not appropriate for comparison. The true number of Axiom 35K genotyped lines that were used for meaningful selection of the diversity maximised set was 1169. The text and figure have now been amended to show this.

The paragraph starting at line 171 references functional traits, functional variations, functional SNPs, functional significance. It is not clear to me what is meant by functional. Please reword to be clear.

We agree that we have used the word functional unnecessarily in many cases with different intended meanings. See below for how we now treat each case:

- **Line 80 (Abstract)** "functionally verified genes", this is now deleted because we no longer include the gene cloning section.
- **Line 119** "WHEAT GENOMIC COMPARISONS REVEALS RICH FUNCTIONAL DIVERSITY" we now replace with "LANDRACES CONTAIN PREVIOUSLY UNTAPPED GENOTYPIC DIVERSITY" because the assignment of function to genotypic variants comes in the next section.
- **Line 176** Replaced "have functional significance" with "affect gene function" (defined by SnpEff).
- **Line 177** "functional SNPs" added "(annotated by SnpEff (v4.3t))".
- **Line 228** "depth of functional diversity" deleted because the *CBF* example is no longer highlighted.
- **Line 551** Clarified "Development of the first wheat haplotype map and haplotype–phenotype association study allowed us to estimate frequencies of these alleles and assess their functional significance" by adding "through their association with high resolution NAM QTL".
- Title of **Fig. 4** "Gene discovery and functional validation for next-generation wheat traits." Now deleted together with other occurrences of functional in legend.
- **Extended Data Fig 1**
 - "Natural and structured populations were then combined in multi-site field experiments to identify novel, functional and useful genetic variations not yet deployed in modern wheat." The word "functional" was deleted.
 - "the Watkins-unique alleles or haplotypes are functional with significant genetic effects (target QTL and MTA)" and "haplotype analysis showed that the Watkins-unique functional alleles or haplotypes are beneficial with increasing effects based on our understanding of physiological trait relationships". The word "functional" replaced with "trait associated"
- **Previous Extended Data fig 9 (now it is Extended Data fig 5)** "Percentage of AG-unique genetic diversity for SNPs, functional SNPs and beneficial haplotypes". Now: "functional SNPs (defined by SnpEff (v4.3t)) and beneficial"

Also, there is mention that there is no "Standing variation [in those regions IBS in all modern cultivars] to improve through traditional breeding." Of course, these regions can be improved through traditional breeding (intercrossing with landraces is one strategy of breeding). Perhaps rewording this to be more explicit that a wide-crossing strategy, coupled with molecular selection (as suggested), would be effective.

Thanks to the referee for helping us to clarify this point. This section now reads:

"Particularly noteworthy among these are the Watkins-unique functional SNPs (annotated by SnpEff v4.3t)⁵⁵ found near the 13,902 genes that are monomorphic in modern wheat, meaning that there is no variation present in elite wheat pedigrees to improve traits associated with these genes which are identical by state (Supplementary Fig. 17; Supplementary Tables 12-13). New allelic variation is available in and could be introduced from landraces with the aid of diagnostic molecular markers."

Line 205 mentions useful QTL alleles that increase phenotypic value when selected in the same direction. It is not clear to this reviewer what is meant by selection in the same direction. Please clarify.

We have added some explanatory text:

“We defined useful QTL alleles as those that influence a phenotypic value for traits in a direction selected in breeding. This can be always the same direction, such as increasing for yield and disease resistance, or it can provide new adaptability for selection in either phenotypic direction (Extended Data Fig. 5), such as heading date which we refer to here as adaptive traits.”

Line 221 of the main text, 1,696 genetic effects were identified as high potential for wheat improvement "based on the direction of allelic effects." Please clarify – do you mean positive effect, that is, those QTL that give substantial improvement based on QTL effect size?

See above.

Line 261 – beneficial traits? Do you mean beneficial phenotypes?

Yes, this was misuse of terminology now changed to referee suggestion. Many thanks.

I was particularly excited to see the results showing Pm4 is the casual gene underlying blast resistance in wheat. There is little variation available for wheat improvement to this devastating disease. The authors refer to a companion paper, with some results presented in a composite figure (Figure 4). The online methods section, however, did not provide this reviewer with sufficient information to really appreciate these results and to assess the robustness of the validation experiments. While I am not privy to the companion paper, it would seem this is a novel aspect of this manuscript that could be bolstered.

The detailed experimental evidence for *Pm4* as the blast resistance is now online in *bioRxiv* (<https://doi.org/10.1101/2023.09.26.559489>). To address the more general comments from the Editor, we have removed the “Gene Discovery” section. We have, however, extended the Discussion section to comment on the value of the Watkins resource for the discovery of *Pm4* and how the Watkins-specific allele (*Pm4f*) confers resistance to the Bangladesh/Zambia isolate. Importantly, this allele was absent from CIMMYT germplasm and is now being introduced as a direct result of this resource.

The paragraph starting at L261 – This section notes the positive effect of Rht1 on reducing plant height (which seems a positive phenotype) but at the expense of plant biomass. The authors then identify a height-increasing QTL on 7BL that together increased plant height (by 10 cm) and grain yield (0.39 t ha⁻¹). This section seemed awkward to this reviewer -- if the intent is to capture reduced height but higher yield (and thus biomass), why not present a QTL where short stature is maintained, but also has a positive effect on yield (and/or associated yield components). This would be quite valuable for breeding and climate resiliency (maintaining short stature but elevated yield). In the next section of the manuscript, Rht8 mapping and gene discovery was facilitated by the Watkins collection. Indeed, two candidate genes were identified where their expression is associated with phenotype. No validation of these candidates was performed to confirm their functional involvement in variable trait expression. Perhaps the authors can combine the Rht8 with the height QTL dissection characterized on 7BL into one section related to height in a more thoughtful way.

This section is now rewritten exactly as the reviewer suggests. Thank you for this suggestion.

The statistical analysis methods presented were very generalized. Given the large amount of phenotypic data conducted in various replicated experiments at multiple environments and over time, I would like to see more information on how the ANOVA was conducted (mixed models, BLUPS?). This would be important to appreciate the robustness of the analysis (especially since such large datasets can be unbalanced) and the final set of means that were used in marker trait association studies.

We have increased our description of the statistical analysis and directly addressed reviewer questions on ANOVA. The added text is:

Lines 959-964 Online Methods “The phenotypic effects observed in the NILs were used to calculate best linear unbiased estimator (BLUEs) for each NIL from a linear model (lm in R) encompassing the whole trial with replication. The phenotypic estimates for each NIL over several environments were received using AMMI (Additive Main-effects and Multiplicative Interaction) by fitting additive main effects for the NILs and environments by an ANOVA procedure

using BLUE and then apply principal component analysis (PCA) using SVD (Singular Value Decomposition) to the remaining residuals after the fitting of main effects. Statistical differences between the two different NIL alleles were tested for each NIL family as ANOVA using the AMMI-BLUEs.”

Referee #2

The greatest strength of this manuscript is the incredible amount of data that was collected and used to produce it. The greatest weakness of this manuscript is that there is simply too much “stuff” for the broad readership of Nature to walk away with a take home message(s) (I suggest a couple of the big ones below and later in my summary). I struggled to follow what the main takeaways were supposed to be and there was so much supplemental information at some point I had to give up on evaluating these figures and tables where they were called out in the manuscript. I believe the overarching theme of this paper is that there is a large amount of genetic diversity in untapped germplasm that breeders can use for wheat improvement (demonstrated) and especially improvement for climate change (I don’t think well demonstrated). This is mostly a pretty conventional theme for crop CNS publications, but this article has an unprecedented scale with a tremendous amount of data collection supporting it. I don’t mean this theme is conventional to denigrate the article, but to imply that it fits the journal and readerships expectations well in subject matter. There are some especially novel components in this article. I appreciated all of the phenotypes taken that led to a discussion and case study on tradeoffs in height (line 261 to 275). The NILs / HIFs developed and tested are an especially strong component and resource, given wheat does not cross easily, this also represents a tremendous effort (although I argue later that a BC2F2 is a HIF not a NIL, so some tempering of wording is needed).

We thank the referee for these comments and address specific points as they are raised in the more detailed breakdown below. We also agree that there is nothing wrong with following a more conventional theme of crop improvement. To this end we have changes the title, removing “for Climate Resilience”. There are several changes reflecting this shift such as changing the first two lines of **Abstract, line 70**,

A major problem this article addresses is THE major limitation to breeders and crop scientists in using germplasm collections, the lack of relevant field phenotypic characterization; the major strength of this article is the unprecedented scale of characterization that was undertaken; if anything the authors could better emphasize this point for the broad readership of Nature by comparing it to the “next largest” study in terms of accessions or RILs or genotyping etc.

Thank you very much for these suggestions, which follow closely from those of Referee 1. We have rewritten these introductory paragraphs and have now included a new paragraph in the discussion to address these comments.

Overall, the authors should, in my opinion, better emphasize their impressive field phenotypic work and findings and remove much of the excess genomics work that did not result in broadly interesting discoveries – that would greatly simplify and shorten the article.

You will see from our response to referee 1 that we have done exactly as recommended. We only show the genomic discoveries that are needed to understand the genetic approach. We have shifted our emphasis towards field phenotyping, the links between genetics and genomics, and the delivery into breeding.

More philosophically, the potential impact of this work for wheat improvement rests on two hypotheses that I take issue with as presented and deserve some thought by the authors:

- 1) *the implicit assumption that the loci discovered are purely additive without epistasis, genetic background effects, or substantial GxE interactions, ready to be used by breeders;*

We really did not intend to give the impression that we considered the genetic effects described to be fully additive with no genetic or environmental interactions. We are extremely grateful to the referee for alerting us to that fact that our writing gave this impression. Removing several sections that highlighted specific traits and genes and technologies (CBF, NUE, Blast, Ppd, FT1, TaNG- see previous replies for details) gave us the room to expand our analysis of NIL effects (previously we crudely summed the effects to try to convey impact with limited text space) including the nature of the experiments (yield and yield component estimation is realistic for breeders because the plots are large, replicated, and multi environment) as well as their statistical treatment.

2) that breeders will trust and use these loci in their own breeding efforts. In my experience, wheat breeders are among the most conservative in their belief of crossing elite by elite and not “wanting to break up those good linkage blocks” as a few breeders have told me. In other words, a century of wheat breeding purged deleterious alleles and pyramided elite ones, why go back and bring in the “junk”? I don’t think commercial wheat breeders will want to use these resources, except for a few economical simply inherited traits (yield, protein). Overall, it is also my belief that the reason exotic germplasm is rarely used in elite “practical wheat improvement through breeding” (line 250) is not simply linkage drag (linked deleterious alleles brought into the genetic background) but also because many times alleles identified in genetic studies fail to work the same way in elite genetic backgrounds (epistasis and genetic backgrounds) which is not addressed. Though this is a hypothesis, it is one of the few explanations and is supported by the authors own findings in NILS (“range of penetrance” line 257 and possibly lines 370 to 384). So, if the authors goal of the paper is to convince wheat breeders to use this pre-breeding material, the focus should be on the economical traits (protein, yield, disease, NUE) and demonstrate no linkage drag and consistent allele effects across genetic backgrounds. If the goal is to present other aspects to the general readership of Nature and policy makers, the focus should remain on much of the current presentation, but the presentation needs to be simplified and honest that this is defensive research and clear that current commercial breeders will be reluctant to use these resources for the reasons above.

We fully acknowledge, and recognise, the language that some breeders use in terms of diversity. However, our experience of working with them over the past ten years, and the resources presented in the current study, have largely changed these perceptions. Evidence of this is the fact that nine breeders representing six companies are co-authors in the manuscript. This co-authorship comes from their intimate involvement in the testing of the NILs (e.g. Fig 3g (current version) and their uptake within their breeding programs. Recognising the time it takes to generate new cultivars, commercial breeders DSV have a National Listed entry with Watkins pedigree (Wat110xRobigusxGerman). The analysis has also changed breeder mindset in terms of the “junk” when they share in our analysis of long-range haplotypes and they are fascinated to see how unchanged much of the varietal chromatin is when compared to the landraces.

It is absolutely correct that our QTL are identified in Paragon derived populations and then validated in a genetic background of Paragon. It would have been very difficult for us to target multiple varietal backgrounds at this scale but as we say in the above breeders are adopting this germplasm into their pedigrees. An important point within pre competitive pre-breeding is that at a certain point the academics have to step away as the material enters the competitive germplasm development phase. We draw the attention of the referee to a recent report from the major UK science agency UKRI on the impact of this effort, which is known in the UK as DFW – Designing Future Wheat (Evaluating the impacts of BBSRC’s investment in wheat research – UKRI). Quoted anonymously in that document some of the participating breeders said:

‘DFW has been extremely useful, as it distils down the best pre-breeding lines on offer...the lines come with markers, allowing the trait of interest to be quickly integrated and tracked through the breeding process.’

‘one of the breeders interviewed as part of this study estimated they will deliver derivatives from 5-10 lines from the DFW Breeders Toolkit in the next 2-3 years, and there will be 5-6 derivatives in their European programme within the next 10-12 years. The fact that lines from BBSRC-funded genetic material have reached National List testing is a big achievement, a ‘big tick in the box’, as stated by one commercial wheat breeder’.

So while agreeing with the referee that all we have done could be done better, exactly as suggested, we are confident that the work reported here is having immediate impact in commercial breeding but that doing more work along the lines suggested would deliver even more.

One of the line by line comments I am moving up here for broader discussion, is some of the individual “NIL” phenotypes (again, not really NILs, more like HIFs; lines 369 to 374), a very exciting and important component of the paper, but which remains lost. However, the specific claimed yield gains of the NILS “4.5 t ha⁻¹” are especially incredible, to the point that I struggle to believe them, especially from a single NIL/HIF. Given the extensive field testing reported I don’t have concerns on if it was tested in the field (although I could not find specific evidence of where this number came from in the mountain of supplemental information). If this NIL/HIF is real I think that would deserve to be it’s own article, as more detailed examination of the materials and methods is needed to understand how these results were obtained. I would especially like to see if there was a decrease in protein. There are often

pleiotropic tradeoffs between yield and protein, and although a 3.2% increase in protein was also noted, I don't believe these were in the same NIL/HIF or same genes as the yield increase? Not clear. More detail on the NIL/HIF evaluation, a confirmation of QTL/ GWAS hits discovered, would improve this paper greatly. A lot of the genomics could be deleted to make space, as there is less novelty or interesting discovery in these sections.

We were delighted to see that the referee shares our excitement for this component of the paper. We have expanded the opening paragraphs of the section "TARGETED DEPLOYMENT OF WATKINS LANDRACE VARIATION IN BREEDING". We hope that this has made the nature of the material clearer. We think it is correct to call them BC₃ derived NILs with 2-3 siblings carrying Watkins alleles compared to 2-3 carrying Paragon. Again, we apologise for the misunderstanding of those apparently enormous effects. They were summed effects from different lines. Now we break things down and thanks to the referee comments have been able to explain ourselves in more detail.

In general, all the main figures are nice, useful and important, though simplification would help. The other figures are also nice, and represent a lot of work, but many are not needed for broad readership.

Extended Data Figure 1 and 2 summarize the incredible amount of work done and also summarize my major issue with the paper: there is so much "stuff" that this group did, that I and any reader will struggle to follow it; the legends for these figures are arguably longer than some manuscripts! On the plus side, whatever methods a reader could want to find (except phenomics – a drone and spectra are shown but no evidence of this – too bad) is in this paper as something they did at very large scale, but with the combined massive volume of work it is hard to find where it was discussed, and with what resource was tested, and critically what important was found from all of this work. At the same time, while Extended Data Figure 1 and 2 focuses on what the authors did and how they did it, I am not sure what the main message is beyond that they were able to do it.

We certainly do not want these extended figures (1 and 2) to just show that we have done a lot of work. The aim was to convey the key concepts and the high-level objective. We have replaced them with an alternative version which is better aligned to the streamlined account that we have tried to achieve in response the overall feedback.

Extended figure 7 also is helpful to show scale of phenotyping. Among the extended data figures, Extended figure 4 is one of a few that shows discoveries I think the broader readership of Nature would be interested in, demonstrating Watkins diversity compared with other modern wheat cultivars. The others are either methods or intermediate findings of the paper.

We take these comments on board and have reduced number and decluttered the extended figures as well as main figures.

Extended data figure 8 also shows some QTL results, but I am not sure what the reader is supposed to take away. Of course, we expect a massive number of QTL from such a large study, so I hope that is not the point the authors are trying to make. Yet, any wheat researcher interested in a specific genomic location or trait will need to go to the tables to learn exact effects and genomic positions. I think the only useful takeaway for readers of studies showing most or all QTL in one figure like this is to present pleiotropic tradeoffs of major loci, but that was not highlighted in a visible way here.

Removed.

Extended data figure 9 is also interesting, but I think part 'g' needs to be presented not in the same line rows, as it looks like 'Yield components' were only measured in AG4 derived RILS? Actually, I am not even sure which parts do connect across the figure parts. I most like part 'f', this is great to show the positive and negative QTL effects from Watkins landraces and could be a main figure and a main point (demonstrating the benefit of diversity + linkage drag), however does part 'f' connect with 'e'? with 'b'? with 'g'? Maybe rotating some parts 90 degrees to columns could help?

Thanks for the valuable comment to improve the Extended Data Figure 9. It is (now is Extended Data Figure 5) updated. Pasted below.

Extended data figure 10 may be the best for the readership of Nature, it shows a trait of broad interest, its genetic dissection and how it is unique distributed in the landraces – this alone could be a singular focus of the article. However, I scarcely remember the discussion of this nitrogen trait in the article – it would need to be made more memorable. Also I am unclear if there were NILs or other specific resources to validate it? If it was simply an identified QTL that was not validated by NILs in a separate relevant experiment with appropriate field data, this would not be reliable and not a major interest of the paper.

There are many individual QTL effects that we might have dwelt on but in taking on the idea of really focussing on the general interest of WWWG2B for Nature readers we have completely shifted emphasis to the description and accessibility of the resource. In that context the individual QTL examples seem out of context. We feel that readers who become users of the resource will actually get more value from using the tables, associated data, and germplasm to explore the whole resource. So we have removed this figure.

I feel fatigued after reading this massive, and very impressive, tour de force manuscript (as the authors probably do by this review). I do not envy the writers putting this all together. I apologize for the delay in my review but in my hundreds of reviews and papers I have edited I have never seen anything like this, it seems fair to say this is about 10 manuscripts in one, and quickly I found myself taking shortcuts from my normal reviewing process of skipping supplemental work as it was called out. There was just too much supplemental information and too many complicated figures to easily follow. I fear readers will feel the same way, that is why some revision of presentation is need to focus on what was discovered and message around that and not on the massive amounts of work conducted.

We agree that the previous ms was somewhat overwhelming. The changes we have made are very helpful in allowing results to flow. We hope that our other responses, most of which are aimed at addressing this overall point, have been helpful.

In summary, there is something interesting and very important here for the readership of Nature. I think it is 1) putting into perspective the unprecedented scale needed for characterization of genetic resources, and a few case study discoveries that this study allowed that past studies have not, AND discussing what this characterization requires (\$, hours, etc.) and why it has not been routinely done in the past. If people want to use genetic dissection for to improve wheat and wheat resilience, make them understand how underinvested this aim is and that studies like

this are needed (if the authors think they are, I do). This must be followed up with 2) the validation through NILS/HIFS and field trials, without this effort I would suggest publishing in a specialty journal as another massive unconfirmed QTL / GWAS study (although with the amount of data, I wonder if cross validation could also be used for validation in this case?). I also think the main point that 3) there is much diversity that is missing from modern wheat, existing in landraces and germplasm banks and this diversity won't get used unless it is phenotypically characterized, and a few case studies of that.

We have fully embraced the spirit of these comments and hope that the changes described above reflect this. In our Discussion we try to get the point across by comparing to landmark studies in rice and putting a few of them together to try to make an equivalent virtual resource to WWG2B (<https://www.wg2b.com/>). This is in **lines 627-633** of the Discussion.

For the broad readership of Nature, everything else seems like showing data for the sake of showing data and is just too much for a reader to digest. Targeting 10 supplemental tables and 10 supplemental figures, and 5 extended data figures seems a good goal compromise. Everything else could be split into different manuscripts. As a quantitative biologist, I am used to dealing with lots of complexity, but this is so complex and so method / data-collection heavy, I think that the main points will be lost to many readers as they were to me.

We have simplified by putting the Gene Discovery section into other manuscripts (see comments above). We have also reduced supplementary tables and simplified figures as much as possible. Given the importance from Reviewer 1 that all data is findable in the Tables we did not want to cut these beyond what was strictly required. We have therefore tried to find a balance and ensured that all the data are still accessible via WWG2B.

Other line by line comments by line

Is the "Warrior" race (line 225) and "yellow rust" (line 230) the same? Are we to assume that a resistance to yellow rust is the same as a resistance to the warrior race? (minor wordsmith)

Yes. We have made this clarification in the text.

On RHT8 (line 287 to 310) the authors conclude "These findings demonstrate the potential of using sequence variations in landraces to identify candidate genes for key agronomic traits." However, they do not demonstrate / validate that they actually identified the functional polymorphism so this point is not made. Furthermore, I am not convinced that other lines would not be able to map this loci the same or better. (moderate point about study – could be minor wordsmith)

We have followed the suggestion of integrating RHT8 results into the chromosome 7BL height/yield section. This is relevant as non-GA routes to height reduction will be needed in future cultivars. It also demonstrates a powerful use of the resource which may not be immediately obvious. We did not need to sequence Mara (carrying RHT8) because the Mara haplotypes were present in Watkins. Our phylogenetic analysis suggest that this would be true for most loci/alleles, so it flags up another general use of the resource for marker design and candidate gene analysis without needing to invest in sequencing.

Nutritional content and composition can be altered by both increasing the nutrient and by decreasing the starch and/or protein (yield) of the crop (lines 335 to 354). When the authors say that they identified a QTL for calcium which upon induced mutation "...resulted in a greater than 10% increase in grain calcium content in five mutants..." were these tested for grain yield as well? Is it possible or likely that the starch or protein was reduced or had other pleiotropic tradeoffs? Any indication from the QTL of co-localization / pleiotropy with any other traits? From the M&M it says "...no significant change in GW (Supplementary Table 47), demonstrating that the GrnCaCnc increase is not simply a result of a reduced grain weight." However, I wonder if this is field / farmer relevant as it appears that these were only greenhouse grown (line 1079) from relatively few plants? This seems a little problematic. (moderate point)

As stated in our general response to the Editor, we have removed the “Gene Discovery” section therefore these results are no longer shown in the current manuscript. We do acknowledge, however, the points made by the referee and made sure to address them in the companion paper which describes the grain mineral atlas and the cloning of the calcium content gene which is now submitted to Nature Food (submission reference number “NATFOOD-23121815”, entitled “Improving wheat grain composition for human health: an atlas of QTLs for essential minerals”). We thank the referee for raising these points.

The ‘tag’ SNP section (lines 370 to 384), mentions two traits with what I assume is Mendelian inheritance 5A awn inhibitor and 7A coleoptile color. Therefore, it seems a little odd that these markers were not fully penetrant in explaining the trait “...resulting in correct predictions of the two phenotypes in 87.4% and 93.7% of accessions, respectively.” Do the authors have any understanding as to why they were not 100%? (minor point that relates to my major concern around context dependency and epistasis for use of any of these loci – especially quantitative ones - in breeding.)

As stated in the main text, both traits are quantitative in nature and there are several loci which control each one. In the case of red coleoptile, the major locus is on chromosome 7A and this is the one we targeted. However, there is also natural variation for red coleoptile which maps to the homoeologous regions on chromosomes 7B and 7D (<https://shigen.nig.ac.jp/wheat/komugi/genes/symbolClassListAction.do?geneClassificationId=141>). Similarly for awns, the major locus is on chromosome 5A, however there are additional known QTL (for example on chromosomes 4A and 6B: <https://doi.org/10.1371/journal.pone.0176148>). Hence for both traits we expected to have enrichment as we targeted the major known locus, but did not expect to have full penetrance with the tested marker.

Lines 385 to 394 – this is only supported by supplementary figures and tables. If the authors or editors are looking for a place to cut, I am not sure this section adds much value compared to the tremendous value of some of the other components of the paper. (minor change)

Thank you for this suggestion. As stated in our general response to the Editor, we have removed this section to streamline the manuscript.

The beginning of the discussion (line 391) the authors state the importance of breeding climate-resilient crops, but I remain unconvinced that the work conducted here supports this goal. Placing the lines in a free CO2 enrichment (FACE) might? Placing them in a rainout shelter that increases heat might? Or detailed physiological studies might, even if they would have limits to extending these to field studies. The authors demonstrate that there is genetic diversity in these populations, some of this diversity could address climate-resilience. However, it seems a little disingenuous to imply these lines will directly address climate change, unless they are further tested and used in elite breeding programs (beyond the papers scope). I think this can be modified or improved with a little wordsmithing that these resources would be a great resource for defensive climate-resilience even if the authors did not directly test that here. (moderate point resulting in wordsmithing throughout, no additional fieldwork is suggested)

Consistent with the reviewer’s comments and as indicated in our response above, we have now modified wording throughout, including the title, to present a more nuanced and balanced perspective.

Line 410 – “... we transferred 44,338 Watkins-unique haplotypes from Watkins into a single elite wheat variety.” There were not this many NILS? Were the NILS not NILS, but biparental crosses or HIFs? Are they counting biparental linkage progeny? This is not clear. (very critical point that needs clarification and hard to find, could be minor wordsmithing – or these are definitely not NILS)

We have detailed the revisions to this section in our response to Reviewer 1. We believe these revisions will clarify the points raised here.

Line 426 – “...nutritious wheat varieties will facilitate dietary shifts towards plant-based diets, with the associated reduction in livestock greenhouse gas emissions (Fig. 4h).” I think this is a sociological question, I don’t believe the authors provide any support or evidence of this claim in their paper. Is there evidence that higher protein wheat

(which is deeply constrained by culture and food processing constraints, and public skepticism on gluten) will transition people away from eating animals? Did the authors survey food consumers if there is no such reference? (minor wordsmithing – delete)

We have deleted this phrase as suggested.

The data is not currently available at “Genome Sequence Archive (GSA) (<https://ngdc.cncb.ac.cn/gsa/>), with BioProject accession number PRJCA019636.” (line 450) , it says “CRA012590 will be available on 2025-09-12” (major problem if some data is not available, guessing minor issue to fix?)

Thank you. We are fully committed to making all our raw and processed data publicly available. The initial deposit of all the raw data was done several months ago (it took weeks to transfer!) and we included a tentative future date for future release as we did not know at that time of the timelines for the revisions. But now, all the sequencing data are all finished in data uploading, and 100% available to download

Line 576 – “Adaptive are effects ...” does not sound right to me, but might be right. (minor)

We have adjusted the wording of the legend here and elsewhere.

Line 722 – what are the “limitations of using short reads for CNV operation?” (minor)

We developed an integrated pipeline to identify CNV based on the read depth variation across the genome and across the population (see Method, lines 1020-1043), to minimize the identification errors caused by the limitations of using short reads for CNV operation. A workflow was given here for a better understanding for the referees.

In general, these limitations (and the taken solutions in this study) includes: 1) short read sequencing data cannot properly distinguish complex repetitive elements of wheat genome, so we only focus on the wheat protein-coding genes in our analyses; 2) short read sequencing data cannot properly distinguish similar paralogous genes duplicated recently, so we analysed each of the protein-coding genes by an all-vs-all CDS alignment using BLASTN and set strict criteria to reduce the depth bias introduced by recent gene duplications; 3) we performed normalization to reduce the sequencing depth bias between accessions; 4) we reduced GC bias correction to obtain the most accurate read depth for each gene in each accession, following the optimized GC distribution across the genome; 5) we also performed read depth correction caused by complex deletions and insertions that the short read technology cannot properly identify. By these sophisticated pipelines, we have confidence that we already produced a high-quality matrix for the CNV for each of the wheat protein-coding genes. However, we indeed need the long-read sequencing technology in the future to continue improve the identification of CNV.

Line 816 – “was” to “were” (minor)

Thank you, we have corrected.

Line 888 – The manuscript says 9,771 RILs, however the link for availability says 8259 RILs. (moderate area of concern)

We list 109 populations in the supplementary table, which are available to the public. These populations contain in total 10,259 RILs (number updated from 9,771 RILs due to a detailed check of new populations when DNA was prepared for genotyping). Currently, 90 populations, with a total of 8359 RILs, are listed by GRU/Seedstor - we assume these are the available RILs the reviewer is referring to.

We mention in the caption of the supplementary table that some populations are awaiting an access code from GRU/Seedstor. Once these codes have been issued, after a Seedstor database update, the missing 19 populations with 1,900 RILs will become visible and also easily available.

Line 941 – A BC2F2 is not a NIL, and calling so misleads in my mind that there is a single QTL or gene isolated in a parental background. Based on line 960: “one cross with opposing parental alleles in the targeted QTL region” it seems more like a heterogeneous inbred family (HIF). (moderate, needs to be better explained in main document and wordsmithing throughout)

The lines under discussion are effectively BC3 as we take a RIL (50% Watkins), cross again to Paragon (25% Watkins), and then again (12.5% Watkins). We then self fertilise and extract 2-3 homozygotes of each allelic class, they for the basis of our field comparisons. We are not sure that HIF is the right designation as the background is really close to Paragon. When the material is grown in the field everything looks like Paragon and the grand phenotypic means for the trial are the same as for the Paragon controls. We have expanded the description of these materials in the Results section so we feel that NIL is the best description, and the reader will not be under any illusion regarding the level of background.

Line 1384 – Extend figure 1 shows “high-throughput field-based phenotyping (c.i-c.viii) of next-generation-gene-discovery populations” – I do not recall seeing this discussed or presented. (moderate)

We have decluttered and adjusted Extended Figure 1 and removed the “high-throughput”. We used various approaches for the phenotyping efforts in the past decade and both in the UK and in China, the unprecedented scale of phenotypic datasets were collected and analysed in a centralized and consistent manner. Although some high-throughput field-based phenotyping has been used to assess field trials, this was not the case for the majority of trials.

Referee #3

The authors provide an impressive/resource intense document detailing the novel genome level variation that remains untapped as judged from the study of released cultivars. The manuscript is undisciplined in trying to illustrate “everything” and fails to really focus on what are the key results. The case for the AG groupings in the Watkins collection, for example, and the fact that only two of the ancestral groups contributed to modern varieties is lost in Figures that are a collection of facts that are poorly described in the respective legends and sometimes not even referred to in the main text.

We fully accept that we lacked discipline in the preparation of the previous ms, we included too much stuff and moved too freely between levels of description, from germplasm, genomics resources, genetics resources, phenotypic resources, and the generation of an academic toolkit (QTL, MTA) and breeding toolkits (new array, diagnostic SNPs to introgressome in pre-breeding), which has been proved challenging for our readers to capture the core message. We very much appreciate this feedback and hope that the overall changes in the text, figures and tables have gone a long way to rectifying the situation (see the responses to the comments from the editor and the other two referees). It was a particular worry to hear that results suggesting AG2/5 domination of modern wheat were lost. We have removed significant volumes of text, figures, and tables to try to get the balance right.

As indicated above a suggested improvement is to get a coherent linkage between the main text, Figures and Figure legends. There are too many content-free parts within the Figures and the different colors deployed are not explained in the legends.

Fig 1d where the 1BS region representation is illustrated but does not comment on the apparent excess of AG2 lines.

Thanks for this nice suggestion. We emphasized that the IBS-based long-haplotype analysis also demonstrated the AG2/5 origins of the modern wheats, as written in the text, Lines 155-159: “These IBS segments are signatures of the close relatives of Watkins landrace accessions that were the founder lines of modern wheat cultivars, which have retained high chromosome-level identity with AG2/5 landraces, often across multi-megabase tracts extending across the majority of a chromosome’s length (Supplementary Fig. 11).”

In Figure 2a the phylogeny of the lines in the Watkins collection does not indicate where the AG groups fit-in and, in fact, a much clearer presentation is provided in the Extended data Figure 3d.

Thanks for pointing out this observation. Although largely overlapped, there is an indeed slight difference between the population structure of ancestral groups generated by ADMIXTURE (Maximum Likelihood estimation) and the phylogenetic analysis that infers accession relationship either by Maximum Likelihood estimation (updated) or by

neighbor-joining methodology (the previous version). This is mostly caused by the extensive admixture with complex reticulate relationship among these ancestral groups/accessions, for which more advanced algorithm is needed. We thus added more explanation in our revised Main Text, Lines 129 – 135: “It is worth noting that a subjective observation of the phylogenetic analysis of Watkins based on the maximum likelihood (Extended Data Fig. 2d), largely places individual accessions of each ancestral group beneath common branchpoints of the phylogenetic tree. Instances where this is not observed reveal extensive admixture (Extended Data Fig. 2a), notably seen in AG1 and AG3. This reflects the reticulate relationship among these ancestral groups and further demonstrates the need for alternative sophisticated methods when the aim is ancient wheat population identification¹⁰.”

Figure 3e (main text) is not mentioned in the main text. In Figure 3f (main text) no scale is indicated for the Y axis.

Thanks for the comment. Now these are corrected and updated.

In the main text no comment is made on the fact that the AG groupings do not correspond the grouping evident in the phylogenetic representation.

Thank you for highlighting this difference. As stated above, we now have addressed this issue by adding more explanation.

The authors should note that TraesCS5A02G543300 (cited in the text) is transcribed at a relatively low level whereas another copy of a related gene is TraesCS5A02G004300 and is much more highly transcribed. The occurrence of multiple copies of genes needs to be considered in interpreting genome information.

Thank you for highlighting this. As stated in our response above, we have removed the “Gene Discovery” section which included the section referring to gene *TraesCS5A02G543300*.

Minor edits

** line 90: effect should read effects*

We have adjusted accordingly.

** line 172: basis for functional significance prediction is not clear*

We have replaced “have functional significance” with “affect gene function” and then state that this was defined by SnpEff.

** line 175: "no standing variation" please provide an alternative to "standing"*

We have modified the wording here to avoid confusion. Thank you.

Annex 1. Below is a comparison of the changes in our manuscript between the previous and this revised version.

Main Figures

The previous version	Changes	This revised version
Figure 1	/	Figure 1
Figure 2	Panel A: Tree display constructed using ML methods;	Figure 2
Figure 3	Deleted NAM GWAS example for PH; Deleted GWAS example for CBF; integrated 7BL and Rht-8.	Figure 3
Figure 4	Removed Gene Discovery on blast disease resistance and calcium (as independent sister papers and mentioned in the Discussion section); moved Rht-8 content into the new Figure 3.	Removed
Figure 5	/	Figure 4

Extended Data Figures

The previous version	Changes	This revised version
Extended data figure 1	simplify panel C, deleted EMS/TILLING because we removed calcium validation work into sister paper.	Extended data figure 1
Extended data figure 2	Removed	Removed
Extended data figure 3	Panel D: updated phylogenetic tree reconstruction using ML methods replacing the previous NJ tree.	Extended data figure 2
Extended data figure 4	/	Extended data figure 3
Extended data figure 5	Removed	Removed
Extended data figure 6	Removed	Removed
Extended data figure 7	/	Extended data figure 4
Extended data figure 8	Removed	Removed
Extended data figure 9	Connecting Panel f with Panel g to indicate the origins of these traits deriving from the specific QTLs.	Extended data figure 5
Extended data figure 10	Removed	Removed

Supplementary Figures

The previous number	Update	Current number
Supplementary figure 1	/	Supplementary figure 1
Supplementary figure 2	/	Supplementary figure 2
Supplementary figure 3	/	Supplementary figure 3
Supplementary figure 4	/	Supplementary figure 4
Supplementary figure 5	/	Supplementary figure 5
Supplementary figure 6	/	Supplementary figure 6
Supplementary figure 7	/	Supplementary figure 7
Supplementary figure 8	/	Supplementary figure 8
Supplementary figure 9	/	Supplementary figure 9
Supplementary figure 10	/	Supplementary figure 10
Supplementary figure 11	/	Supplementary figure 11
Supplementary figure 12	/	Supplementary figure 12
Supplementary figure 13	/	Supplementary figure 13
Supplementary figure 14	/	Supplementary figure 14
Supplementary figure 15	/	Supplementary figure 15
Supplementary figure 16	/	Supplementary figure 16
Supplementary figure 17	/	Supplementary figure 17
Supplementary figure 18	/	Supplementary figure 18

Supplementary figure 19	/	Supplementary figure 19
Supplementary figure 20	Removed	Removed
Supplementary figure 21	Removed	Removed
Supplementary figure 22	Removed	Removed
Supplementary figure 23	Removed	Removed
Supplementary figure 24	Removed	Removed
Supplementary figure 25	Removed	Removed
Supplementary figure 26	Removed	Removed
Supplementary figure 27	/	Supplementary figure 20
Supplementary figure 28	Removed	Removed
Supplementary figure 29	/	Supplementary figure 21
Supplementary figure 30	Removed	Removed
Supplementary figure 31	/	Supplementary figure 22

Supplementary Tables

The previous number	Update	Current number
Supplementary table 1	/	Supplementary table 1
Supplementary table 2	/	Supplementary table 2
Supplementary table 3	Removed	Removed
Supplementary table 4	/	Supplementary table 3
Supplementary table 5	/	Supplementary table 4
Supplementary table 6	/	Supplementary table 5
Supplementary table 7	/	Supplementary table 6
Supplementary table 8	/	Supplementary table 7
Supplementary table 9	/	Supplementary table 8
Supplementary table 10	/	Supplementary table 9
Supplementary table 11	/	Supplementary table 10
Supplementary table 12	/	Supplementary table 11
Supplementary table 13	Removed	Removed
Supplementary table 14	Removed	Removed
Supplementary table 15	/	Supplementary table 12
Supplementary table 16	/	Supplementary table 13
Supplementary table 17	Removed	Removed
Supplementary table 18	/	Supplementary table 14
Supplementary table 19	/	Supplementary table 15
Supplementary table 20	/	Supplementary table 16
Supplementary table 21	/	Supplementary table 17
Supplementary table 22	/	Supplementary table 18
Supplementary table 23	/	Supplementary table 19
Supplementary table 24	/	Supplementary table 20
Supplementary table 25	/	Supplementary table 21
Supplementary table 26	/	Supplementary table 22
Supplementary table 27	/	Supplementary table 23
Supplementary table 28	Removed	Removed
/	updated table that describes the frequency distribution of haplotypes carried by two parents in subgroups. This table provides detailed insights into the QTL peak marker block-haplotype analyses	Supplementary table 24

Supplementary table 29	/	Supplementary table 25
Supplementary table 30	/	Supplementary table 26
Supplementary table 31	Removed	Removed
Supplementary table 32	Removed	Removed
Supplementary table 33	Removed	Removed
Supplementary table 34	/	Supplementary table 27
Supplementary table 35	/	Supplementary table 28
Supplementary table 36	/	Supplementary table 29
Supplementary table 37	Removed	Removed
Supplementary table 38	Removed	Removed
Supplementary table 39	Removed	Removed
Supplementary table 40	Removed	Removed
Supplementary table 41	/	Supplementary table 30
Supplementary table 42	/	Supplementary table 31
Supplementary table 43	/	Supplementary table 32
Supplementary table 44	/	Supplementary table 33
Supplementary table 45	/	Supplementary table 34
Supplementary table 46	Removed	Removed
Supplementary table 47	Removed	Removed
Supplementary table 48	/	Supplementary table 35
Supplementary table 49	/	Supplementary table 36
Supplementary table 50	/	Supplementary table 37
Supplementary table 51	Removed	Removed
Supplementary table 52	Removed	Removed

Reviewer Reports on the First Revision:

Referees' comments:

Referee #1 (Remarks to the Author):

First, I want to congratulate the authors on a revised manuscript that has dealt with the majority of reviewers comments in a productive way. In my opinion, the manuscript is greatly improved and is much easier to follow. Indeed, there are vast data sets and analyses that took this reviewer much time to again assess and comprehend -- but this I believe is the strength of the manuscript – a vast dataset of appropriate genetic/genomic resources that together will have a significant impact on wheat improvement. Shortening the manuscript with focus is welcomed, and the figures of main text are improved.

Some comments that I think are still worth while to address:

1. The authors indicated that they have developed sufficient genomic, genetic and phenotypic resources that will allow breeders to harness untapped wheat landrace diversity. While genetic and genomic resources pave the way to utilizing these resources, there is still the significant bottleneck of recombination and and potential linkage drag. This is difficult to overcome despite genomic and phenotypic resources. I would like to see this added, as it implies genetic and genomic resources alone will pave the way to effective utilization of resources. This is partly true, but the onus will still be on breeders to effectively recombine these alleles with the adaptative variation already regionally deployed.
2. Pg 5 132. The authors note that their analysis reflects “the reticulate relationship among ancestral groups and further demonstrates the need for alternative methods when the aim is ancient wheat population inference”. Could the authors clarify what alternative methods and for what purpose. The sentence seems almost incomplete.
3. Page 8, 209 This can be always the same direction, such as increasing for yield and disease resistance..... This sentence was awkward to this reviewer.
4. The authors note that although RHT8 has been mapped at high genetic resolution, breeders do not count with diagnostic markers. What is meant by counting with diagnostic markers? I was a bit perplexed as to what this meant.
5. On Page12 L305, the authors note haplotypes were identified that resulted in 0.91 tonnes per hectare increase. This sounds impressive, but does not provide context because an increase must be “relative to something”. For example, a 0.91 tonne increase would represent a 25-35% increase in grain yield in my regional breeding program, but perhaps 10-15% in a higher yielding environment. It may be worthwhile here to also indicate a percentage increase relative to the trial mean.
6. P12 L307. The authors indicate that they identified five haplotypes that conferred significant increases in grain protein, there of came without a negative impact for grain yield. Can the authors point to the data in the main text? I did have to dig to find it. I think its worth while that claims in the main paper are referenced to the supporting data – especially given this is indeed a significant finding.
7. On page P12 L315. The authors note that the arithmetic sums of allelic effects are helpful in conveying the breeding potential of this resource. This is true (there are other ways). I was however

confused at the sentence “Looking at significant yield component effects in this way reveals a 4.5 t ha⁻¹ increase in grain yield, an increase of 11,500 grains, and an increase in thousand grain weight of 55.6 mg. I could not ascertain the relevance of these, where these calculations were coming from and what it really means relative to the mean. These appear to be impressive increases, and I want to be sure the authors reference the appropriate data so that readers can appreciate the meaning.

8. P13 L342: Development of the first wheat haplotype map. I am not sure this is the first, given previous haplotype maps of wheat being published. Please confirm.

9. I appreciate that the authors have reduced the main text and have worked to introduce those parts that were cut into the discussion section. However, to me the discussion seems rushed, and not yet fully interconnected. For example, the authors noted that their study showed that the key resistance allele Pm4f was only detected in Watkins. This reviewer is familiar with this resistance, but a general audience would not be. Also, how can the authors be sure that the Pm4f allele is now being deployed in breeding programmes worldwide as a direct result of the Watkins resources?

In my last review I had asked the reviewers to stress the limitations of short reads, and their response indicates a discussion point devoted to this. Perhaps I missed it but I don't see it other than their work mentions recent long read assemblies. My preference would be that the authors note the limitations of the data sets. For example, structural variation maybe missing from their datasets. There are other limitations as well. This is not to be critical to the work that was done, but I feel it would be good to point out these limitations and to stress that there is likely more variation that will still need to be characterized.

The authors note that Watkins resources have also been used to create a mineral atlas for wheat with new variants for the breeding of more nutrient dense wheat (Nature Food submission with reference number “NATFOOD-23121815”, entitled “Improving wheat grain composition for human health: an atlas of QTLs for essential minerals”). What are these references? Please clarify.

Referee #1 (Remarks on code availability):

While I am not an expert on all the codes/analysis provided, the analyses fits to that which is typical of the discipline (including QTL/GWAS, field trials, statistics).

Referee #2 (Remarks to the Author):

Requested points for reviewers to address.

A) The key results remain the same. There is great untapped diversity for crop improvement in a unique wheat germplasm collection and the authors do extensive work to characterize the phenotypic and genomic diversity in various ways. This demonstrates the value of ex situ collections for food security through crop improvement as well as the extensive amount of effort needed to meaningfully tap into this diversity.

B) This work is highly original and significant.

C) The data is valid and of high quality. The presentation is very good, but more editing should be done to make it excellent.

D) Statistics and uncertainty treatment is appropriate.

E) The conclusions are robust, valid, reliable and strong. Some editing could better bring out the main points for readers still but these are much improved.

F) No additional experiments or data are suggested, arguably there are already too many experiments for readers to comprehend. However, improvements in clarity of presentation could still be made.

G) References. Seemed appropriate but I did not fully investigate.

H) Clarity and context is much improved and good, but some minor edits and reorganization of the main paper could make it excellent.

Overall this manuscript is much improved and simplified, responded well to reviewers, and is a very important study and among the most impressive data sets and reporting of such important information; so it should be published quickly. However, as discussed in the last review, with such an impressive study comes very difficult challenges in clear and effective presentation on the discovery and potential impacts. I found it was still painful (but much less so!) to go through as a reader / reviewer and I am sure even more painful for the authors (I was especially impressed how quickly they made major revisions without errors or typos – except for maybe in M&M). It was most painful because the figures were in 3 separate places rather than in order of callout- not sure Nature allows them to be placed in body of manuscript for review, but would have made it much more pleasant to read such a massive amount of work.

My main major remaining issues with the manuscript is that I think there is still excessive genomics information unimportant for the main message and discussed as methods or results, but not as a discovery or outcome – which is what readers will care about. I think the authors still focus on genomics variation being interesting in and of itself, when over the last 10 years as inexpensively “sequencing anything” has become feasible, researchers and breeders are now focused on functional diversity being the interesting component, of which the authors have lots of data and demonstrate, but this gets lost. The excessive genomics focus is especially (but not exclusively) true in the supplemental figures and materials and methods – I suggest there is no reason to call most of these genomic supplemental figures out in the main text with little discussion. As an example - Supplementary Figures 1, 2 will likely only be of interest to reviewers to check the data methods were appropriate and be impressed with data volume. There are no biological insights from this. One possibility is to put all of this genomics quality control “stuff” into a supplemental note and only refer to it there, so then in the main manuscript the authors can just say extensive quality control was applied to this sequence data (see Supplementary Note). As a separate genomics example in Supplementary Figure 9 – a figure for pi is used, but not really discussed in a way that shows it is meaningful to readers, more of an academic exercise; similarly Supplementary Figure 10 presents

the locations in gene models this diversity is located, but why should readers care since the functional value is what matters? Both S9 and S10 and others are trying to show with different statistics that the diversity in Watkins is unique but how many ways does this need to be presented; is it a major finding leading to new insights? Again, moving this to a supplemental note (e.g. “uniqueness of Watkins genomic diversity”) containing all related figures that a minority of readers will be interested in could make understanding the major findings easier. Overall these genomics supplemental figures focused strongly on the volume of data and insinuated insights, which I found to be a distraction from the important main messages and had lost enthusiasm by the time I got to those locations. If Nature will not let supplemental figures only be referred to in supplemental notes, I suggest continuing to reduce the supplemental figures and Materials and Methods (or move some M&M to supplemental), asking the important question of what the main discoveries were and what a reader would be interested in? Given the number of genomics supplemental figures, I would expect there to have been new methods / genomic analysis approaches that had to be developed which 1) would probably better fit a specialty journal, and 2) are not clarified as innovations in the main manuscript – so maybe they are not novel and just standard QC and statistical analyses?

Looking over them again, supplementary Figures 3,4,5,8,11,17,19 and maybe 13,15,20 are the only ones I think are important for the main messages in the main manuscript. The rest I think could just be in supplemental notes (or if not allowed – deleted).

Overall this review is meant to be helpful and I hope this does not come across as overly critical, there is a lot of great work in this manuscript and the authors have done a great job, especially in revision and should be commended. Yet it is still a lot for readers and reviewers to digest. Continuing to try to see it from a target audience reader and reviewers perspective and presenting it more clearly will lead to greater visibility and impact from all of their work.

A few other general notes

The name cultivars should be bracketed by apostrophes in first use (e.g. ‘Jagger’) following plant breeding convention and including in figures. Lines with release notes (or PVP, patents) should be cited, this is important to credit the developers of these lines. Both recommendations are a best practice, unless Nature requires otherwise.

I remain unclear on what ‘Watkins’ is. I thought it was a collection, but in Supplementary Fig. 20 and in text it appears that it is a cultivar? Please clarify and explain.

I am still unclear on how the “15” and “18” cultivars were selected, why, and what they were used for. A couple more sentences on this would help clarify. Maybe I missed it.

The authors mention epistasis in the response to reviewers but I did not see it acknowledged in the main manuscript and is a caveat that breeders recognize but molecular breeders rarely do – there should be a sentence on this, that not all alleles performed as expected in a different genetic background.

Line by line comments:

Line 102-103: The challenge is not a lack of associated sequence information, it is the trait and the phenotyping, which this study closes the gap on. Saying the sequence information is important is putting the cart before the horse. I think if it was a marker for selection that would make more sense as important for breeders, but sequence information is not really needed in and of itself for breeding?

Line 120: If the goal is to discover underutilized landraces it should be variation and not variations?

Line 121+: I did not find supplementary Table 1 (or others) when I downloaded the Zipped file of everything from Nature review website and had to review this offline. So, I did not evaluate supplemental tables during my review, but found them on the website as I was about to submit my review. I apologize to the authors and editors but I do not have time to evaluate these now for timely review submission.

Line 124/ Sup fig. 2: It is not clear what was used as the reference allele?

Line 136: Unclear what the 15 previously described accessions are? Each article should stand alone. I have three different windows open to try to follow this and should not have to dig up a publication to determine what they are or why they were selected. Maybe these are the 15 lines in figure 1b?

Line 150: Sup Fig. 9 does not show the point or add much to manuscript. Sup Fig 10b I am not sure what AG unique is, if it is not Watkins unique 10a. Is 10b unique to AG groups within Watkins and 10a is Watkins unique to modern cultivars? Some minor additions to figure description would clarify.

Line 154: I don't understand Figure 1d? Is this averaging what is shown in Sup Fig 11 or how was this calculated differently?

Line 158: on Sup Fig. 11 – how were these “18” chosen? Why do they differ from the “15” mentioned earlier (if they do)?

Line 160: I can not imagine how someone will use Sup. Fig 12. Furthermore, this is expected. I think this figure could be deleted or moved to just in the supplemental note. At least the novelty should be discussed if there is novelty, conventional knowledge would have predicted this and I am looking for places to simplify.

Line 179: Why do the colors switch between modern and Watkins in part A and B of Sup. Fig. 17. It took me awhile to figure this out.

Up to this point, many of the numerous figures (supplemental especially) seemed relatively unnecessary to main message as they focus on genomic differences and not functional differences. I would encourage the authors to halve these or move just to supplemental note callouts.

Line 191: Sup. Fig. 18 - Most of these images are unnecessary and don't provide insight. The map is

useful but I think presented elsewhere?

Line 210: I still find this Extended Data Figure 5 confusing, not sure if it was updated as authors say in revision? Parts a, b, c make sense (although I feel like these were presented earlier and are more about germplasm diversity). Part e,f,g are pretty difficult to discern. The color on part Extended Data Figure 5e I am guessing is supposed to match to something? I can't see color well (the black outline could be removed) and it does not line up with part 5b? It mentions 66 RIL parents, but 72 are discussed before then, what happened to the other 6? Part 5f is clear, but why order so they don't line up with 5g? We cannot see where the curved lines go? Parts 5f and 5G are the only important parts of the figure as far as I can tell (assuming 5a and 5c were presented earlier). I suggest an entirely new figure easier for readers focused on parts e,f,g.

Line 219: Figure 2e does not support the statement, it shows a QTL but does not show the 33 accessions?

Line 252: Figure 3C shows the QTL, but does not really support the statement? Figure 3D – what does the white color mean, as that is the locus under the QTL so should be the most important but is not in the legend? Maybe monomorphic, but if so there would not be a QTL?

Line 254: I think the word “average” needs to be added - “...associated with an AVERAGE height increase...” (also it is not shown) and “...and an AVERAGE grain yield...” as some seem to decrease yield.

Line 261: “...do not count with...” is awkwardly worded. Do you mean breeders don't measure the Mara allele?

Line 290/291: Fig. 1C does not involve QTL, maybe the authors meant a different figure? Similarly Fig. 4A shows a trait response and does not support the statement.

Line 292: similarly Fig 4D does not support the statement, and I don't think 4B and 4C were mentioned previously?

Line 304/305: I can not see this from Fig 4C.

Line 361/362: I think the goal is to have “...increased yield potential and other agronomic traits, such as reduced reliance on nitrogen fertilisers,...” and that may be done by creating “...genetic mosaics that include AGs 1, 3, 4, 6, and 7...”, not the other way around as written? The mosaics are not the goal, improved phenotypes are the goal? Also, the AG coloring is very difficult to see; in the 'NILs' of Fig. 4b; I do not see AG7 and I do not see mosaics? The mosaics appear in the existing varieties, so is Watkins needed for this, it seems to contradict the point? I think some minor wordsmithing can fix both issues.

Line 364: I did not find units or legend of abbreviation for Fig. 4C, so from those abbreviations if there is something about fertilizer and greenhouse gas emissions I could not follow it – maybe this is in Supplementary Table 17, but it should be more easily discerned from a main figure. Similarly, Fig.

3E shows grain yield and not fertilizer or greenhouse gas emissions. The main concern of this comment is support of the statement made – adding a few words (such as the names and abbreviations of fertilizer and greenhouse gas traits to the main text) could fix this.

Fig 1. A number of words are unnecessarily capitalized (e.g. Genome, Indels). Part d – 18 modern wheat cultivars vs. 15 mentioned elsewhere?

Line 524: the period between environment and highlighting should be a comma or deleted. Also unnecessary capitalization (e.g. Grain Weight vs. Grain Number and Grain Yield vs. Grain Protein)

Line 592: what is a “outdoor glasshouse condition”? Aren’t all glasshouses outdoors and the plants in them indoors?

Line 637: I think “typically” should be “typical”; also I don’t think 261 million SNPs + 17 million InDels is typical; I think that is a new high record? If so, that is part of the novelty of the manuscript and should be emphasized / compared with the next highest study in a few words of main text. Not to focus on the genomics but to emphasize that unprecedented data collection was conducted.

Line 667/668: Authors mention Insertions (no need to capitalize) and Deletions (same) and then mention CNVs. I think they mean these to be synonymous but would be helpful to be explicitly clear on this.

Line 686: “were” to “was”

Line 702: “comprises” to “was comprised of”

Line 791/ 829/ 832/ etc.: Nitrogen need not be capitalized.

Line 901/902 and all of M&M: I don’t remember a trait-trait relationship analysis being presented. Are there methods that no longer apply to the current version of the manuscript? This would be an interesting figure to support the main messaging, but would unfortunately rely on averages. I like that Fig 2 show these trait-trait relationships in individual lines.

Line 927/928: awkwardly worded

Line 791/ 829/ 832 / 841/ 966 and many others: nitrogen treatment is discussed a lot in M&M, but to my memory and searching, it does not play a role in the results and should be deleted. 966-971 is a good example.

Line 975: I thought it was 73 not seventy-two? Maybe I am confusing RILS with this introgression, but please check?

Line 996: What is heterozygous 0.9? I would understand -1,+1 or 0. Given how analysis was done, 0.9 would not penalize the right things?

Line 998/999: Suggest changing "Let's denote..." to "The state of the SNP was in haplotype clusters 1 and 2 were denoted as..." Further commentary in this paragraph should probably be past tense.

Line 1193-1195+: do these lines belong in a figure?

Supplementary Note 1: This note is excellent and very interesting. The fact that 118 lines originally collected in China were repatriated thanks to ex-situ conservation seems like a really big deal for germplasm resources. I suggest putting this in a few words in the main manuscript – it is implied but the 118 would give it specificity.

Referee #3 (Remarks to the Author):

* a key output is the finding that only 20-30% of wheat variation in the Watkins landrace collection has been captured in so-called modern varieties. In deriving this output the authors establish an extremely impressive resource data set for wheat.

* the Originality and significance, Data & methodology, Appropriate use of statistics, and Conclusions are satisfactory.

* Suggested improvements (minor and major edits combined):

line 90. Since the Ukraine, Australia and Argentina form a clear group in terms of the % of world trade, change five countries to seven countries and include Australia and Argentina in the list.

line 135. "..., and the 15 ..." should read "..., as well as the 15 ..."

line 143. the text needs to clarify how the AG2/5 groups relate to the reticulate nature of the AGs as exemplified by the AG1/3 groups

lines 278-280 DELETE sentence "This knowledge" preaching to the converted

lines 329-330 DELETE sentence "We cannot overstate" preaching to the converted

lines 356-357 DELETE sentence "Importantly" or clarify why the point is made in the middle of a paragraph related more directly to the manuscript.

Figure 1. "Modern wheat" is a terminology used in the main text and Figure legends. this needs a clear definition.

Figure 2.

2b The cartoon diagram of the traits are basically content free. DELETE

2c The resolution of the pics in the top and middle panels are not publication quality. Improve or DELETE.

The pots with wheat plants in the middle panel is content free-DELETE.

The field plots in the lower panel are completely fictitious and are an embarrassment to the high quality of the data presented. DELETE.

2f, 2g The gray and pink colors are difficult to discern in the dots. Use well defined colors rather than pastel-type

Figure 3

3a The Y axis needs to start from 0 cm and show breaks in the axis at ca 20 cm and ca 80 cm to cover the wide range in heights. Present annotation is not acceptable

3b The cartoon of bags of grain is uninformative - DELETE

3d The color coding of the "share" and "modern-enriched" are virtually impossible to distinguish. Make one of them a definite color rather than pastel

3c. The X-axis indicates a blue line with a dial. Explain or DELETE

Figure 4

4a. The color codes for the AGs on the left hand-side need to be provided as lines with much more weight so that the colors are clear. This Figure is a main feature in the paper and needs to be unambiguous.

* It is not clear if Extended Data Figures will appear as such in the main paper

* Extended Data Figures 1 and 2 still have major issues and parts are either not of publication quality or not relevant. The authors are persisting in using content free Biorender type illustrations. The order of parts a - g seems to be random.

Figure 1a needs to show publication quality photographs. Also the term "modern" is used here, and in the text, and needs to be defined

Figures 1b and 1c are not acceptable. The information in 1b-bi is content free, bii is the only relevant part but with arrows going to Figures c and d is basically inexplicable. In 1c a completely fictitious cartoon and content free illustration is provided.

Figure 1d - has parts di, dii and diii out of a normal order going from left to right. Part dii has "seletion" in stead of "selection"

Figure 1f - the authors are still including content a free cartoon illustrating diagnostic SNPs

Figure 1g seems to be illustrating something one might use in a lecture but is irrelevant in this paper. If the different spike phenotypes are relevant the pics need to be of publication quality. The legend to Figure 1g includes what appears to include text from the main manuscript and certainly does not clarify what is illustrated.

Figure 2b the colors used in the form of shades of pink are too difficult to decipher, especially in Figure 2a

Author Rebuttals to First Revision:

Referee #1 (Remarks to the Author):

First, I want to congratulate the authors on a revised manuscript that has dealt with the majority of reviewers comments in a productive way. In my opinion, the manuscript is greatly improved and is much easier to follow. Indeed, there are vast data sets and analyses that took this reviewer much time to again assess and comprehend -- but this I believe is the strength of the manuscript – a vast dataset of appropriate genetic/genomic resources that together will have a significant impact on wheat improvement. Shortening the manuscript with focus is welcomed, and the figures of main text are improved.

Some comments that I think are still worth while to address:

1. The authors indicated that they have developed sufficient genomic, genetic and phenotypic resources that will allow breeders to harness untapped wheat landrace diversity. While genetic and genomic resources pave the way to utilizing these resources, there is still the significant bottleneck of recombination and and potential linkage drag. This is difficult to overcome despite genomic and phenotypic resources. I would like to see this added, as it implies genetic and genomic resources alone will pave the way to effective utilization of resources. This is partly true, but the onus will still be on breeders to effectively recombine these alleles with the adaptative variation already regionally deployed.

Thanks for all of these useful suggestions. Now we added sentences in the Discussion section: “For breeders, there are still significant barriers to combining novel Watkins alleles in one single variety. New innovations in breeding technology are still required to overcome linkage drag, so that new beneficial alleles can be introduced while maintaining optimal combinations already regionally deployed.”

2. Pg 5 132. The authors note that their analysis reflects “the reticulate relationship among ancestral groups and further demonstrates the need for alternative methods when the aim is ancient wheat population inference”. Could the authors clarify what alternative methods and for what purpose. The sentence seems almost incomplete.

Thanks! We have changed the sentence by adding the underlined part: “This reflects the reticulate relationship among these ancestral groups and further demonstrates the need for alternative methods, particularly sophisticated models reliant on haplotype-based clustering for accurate inference of ancient wheat populations¹⁰”. **The model based method we use is the one described in reference¹⁰**

3. Page 8, 209 This can be always the same direction, such as increasing for yield and disease resistance..... This sentence was awkward to this reviewer.

We have deleted this sentence and replaced it with:

For traits subject to directional selection, such as yield and disease resistance, all useful alleles were acting in the same phenotypic direction but for traits like heading date, that are subject to stabilising or disruptive selection, both allelic effect directions were considered useful.

4. The authors note that although RHT8 has been mapped at high genetic resolution, breeders do not count with diagnostic markers. What is meant by counting with diagnostic markers? I was a bit perplexed as to what this meant.

We have changed it to: “...breeders do not have access to diagnostic markers for the Mara allele.”

5. On Page12 L305, the authors note haplotypes were identified that resulted in 0.91 tonnes per hectare increase. This sounds impressive, but does not provide context because an increase must be “relative to something”. For example, a 0.91 tonne increase would represent a 25-35% increase in grain yield in my regional breeding program, but perhaps 10-15% in a higher yielding environment. It may be worthwhile here to also indicate a percentage increase relative to the trial mean.

We have improved it to: “By comparing the allelic effects between Watkins and Paragon in an isogenic Paragon background, we found variation in heading date ranging from 6 days earlier to 2 days later, height effects varying from 5 cm reduction to 13 cm increase, ”

Considering the 0.91 tonne increase in yield we stand by this but we need to provide context (UK high yield potential and our use of ‘Paragon’ a now outclassed spring wheat, which we actually grow as a winter wheat (October drilling). So we have also added to the Discussion: “It is also the case that most of the genetic gains that we have described have been quantified using NILs in the genetic background of the UK spring wheat variety ‘Paragon’ which, released in 1998, expresses ~70% of the yield potential of modern UK winter wheat varieties. It remains to be seen whether Watkins alleles for yield increase will deliver benefits in the next generation of varieties.”

6. P12 L307. The authors indicate that they identified five haplotypes that conferred significant increases in grain protein, there of came without a negative impact for grain yield. Can the authors point to the data in the main text? I did have to dig to find it. I think its worth while that claims in the main paper are referenced to the supporting data – especially given this is indeed a significant finding.

Thanks for this comment. We agree with the reviewer that this is worthwhile, now directing the reader to Supplementary Table S26: “Our allele prioritisation for breeding included the consideration of deviation from trait antagonism (Fig 2g and with detail in Supplementary Table 26).”

7. On page P12 L315. The authors note that the arithmetic sums of allelic effects are helpful in conveying the breeding potential of this resource. This is true (there are other ways). I was however confused at the sentence “Looking at significant yield component effects in this way reveals a 4.5 t ha⁻¹ increase in grain yield, an increase of 11,500 grains, and an increase in thousand grain weight of 55.6 mg. I could not ascertain the relevance of these, where these calculations were coming from and what it really means relative to the mean.

These appear to be impressive increases, and I want to be sure the authors reference the appropriate data so that readers can appreciate the meaning.

We apologise for the unintentional ambiguity here. We now improved the text to remove the ambiguity and to point to the data:

“Looking at the sum of significant allelic effects (see Supplementary Table 26) for yield components in this way reveals a 4.5 t ha⁻¹ increase in grain yield, an increase of 11,500 grains m², and an increase in thousand grain weight of 55.6 mg. These are valuable breeding targets validated in isogenic backgrounds, but realising the benefits by combining these alleles within breeding pedigrees will be dependent on genetic interactions, which fall outside the scope of the current study.“

8. P13 L342: Development of the first wheat haplotype map. I am not sure this is the first, given previous haplotype maps of wheat being published. Please confirm.

Apologies, this was unintentional, and reading this back it does come across as disrespectful to previous work such as (Jordan, K. W., Wang, S., Lun, Y., Gardiner, L. J., MacLachlan, R., Hucl, P., ... & Akhunov, E. (2015). A haplotype map of allohexaploid wheat reveals distinct patterns of selection on homoeologous genomes. *Genome biology*, 16, 1-18.), which was accomplished before the full-whole genome reference was done. What we meant was:

Our development of a wheat haplotype map based on whole genome sequence, coupled with a haplotype–phenotype association allowed us to estimate frequencies of beneficial alleles and assess their functional significance by association with high resolution NAM QTL.

9. I appreciate that the authors have reduced the main text and have worked to introduce those parts that were cut into the discussion section. However, to me the discussion seems rushed, and not yet fully interconnected. For example, the authors noted that their study

showed that the key resistance allele Pm4f was only detected in Watkins. This reviewer is familiar with this resistance, but a general audience would not be.

Thanks for the comment. We agree that the structure of the discussion had been somewhat lost. Now the discussion section is improved by consolidating common themes. In this section of text, we tried to demonstrate that our open collaboration with these data sets, even before publication, was leading to high impact discoveries (Septoria, blast, minerals, new array). This is important as it will inspire readers to do the same. We have made the small change shown below to make this absolutely clear. We think that the rest of the sentence as it stands is enough to make the main point and if readers want to find out more it really would be more appropriate to go to the relevant publications (now accepted). The change is:

“This includes the identification of novel genes for Septoria resistance³⁵ and the first wheat gene conferring resistance to the devastating Bangladesh/Zambia MoT isolate³⁶ for which the key resistance allele Pm4f was only detected in Watkins.”

Also, how can the authors be sure that the Pm4f allele is now being deployed in breeding programmes worldwide as a direct result of the Watkins resources?

Thanks for pointing this out. The reviewer should be assured that these outputs are being deployed. For example this project:

<https://www.jic.ac.uk/press-release/lead-role-for-john-innes-centre-as-the-uks-first-cgiar-centre-is-launched-by-rishi-sunak/>

includes the introgression of these alleles into CIMMYT backgrounds. It does not seem appropriate to add this kind of detail to our Discussion. However, the resistant allelic information and the lines carrying are all fully accessible to any user (true for all of the Watkins germplasm).

In my last review I had asked the reviewers to stress the limitations of short reads, and their response indicates a discussion point devoted to this. Perhaps I missed it but I don't see it other than their work mentions recent long read assemblies.

See below.

My preference would be that the authors note the limitations of the data sets. For example, structural variation maybe missing from their datasets. There are other limitations as well. This is not to be critical to the work that was done, but I feel it would be good to point out these limitations and to stress that there is likely more variation that will still need to be characterized.

We agree with the reviewer and re-emphasize these limitations:

Consideration of limitations and further steps that will facilitate even better utilisation of the Watkins genetic resources raises questions of alternative sequencing/bioinformatic

approaches and, crucially, the deployment of variation in registered varieties. Large scale structural and copy number variations are an important component of genetic variation in wheat but they escape detection using the short read sequencing technologies deployed in this study. Following the model of the 10+ Wheat Genomes project¹¹, and incorporating long-read sequencing technologies³⁴ would provide significant uplift to the value of the Watkins resources.

The authors note that Watkins resources have also been used to create a mineral atlas for wheat with new variants for the breeding of more nutrient dense wheat (Nature Food submission with reference number “NATFOOD-23121815”, entitled “Improving wheat grain composition for human health: an atlas of QTLs for essential minerals”). What are these references? Please clarify.

We are sorry that the reference number did not take the reviewer to the manuscript. We now say:

“The Watkins resources have also been used to create a mineral atlas for wheat with new variants for the breeding of more nutrient dense wheat ([Research Square preprint: “https://doi.org/10.21203/rs.3.rs-3714819/v1”](https://doi.org/10.21203/rs.3.rs-3714819/v1), entitled “Improving wheat grain composition for human health: an atlas of QTLs for essential minerals”).”

Referee #1 (Remarks on code availability):

While I am not an expert on all the codes/analysis provided, the analyses fits to that which is typical of the discipline (including QTL/GWAS, field trials, statistics).

Referee #2 (Remarks to the Author):

Requested points for reviewers to address.

A) The key results remain the same. There is great untapped diversity for crop improvement in a unique wheat germplasm collection and the authors do extensive work to characterize the phenotypic and genomic diversity in various ways. This demonstrates the value of ex situ collections for food security through crop improvement as well as the extensive amount of effort needed to meaningfully tap into this diversity.

B) This work is highly original and significant.

C) The data is valid and of high quality. The presentation is very good, but more editing should be done to make it excellent.

D) Statistics and uncertainty treatment is appropriate.

E) The conclusions are robust, valid, reliable and strong. Some editing could better bring out the main points for readers still but these are much improved.

F) No additional experiments or data are suggested, arguably there are already too many experiments for readers to comprehend. However, improvements in clarity of presentation could still be made.

G) References. Seemed appropriate but I did not fully investigate.

H) Clarity and context is much improved and good, but some minor edits and reorganization of the main paper could make it excellent.

Overall this manuscript is much improved and simplified, responded well to reviewers, and is a very important study and among the most impressive data sets and reporting of such important information; so it should be published quickly. However, as discussed in the last review, with such an impressive study comes very difficult challenges in clear and effective presentation on the discovery and potential impacts. I found it was still painful (but much less so!) to go through as a reader / reviewer and I am sure even more painful for the authors (I was especially impressed how quickly they made major revisions without errors or typos – except for maybe in M&M). It was most painful because the figures were in 3 separate places rather than in order of callout- not sure Nature allows them to be placed in body of manuscript for review, but would have made it much more pleasant to read such a massive amount of work.

My main major remaining issues with the manuscript is that I think there is still excessive genomics information unimportant for the main message and discussed as methods or results, but not as a discovery or outcome – which is what readers will care about.

Thanks for all of these very careful comments with many helpful suggestions. We agree and reduce the excessive genomics results as suggested. See below

I think the authors still focus on genomics variation being interesting in and of itself, when over the last 10 years as inexpensively “sequencing anything” has become feasible, researchers and breeders are now focused on functional diversity being the interesting component, of which the authors have lots of data and demonstrate, but this gets lost. The excessive genomics focus is especially (but not exclusively) true in the supplemental figures and materials and methods – I suggest there is no reason to call most of these genomic supplemental figures out in the main text with little discussion. As an example - Supplementary Figures 1, 2 will likely only be of interest to reviewers to check the data methods were appropriate and be impressed with data volume. There are no biological insights from this.

Thanks again. We deleted Supplementary Figures 1, 2, 6, 7, 9, 10, 12, 14, 16, 18 and 19. We only retained Supplementary Figures 3, 4, 5, 8, 11, 13, 15, 17, 19 and 20 as suggested below. We tried our best to keep all of the supporting results concise, complete and informative to deliver the key messages to our readers.

One possibility is to put all of this genomics quality control “stuff” into a supplemental note and only refer to it there, so then in the main manuscript the authors can just say extensive quality control was applied to this sequence data (see Supplementary Note). As a separate genomics example in Supplementary Figure 9 – a figure for pi is used, but not really discussed in a way that shows it is meaningful to readers, more of an academic exercise; similarly Supplementary Figure 10 presents the locations in gene models this diversity is located, but why should readers care since the functional value is what matters? Both S9 and S10 and others are trying to show with different statistics that the diversity in Watkins is unique but how many ways does this need to be presented; is it a major finding leading to new insights? Again, moving this to a supplemental note (e.g. “uniqueness of Watkins genomic diversity”) containing all related figures that a minority of readers will be interested in could make understanding the major findings easier. Overall these genomics supplemental figures focused strongly on the volume of data and insinuated insights, which I found to be a distraction from the important main messages and had lost enthusiasm by the time I got to those locations. If Nature will not let supplemental figures only be referred to in supplemental notes, I suggest continuing to reduce the supplemental figures and Materials and Methods (or move some M&M to supplemental), asking the important question of what the main discoveries were and what a reader would be interested in? Given the number of genomics supplemental figures, I would expect there to have been new methods / genomic analysis approaches that had to be developed which 1) would probably better fit a specialty journal, and 2) are not clarified as innovations in the main manuscript – so maybe they are not novel and just standard QC and statistical analyses?

Looking over them again, supplementary Figures 3,4,5,8,11,17,19 and maybe 13,15,20 are the only ones I think are important for the main messages in the main manuscript. The rest I think could just be in supplemental notes (or if not allowed – deleted).

Overall this review is meant to be helpful and I hope this does not come across as overly critical, there is a lot of great work in this manuscript and the authors have done a great job, especially in revision and should be commended. Yet it is still a lot for readers and reviewers to digest. Continuing to try to see it from a target audience reader and reviewers perspective and presenting it more clearly will lead to greater visibility and impact from all of their work.

We are very grateful for the detailed comments and suggestions from this reviewer and we appreciate the help in improving the manuscript. We had a careful check and reduced the Supplementary Figures as suggested.

A few other general notes

The name cultivars should be bracketed by apostrophes in first use (e.g. ‘Jagger’) following plant breeding convention and including in figures. Lines with release notes (or PVP, patents) should be cited, this is important to credit the developers of these lines. Both recommendations are a best practice, unless Nature requires otherwise.

Thanks for pointing this out. First mention of variety is now in apostrophes. In these cases (Paragon, Mara etc), there are no PVP or release notes.

I remain unclear on what ‘Watkins’ is. I thought it was a collection, but in Supplementary Fig. 20 and in text it appears that it is a cultivar? Please clarify and explain.

When we say ‘Watkins’ in this manuscript we are referring to the 827 landrace cultivars of bread wheat of the AE Watkins landrace collection. (There are also Durum wheats in a wider collection, to which we do not refer to in this manuscript). Our use of landrace cultivar and modern cultivar is now consistent throughout the text.

I am still unclear on how the “15” and “18” cultivars were selected, why, and what they were used for. A couple more sentences on this would help clarify. Maybe I missed it.

The “15” indicates the 15 previously described wheat cultivars from the 10+ wheat pan-genome project (<https://www.nature.com/articles/s41586-020-2961-x>, <https://10wheatgenomes.com/>), this has now been clarified in the legend of Figure 1. The “18” indicates the selected 18 representative modern wheat cultivars (released from 1920 to 2011, see Source Data) to show the mosaic structure and timeline diversity of modern wheats composed of Watkins landrace wheat.

The authors mention epistasis in the response to reviewers but I did not see it acknowledged in the main manuscript and is a caveat that breeders recognize but molecular breeders rarely do – there should be a sentence on this, that not all alleles performed as expected in a different genetic background.

This is absolutely right. We have added:

“These are valuable breeding targets validated in isogenic backgrounds, but realising the benefits by combining these alleles within breeding pedigrees will be dependent on genetic interactions, which fall outside the scope of the current study.”

Line by line comments:

Line 102-103: The challenge is not a lack of associated sequence information, it is the trait and the phenotyping, which this study closes the gap on. Saying the sequence information is important is putting the cart before the horse. I think if it was a marker for selection that would make more sense as important for breeders, but sequence information is not really needed in and of itself for breeding?

We appreciate this point so have modified the first sentence: “Among these challenges is the lack of genetic resources and appropriate phenotypic datasets underpinned by the sequence information that is necessary for identifying alleles and haplotypes present in the resource but absent in modern cultivars.”

And deleted this one:

Additionally, the lack of relevant phenotypic value for quantitative traits of interest to breeders further complicates the integration of landrace diversity into breeding programs.

Line 120: If the goal is to discover underutilized landraces it should be variation and not variations?

Thanks for pointing this out. We replaced “variations” by “variation”.

Line 121+: I did not find supplementary Table 1 (or others) when I downloaded the Zipped file of everything from Nature review website and had to review this offline. So, I did not evaluate supplemental tables during my review, but found them on the website as I was about to submit my review. I apologize to the authors and editors but I do not have time to evaluate these now for timely review submission.

Thanks to the reviewer for the effort and time to comment our manuscript with great suggestions. We totally understand the point from this reviewer and try to reduce the unnecessary excessive genomics information unimportant for the main messages both from the supplementary figures (already finished as above) and from the supplementary tables, to deliver the key discoveries. Based on this principle, we removed Supplementary Table 5, 8, 12, and 13 and improve the legend description for some of the other tables to highlight the key messages for each table. We updated Supplementary Table 4 to make the key points and take home messages clearer (also to address the question from referee#3). We think that the remaining supplementary tables are important either to support the key statements in the Main Text, or as supporting evidence to the Main Figures or Extended Data Figures, so we would really prefer not to delete these. We also updated some sections in the Online Methods. Furthermore, to facilitate finding all the useful information from our manuscript, we provided additional information for our manuscript into our WWWG2B website. We believe that different readers will find different and important angles for reuse of these datasets (the R in FAIR) and we really think that the inclusion of these analysis entry points provides huge value to these readers in the pursuit of their own studies inspired by different expertise and taste.

Line 124/ Sup fig. 2: It is not clear what was used as the reference allele?

The reference allele is the IWGSC RefSeq v1.0, Chinese Spring. We now write: “We aligned these sequences to the IWGSC RefSeq v1.0 bread wheat reference genome⁸”

Line 136: Unclear what the 15 previously described accessions are? Each article should stand alone. I have three different windows open to try to follow this and should not have to dig up

a publication to determine what they are or why they were selected. Maybe these are the 15 lines in figure 1b?

The first reviewer also pointed it out on the same issue. We addressed this as already written in the replies above:

The “15” indicates the 15 previously described wheat cultivars from the 10+ wheat pan-genome project (<https://www.nature.com/articles/s41586-020-2961-x>, <https://10wheatgenomes.com/>), this has now been clarified in the legend of Figure 1:

“The distribution of the 15 genome lines from the 10+ wheat pan-genome project¹¹ and Chinese Spring are shown.”

The “18” indicates the selected 18 representative modern wheat cultivars (released from 1920 to 2011, see Source Data) to show the mosaic structure of Modern wheats composed by Watkins landrace wheat.

Line 150: Sup Fig. 9 does not show the point or add much to manuscript. Sup Fig 10b I am not sure what AG unique is, if it is not Watkins unique 10a. Is 10b unique to AG groups within Watkins and 10a is Watkins unique to modern cultivars? Some minor additions to figure description would clarify.

The Figure has been deleted.

Line 154: I don't understand Figure 1d? Is this averaging what is shown in Sup Fig 11 or how was this calculated differently?

Apologies for the confusion. In Methods, we tried a careful description on the K-mer based IBS (identity by state) approach, which was how Figure 1d and the old Sup Fig 11 was generated; we now improved the legend of Figure 1d, to make this clearer. We coloured the shared haploblocks between modern wheats and Watkins landraces according to the seven ancestral subgroups. The illustration strategy is the same as that used in our citation (Brinton, J. et al. A haplotype-led approach to increase the precision of wheat breeding. *Communications Biology* 3, 712 (2020). <https://doi.org:10.1038/s42003-020-01413-2>).

Line 158: on Sup Fig. 11 – how were these “18” chosen? Why do they differ from the “15” mentioned earlier (if they do)?

We have addressed this point above.

Line 160: I can not imagine how someone will use Sup. Fig 12. Furthermore, this is expected. I think this figure could be deleted or moved to just in the supplemental note. At least the novelty should be discussed if there is novelty, conventional knowledge would have predicted this and I am looking for places to simplify.

Thanks for the suggestion. We have deleted the figure now.

Line 179: Why do the colors switch between modern and Watkins in part A and B of Sup. Fig. 17. It took me awhile to figure this out.

Thanks for pointing this out, we have updated the figure (now Sup. Fig. 6) and made the colours of the panels more consistent.

Up to this point, many of the numerous figures (supplemental especially) seemed relatively unnecessary to main message as they focus on genomic differences and not functional differences. I would encourage the authors to halve these or move just to supplemental note callouts.

Thanks again. As said above, we have deleted the unnecessary figures.

Line 191: Sup. Fig. 18 - Most of these images are unnecessary and don't provide insight. The map is useful but I think presented elsewhere?

The map seems useful to make it clear where the phenotypic datasets were collected. This may be important because of the huge amount of phenotyping datasets from different climate zone and experimental stations. We hope that the multiple-site-multiple-year data sets make the study more reliable and robust. This is now Sup. Fig. 7

Line 210: I still find this Extended Data Figure 5 confusing, not sure if it was updated as authors say in revision? Parts a, b, c make sense (although I feel like these were presented earlier and are more about germplasm diversity). Part e,f,g are pretty difficult to discern. The color on part Extended Data Figure 5e I am guessing is supposed to match to something? I can't see color well (the black outline could be removed) and it does not line up with part 5b? It mentions 66 RIL parents, but 72 are discussed before then, what happened to the other 6? Part 5f is clear, but why order so they don't line up with 5g? We cannot see where the curved lines go? Parts 5f and 5G are the only important parts of the figure as far as I can tell (assuming 5a and 5c were presented earlier). I suggest an entirely new figure easier for readers focused on parts e,f,g.

Thanks for the detailed comments. As suggested, we now update the figure as below:

Line 219: Figure 2e does not support the statement, it shows a QTL but does not show the 33 accessions?

In panel 2e, we were not attempting to fully support the statement. It is an example of the power of biparental QTL analysis. We now point the reader to the detail of resistant accessions and QTL discovered in the text:

“Supplementary Table 23), a recently emerged, highly aggressive race with increased pathogenicity at elevated temperatures²⁴. Iran is the dominant country of origin for these resistant accessions (14 out of 33). GWAS did not identify significant MTAs for these resistance loci, but biparental QTL mapping identified 15 new loci conferring yellow rust resistance in the UK and Australia, including to the Warrior race (Supplementary Tables 24-25)”

Line 252: Figure 3C shows the QTL, but does not really support the statement?

Thanks for the comment. We updated Figure 3c/3d (as well as 3b) to make the message clearer. As below:

Figure 3D – what does the white color mean, as that is the locus under the QTL so should be the most important but is not in the legend? Maybe monomorphic, but if so there would not be a QTL?

We added the missing explanation in the legend for the white colour, “and other (white, refers to no haploblock detected in this region)”. As above, we have updated Figure 3c/3d.

Line 254: I think the word “average” needs to be added - “...associated with an AVERAGE height increase...” (also it is not shown) and “...and an AVERAGE grain yield...” as some seem to decrease yield.

Thanks! We changed as suggested.

Line 261: “...do not count with...” is awkwardly worded. Do you mean breeders don’t measure the Mara allele?

Thanks! We rephrased this sentence, see above.

Line 290/291: Fig. 1C does not involve QTL, maybe the authors meant a different figure?

Apologies for the ambiguity. We are not referring to the QTL but to the landraces which carry the alleles identified in the QTL analysis. Now rephrased:

“Out of these 127 prioritised QTL targets, 107 originated from AGs 1, 3, 4, 6 and 7, which did not contribute to the genomes of modern wheat, as our analysis had shown (Fig. 1c, Fig. 4a and Supplementary Table 31).”

Similarly Fig. 4A shows a trait response and does not support the statement.

4a refers to graphical genotypes on the NILs with foreground introgressions coloured by ancestral group. The trait response is in 4c. We used these insets to avoid excessive space with no additional information. We update the representation of the 7 colours for the AGs.

Line 292: similarly Fig 4D does not support the statement,

4D summarises the 44,338 LD-based haplotypes unique to Watkins but now introgressed in the NILs. So we hope that it does support the statement.

and I don't think 4B and 4C were mentioned previously?

Many thanks, the mentioning was lost during editing. Now rectified with the addition of:

“For each chromosome the enhanced AG diversity, beyond AG2/5 is evident (Fig. 4b) and the associated allelic effects are promising for wheat improvement (Fig. 4c).”

Line 304/305: I can not see this from Fig 4C.

At this resolution it is difficult to pick out the individual effects. We have added reference to Supplementary Table 26 where the results can be seen more easily.

Line 361/362: I think the goal is to have “...increased yield potential and other agronomic traits, such as reduced reliance on nitrogen fertilisers,...” and that may be done by creating “...genetic mosaics that include AGs 1, 3, 4, 6, and 7...”, not the other way around as written? The mosaics are not the goal, improved phenotypes are the goal?

Thanks again! We agreed and wrote:

“We envision a future in which wheat crops will be developed by leveraging the full spectrum of useful diversity present in landraces. A large-scale genomic signature of this transformation will be the fact that new varieties will be genetic mosaics, integrating ancestral genomes (AGs 1, 3, 4, 6, and 7) (Fig. 4b) with a diluted contribution from AGs 2

and 5. In this study, we have shown the potential of these untapped ancestral groups to increase yield, which in turn can mitigate the reliance on nitrogen fertilisers. Such endeavours in landrace-based breeding hold promise for significant contributions to reducing the climate footprint of wheat farming. (Fig. 3e and 4c).”

But after restructuring the Discussion we felt that these sentences became unnecessary and have deleted them.

Also, the AG coloring is very difficult to see; in the ‘NILs’ of Fig. 4b; I do not see AG7 and I do not see mosaics? The mosaics appear in the existing varieties, so is Watkins needed for this, it seems to contradict the point? I think some minor wordsmithing can fix both issues.

The figures are now improved to show this colouring more clearly. Regarding mosaics, the landrace introgressions we show in 4a and the lower half of 4c do not appear as mosaics because they are intact landrace segments, and the AG designation refers to the assignment of those particular accessions to the ancestral groups. In 4d, the short range (plink) haplotype composition of Watkins unique haplotypes of those segments is summarised.

Line 364: I did not find units or legend of abbreviation for Fig. 4C, so from those abbreviations if there is something about fertilizer and greenhouse gas emissions I could not follow it – maybe this is in Supplementary Table 17, but it should be more easily discerned from a main figure. Similarly, Fig. 3E shows grain yield and not fertilizer or greenhouse gas emissions. The main concern of this comment is support of the statement made – adding a few words (such as the names and abbreviations of fertilizer and greenhouse gas traits to the main text) could fix this.

We do not show specific statistics on N use here. The yield effects we show include Watkins alleles that increase yield compared to the Paragon allele at the same N level. Deploying those alleles in breeding increases yield without the need for additional N fertiliser. We consider yield increases at the same N level to be a positive contribution to reducing GHG emissions.

Fig 1. A number of words are unnecessarily capitalized (e.g. Genome, Indels).

Thanks for pointing this. Now corrected and updated.

Part d – 18 modern wheat cultivars vs. 15 mentioned elsewhere?

We address this point as above and update the Figure legend to make the message clear.

Line 524: the period between environment and highlighting should be a comma or deleted. Also unnecessary capitalization (e.g. Grain Weight vs. Grain Number and Grain Yield vs. Grain Protein)

We updated it as suggested.

Line 592: what is a “outdoor glasshouse condition”? Aren’t all glasshouses outdoors and the plants in them indoors?

Thanks for the comment. We now changed it to: “Following vernalization in controlled environment rooms, plants were transferred to glasshouse conditions with automated watering and following 2 weeks further growth, transplanted into...”

Line 637: I think “typically” should be “typical”;

Based on the reviewers next comment we deleted this.

also I don’t think 261 million SNPs + 17 million InDels is typical; I think that is a new high record? If so, that is part of the novelty of the manuscript and should be emphasized / compared with the next highest study in a few words of main text. Not to focus on the genomics but to emphasize that unprecedented data collection was conducted.

Thank you for this suggestion. We added some words to the Discussion:

“The integrated Watkins resources described represent a major step forward for cereal research and breeding. Even in species such as rice and maize, which have transitioned into the post-genomics era well ahead of wheat, there is currently no publicly accessible resource in a project that encompasses large scale genomic analysis (which identified 261 M SNPs and 17 M indels) in primary germplasm resources integrated with comprehensive field-based phenotyping. This includes the development of extensive genetic association mapping populations, the construction of both the LD-based haplotypes and the IBS-based long-range haplotypes, and the validation of QTL effects, tagged with diagnostic haplotypes.”

Line 667/668: Authors mention Insertions (no need to capitalize) and Deletions (same)

Capitals removed as suggested.

and then mention CNVs. I think they mean these to be synonymous but would be helpful to be explicitly clear on this.

These are different. Insertions and deletions specifically mean gains/presence (from 0 to 1/N) or lost/absent (from 1/N to 0) in the genome reference, and CNVs means copy number variation (change between 1 to N) in individual accessions.

Line 686: “were” to “was”

Changed as suggested.

Line 702: “comprises” to “was comprised of”

Changed as suggested.

Line 791/ 829/ 832/ etc.: Nitrogen need not be capitalized.

Changed except at start of sentences.

Line 901/902 and all of M&M: I don't remember a trait-trait relationship analysis being presented. Are there methods that no longer apply to the current version of the manuscript? This would be an interesting figure to support the main messaging, but would unfortunately rely on averages. I like that Fig 2 show these trait-trait relationships in individual lines.

The trait-trait relationship is presented in Supplementary table 26 and it formed the basis on which Fig 2 relationships are show. We respectfully suggest that this is retained.

Line 927/928: awkwardly worded

Reworded as suggested.

Line 791/ 829/ 832 / 841/ 966 and many others: nitrogen treatment is discussed a lot in M&M, but to my memory and searching, it does not play a role in the results and should be deleted. 966-971 is a good example.

Although we do not make a major theme of it, here we report QTL for components of NUE and the QTL prioritisation did include QTL for NUTE and GPD so we think that these methods are important for the specialist reader.

Line 975: I thought it was 73 not seventy-two? Maybe I am confusing RILS with this introgression, but please check

These numbers are coincidentally close as the seventy-two refers to a 'Mara' x 'Capelle Desprez' population developed a long time ago by Korzun et al. A line has been added in parenthesis to avoid any further confusion:

“RHT8 fine mapping and gene discovery. Seventy-two recombinant lines (not to be confused with the 73 Paragon x Watkins RIL populations that make up the NAM used in this study) within the RHT8 locus (Supplementary Table 30) were used for genetic mapping²⁹.”

Line 996: What is heterozygous 0.9? I would understand -1,+1 or 0. Given how analysis was done, 0,9 would not penalize the right things?

Thanks for pointing this out. We tested parameters in several rounds of optimization of the haplotype clustering algorithm. Results for 0.9 and 0 for heterogeneous were exactly the same. To avoid confusion, we simplified the description: “Reference allele (homozygous): -1; Alternative allele (homozygous): 1; Missing: NA; Heterogeneous: 0.”

Line 998/999: Suggest changing “Let’s denote...” to “The state of the SNP was in haplotype clusters 1 and 2 were denoted as...” Further commentary in this paragraph should probably be past tense.

Thanks. We changed it as suggested.

Line 1193-1195+: do these lines belong in a figure?

Yes, these lines align with Extended Data Figure 1g.

Supplementary Note 1: This note is excellent and very interesting. The fact that 118 lines originally collected in China were repatriated thanks to ex-situ conservation seems like a really big deal for germplasm resources. I suggest putting this in a few words in the main manuscript – it is implied but the 118 would give it specificity.

It is wonderful that this referee has read the note so carefully, we would like to follow this suggestion. Now added to the discussion:

“The Watkins collection, assembled from diverse regions worldwide a century ago, is now reverting to its global significance. This is exemplified by its introduction since 2019 to China, where it is being cultivated across various regions with extensive phenotyping and crossing experiments. Within the Watkins collection are 118 landrace accessions originally collected from China but now repatriated thanks to long-term ex-situ conservation efforts.” We added one more historical fact in the efforts of large-scale germplasm exchange that would be interesting to remind the people on the importance of international communication and collaboration, particularly between China – one of the largest agricultural countries, and the rest of the world: “A similar initiative dates back to 1932 when a Chinese colleague, Shen Zonghan, introduced approximately 1,700 wheat accessions from the John Percival Collection³³, significantly contributing to wheat genetic improvement and breeding in China.”

Referee #3 (Remarks to the Author):

* a key output is the finding that only 20-30% of wheat variation in the Watkins landrace collection has been captured in so-called modern varieties. In deriving this output the authors establish an extremely impressive resource data set for wheat.

* the Originality and significance, Data & methodology, Appropriate use of statistics, and Conclusions are satisfactory.

* Suggested improvements (minor and major edits combined):

line 90. Since the Ukraine, Australia and Argentine form a clear group in terms of the % of

world trade, change five countries to seven countries and include Australia and Argentine in the list.

Thanks for the comment. We updated the sentence.

line 135. "..., and the 15 ..." should read "..., as well as the 15 ..."

We update it as suggested.

line 143. the text needs to clarify how the AG2/5 groups relate to the reticulate nature of the AGs as exemplified by the AG1/3 groups

We improve the presentation for Supplementary Table 4, and refer the relationships between each of the individuals and the AG1-7 in the updated Supplementary Table 4, and illustrate this information in Extended Data Figure 2a. These results show the reticulate relationship. We both added the figures and the tables to deliver this message. In the text we now say for AG2/5:

The highest level of admixture for AG2 is from AG5 (approximately 11.2%), for AG5 it is from AG7 (approximately 7.5%). This reflects the reticulate relationship among these ancestral groups and further demonstrates the need for alternative methods.

We remove the previous text on the other AG relationships as we think that this gets quite confusing to read and are sure that where readers are more interested they will refer directly to the table.

lines 278-280 DELETE sentence "This knowledge" preaching to the converted

We update it as suggested.

lines 329-330 DELETE sentence "We cannot overstate" preaching to the converted

We update it as suggested.

lines 356-357 DELETE sentence "Importantly" or clarify why the point is made in the middle of a paragraph related more directly to the manuscript.

We deleted it as suggested.

Figure 1. "Modern wheat" is a terminology used in the main text and Figure legends. this needs a clear definition.

Thanks for the comment. Currently we say:

These cultivars were genotyped using the Wheat Breeders' Array¹² and are independent of Watkins (hereafter 'modern wheat') (Supplementary Table 1).

We have added:

“Our modern wheat sample of cultivars comprises registered crop varieties which were developed within systematic wheat breeding programmes. This is in contrast to Watkins, which comprises landrace cultivars, which are not the products of systematic crop breeding.”

Figure 2.

2b The cartoon diagram of the traits are basically content free. DELETE

Thanks for the comment. We prefer to keep the diagram in 2b because we strongly believe that this could facilitate the general readership and provide an immediate impression about the messages we are delivering. In particular, we list the number of traits, which is important when assessing the robustness of the results.

2c The resolution of the pics in the top and middle panels are not publication quality. Improve or DELETE.

Thanks for the comment. We have improved the resolution as suggested.

The pots with wheat plants in the middle panel is content free-DELETE.

The field plots in the lower panel are completely fictitious and are an embarrassment to the high quality of the data presented. DELETE.

Thanks for these comments. Again, we keep these panels, the idea is to help the reader to understand the process and make sure they know that this is field data.

2f, 2g The gray and pink colors are difficult to discern in the dots. Use well defined colors rather than pastel-type

Thanks for the comment. We have now updated the Figure 2f/2g as suggested, as below.

Figure 3

3a The Y axis needs to start from 0 cm and show breaks in the axis at ca 20 cm and ca 80 cm to cover the wide range in heights. Present annotation is not acceptable

We appreciate the reviewer's comment. However, we still felt that not starting from zero in Fig 3a was clearer in highlighting the change of wheat height during the centuries

3b The cartoon of bags of grain is uninformative – DELETE

Thanks for this comment. We have deleted them and update Figure 3b, as below:

3d The color coding of the "share" and "modern-enriched" are virtually impossible to distinguish. Make one of them a definite color rather than pastel

Thanks for this comment. We updated Figure 3c/3d as suggest. The new plots are given above.

3c. The X-axis indicates a blue line with a dial. Explain or DELETE

The blue line was the Confidence Interval; however, to make the figure clearer, we now use a way of zooming in, which also connects 3c and 3d.

Figure 4

4a. The color codes for the AGs on the left hand-side need to be provided as lines with much more weight so that the colors are clear. This Figure is a main feature in the paper and needs to be unambiguous.

Thanks for the comment. We have updated Figure 4a with stronger colour codes for the AGs.

* Extended Data Figures 1 and 2 still have major issues and parts are either not of publication quality or not relevant. The authors are persisting in using content free Biorender type illustrations. The order of parts a - g seems to be random.

Figure 1a needs to show publication quality photographs. Also the term "modern" is used here, and in the text, and needs to be defined. Figures 1b and 1c are not acceptable. The information in 1b-bi is content free, bii is the only relevant part but with arrows going to Figures c and d is basically inexplicable. In 1c a completely fictitious cartoon and content free illustration is provided.

We changed the text to make this clearer: “the entire Watkins collection ($n = 827$) and modern wheat cultivars, which are the outputs of breeding programmes ($n = 224$; comprising 208 cultivars sequenced in this study and 15 previously described wheat cultivars from the 10+ wheat pan-genome project¹¹ as well as Chinese Spring).”

Figure 1d - has parts di, dii and diii out of a normal order going from left to right. Part dii has "seletion" in stead of "selection"

Figure 1f - the authors are still including content a free cartoon illustrating diagnostic SNPs

Figure 1g seems to be illustrating something one might use in a lecture but is irrelevant in this paper. If the different spike phenotypes are relevant the pics need to be of publication quality. The legend to Figure 1g includes what appears to include text from the main manuscript and certainly does not clarify what is illustrated.

Thanks for these comments. For Extended Data Figure 1 and 2, we have improved the resolution to publication quality and added sub-panel number to make it clearer and have clarified the link with the Results section. We kept some illustration plots, while agreeing that they do not carry direct scientific content, we feel that they will really help our readers to follow the overall picture of our research strategy. (Given the size of the datasets, from genomics resources, phenomics resources, genetics resources, QTL/gene and allele discovery, to pre-breeding efforts). We re-ordered the plots for the Extended Data Figures as suggested, which was indeed very helpful in improving the readability.

Figure 2b the colors used in the form of shades of pink are too difficult to decipher, especially in Figure 2a

We updated Figure 2a/2b as suggested. Thanks for the comment.

Reviewer Reports on the Second Revision:

Referees' comments:

Referee #2 (Remarks to the Author):

Requested points A through H were addressed in my previous review. I believe all are appropriate now, however I again did not perform a detailed investigation of references or external links beyond a few spot checks.

Overall, this is a further improved manuscript and nearly ready for publication. I find each time I read it I appreciate various components presented more and also find new scientific treasures I missed. As I have said before, this is such a tremendous amount of work, with enormous insights, and there are many additional stories the figures support which just cannot be told in a single manuscript. Yet the authors do a remarkable job and many of the most important modern big picture concepts of crop molecular genetics and breeding, as well as germplasm resources are already told in this singular manuscript. One of my favorite findings, well supported but not well highlighted, is that the QTL detection performed the best in the case study, but GWAS, NAM, and NILS were needed to dissect these QTL (no change needed, just a nice example). This massive study will allow many of us who work in adjacent areas to have empirical experience to share with students and colleagues about how to ideally conduct molecular quantitative genetics in crops. I expect no similar studies to be conducted in the scientific community, as this was such a massive undertaking. The manuscript still remains painful to get through as a reviewer, connecting each of the large numbers of figures and tables to ensure the results and discussion were supported. However, I think the main text reads both clearly and appropriately and is an excellent selection of stories to tell the scientific community. Most readers will not read in enough detail to investigate each table and figure but those willing to put in the effort will be rewarded.

There are no other major corrections needed that I found (I fear with so much information, every time I read it I will continue to find new questions). This revision I identified a few errors, minor corrections and observations to share with the authors, but will leave it to authors and the editor to determine what actually needs to be changed for publication. With so much here, I apologize to authors and editor if I missed finding any issues.

Congratulations again on a tour de force! I look forward to my graduate class reading and discussing this landmark manuscript once it is published!

Line by line comments:

Line 112 – should refer to supplemental note 1 I think?

Table S4 – Will it be clear to a reader how the authors have K=2 (for example) multiple times? A column should separate each grouping of K for clarity, I think.

Line 136 – “...as well as *THE* 15 previously described accessions”. This was addressed in the response to reviewers but is still not clear in the manuscript to me. I think if you delete “the” then readers will not look for where you have described these and realize it is a reference.

Table S9 – I believe this is miscaptioned. It says “with the largest (WATDE0352) to the smallest (WATDE0982) contribution to the modern cultivars” but I think it should be “with the SMALLEST (WATDE0352) to the LARGEST (WATDE0982) contribution to the modern cultivars”?

Line 209 – I like how the Yellow stripe rust (PstriSev_E_1to9) was highlighted in Table S20 rows. However, in table S21 I did not initially understand why this trait was not included. It was not clear why it was not in NAM, since NAM contains the QTL populations this did not make initial sense? Similarly, there was no YS QTL detected in the “natural population” anywhere near chr2B_188936527_189936527_block2131. I figured out later that this simply was not detected by any method except using QTL mapping. This is very interesting, but was initially confusing. I don’t have clear suggestions on how to improve, but it could be a very interesting point that this data set empirically shows uniquely.

Line 213 – I do not think the authors have fully considered my comments on how confusing Extended Data Figure 5 is. I don’t think it has meaningfully improved for readers.

Line 222 – The manuscript states “... increased pathogenicity at elevated temperatures.” However, the Table S23 says “cold glasshouse conditions”? This seems to be a contradiction, possibly a typo?

Line 254 – “either absent or present at low frequency in modern wheat (Fig. 3c-d)” The authors do not actually show this, they show this in Paragon only as an example, it would be very interesting to see across modern wheat. I supposed it could be figured out from the supplementary files, or maybe it is in one of the many other figures. Too bad it is not clearly shown, this is a great case study I could use to teach my plant breeding class. Maybe it can be its own paper someday?

Line 300 / Figure 4 – Figure 4C is not the right callout as shown in the figure. Based on legend, I think the figure label for B and C should be swapped? Except that they are currently correct in lines 292 and 293 – so Fig 4B and 4C should be swapped in those line numbers if swapped in figure.

Line 653 – “Conducted” is misspelled

Line 694& 695 – something odd is going on with the brackets here “[vs “]”

Line 710 – “was conducted” should replace “were conducted”

Line 770 – what is (dataset 2)?

Line 774 – needs a space between “potential” and “to”

Line 797 – I think this should be sup. Fig. 7 now

Line 1030 – I think the formula is missing?

Lines 1206/ 1218 - WWWG2B is defined a couple of sentences after the acronym was introduced which is awkward.

Line 1264, Extended Data 4 – I believe this should be e.ii at end of sentence? Those were the RILS?

Sup. Figures Line 147 – “...continue to grow...” instead of “...continuously be growing...”

Sup. Figures Line 189 – “...effort...” instead of “...efforts...”

Author Rebuttals to Second Revision:

Response to Referee#2:

Referee #2 (Remarks to the Author):

Requested points A through H were addressed in my previous review. I believe all are appropriate now, however I again did not perform a detailed investigation of references or external links beyond a few spot checks.

Overall, this is a further improved manuscript and nearly ready for publication. I find each time I read it I appreciate various components presented more and also find new scientific treasures I missed. As I have said before, this is such a tremendous amount of work, with enormous insights, and there are many additional stories the figures support which just cannot be told in a single manuscript. Yet the authors do a remarkable job and many of the most important modern big picture concepts of crop molecular genetics and breeding, as well as germplasm resources are already told in this singular manuscript. One of my favorite findings, well supported but not well highlighted, is that the QTL detection performed the best in the case study, but GWAS, NAM, and NILS were needed to dissect these QTL (no change needed, just a nice example). This massive study will allow many of us who work in adjacent areas to have empirical experience to share with students and colleagues about how to ideally conduct molecular quantitative genetics in crops. I expect no similar studies to be conducted in the scientific community, as this was such a massive undertaking. The manuscript still remains painful to get through as a reviewer, connecting each of the large numbers of figures and tables to ensure the results and discussion were supported. However, I think the main text reads both clearly and appropriately and is an excellent selection of stories to tell the scientific community. Most readers will not read in enough detail to investigate each table and figure but those willing to put in the effort will be rewarded.

There are no other major corrections needed that I found (I fear with so much information, every time I read it I will continue to find new questions). This revision I identified a few errors, minor corrections and observations to share with the authors, but will leave it to authors and the editor to determine what actually needs to be changed for publication. With so much here, I apologize to authors and editor if I missed finding any issues.

Congratulations again on a tour de force! I look forward to my graduate class reading and discussing this landmark manuscript once it is published!

We can't thank the reviewer enough for their strong support of our work. This is very kind, and we appreciate it!

Line by line comments:

Line 112 – should refer to supplemental note 1 I think?

As suggested, we now refer to “Supplementary Notes” to give our readers easy access to background information on the Watkins wheat germplasm collection.

Table S4 – Will it be clear to a reader how the authors have K=2 (for example) multiple times? A column should separate each grouping of K for clarity, I think.

Table S4 has been updated, by adding more explanation in the legend on the multiple runs for each K and an additional column that separate each grouping of K. The caption now reads: **Supplementary Table 4a.** The results of the ADMIXTURE analysis (from K = 2 to K = 9) predicting the association of each accession to an ancestral groups (AG1-7) and the genetic composition of each accession regarding the contribution from the different AGs. For the analysis 828 accessions (827 Watkins and Chinese Spring) were used and the most likely number of AGs (K), which was found to be seven.

Line 136 – “...as well as *THE* 15 previously described accessions”. This was addressed in the response to reviewers but is still not clear in the manuscript to me. I think if you delete “the” then readers will not look for where you have described these and realize it is a reference.

Thanks for the suggestion. Indeed, the “15” indicates the 15 previously described wheat cultivars from the 10+ wheat pan-genome project (<https://www.nature.com/articles/s41586-020-2961-x>, <https://10wheatgenomes.com/>), this has now been clarified in the legend of Figure 1. We agree with the reviewer to delete “THE” to make it clear to our readers.

Table S9 – I believe this is miscaptioned. It says “with the largest (WATDE0352) to the smallest (WATDE0982) contribution to the modern cultivars” but I think it should be “with the SMALLEST (WATDE0352) to the LARGEST (WATDE0982) contribution to the modern cultivars”?

We thank the reviewer for pointing this out. This urges us to double check the results and we found that our writing of the legend in Table S7 (previously it was Table S9) was correct to describe the data we present. “with the largest (WATDE0352) to the smallest (WATDE0982) contribution to the modern cultivars”, because here is describing the “cumulative contribution” from Watkins accessions to each modern cultivar, and the contribution from WATDE0352 ranked #1, it is the largest.

Line 209 – I like how the Yellow stripe rust (PstriSev_E_1to9) was highlighted in Table S20 rows. However, in table S21 I did not initially understand why this trait was not included. It was not clear why it was not in NAM, since NAM contains the QTL populations this did not make initial sense? Similarly, there was no YS QTL detected in the “natural population” anywhere near chr2B_188936527_189936527_block2131. I figured out later that this simply was not detected by any method except using QTL mapping. This is very interesting, but was initially confusing. I don’t have clear suggestions on how to improve, but it could be a very interesting point that this data set empirically shows uniquely.

Table S20 is now Table S17. We are glad that the reviewer finds this point equally intriguing as we do. We try to highlight exactly this fact, that some QTL are only found in one of the three

association/linkage approaches (here QTL mapping) and not with other methods, when we wrote in the text (line 230 – 235):

GWAS did not identify significant MTAs for these resistance loci, but biparental QTL mapping identified 15 new loci conferring yellow rust resistance in the UK and Australia, including to the Warrior race (Supplementary Tables 24-25). Twelve of these resistance loci are carried by accessions outside of the modern wheat AG2/5 founder complex, again with the majority from Iran (5). These results highlight the potential of using the large set of genetic effects identified here to help deliver new traits of agronomic value.

Line 213 – I do not think the authors have fully considered my comments on how confusing Extended Data Figure 5 is. I don't think it has meaningfully improved for readers.

We believe that the Figure is clear. The message summarized and delivered here is simple: 1) we identified a large fraction of diversity are Watkins-unique variations (>67%); 2) we demonstrate these Watkins-unique variations functional and beneficial by measuring their genetic effects and phenotypic effects (the construction of genetic population resources like NAM RIL populations, and a large-scale phenotyping work); 3) we quantified the beneficials for traits carrying target QTLs with favorable or unfavorable haplotypes.

Line 222 – The manuscript states "... increased pathogenicity at elevated temperatures." However, the Table S23 says "cold glasshouse conditions"? This seems to be a contradiction, possibly a typo?

This is not a typo but it is confusing! By "cold glasshouse conditions" we mean that the glasshouse is not heated using radiators this is because the greenhouse effect is sufficiently warming. We now change the wording in the supplementary table to "unheated".

Line 254 – "either absent or present at low frequency in modern wheat (Fig. 3c-d)" The authors do not actually show this, they show this in Paragon only as an example, it would be very interesting to see across modern wheat. I supposed it could be figured out from the supplementary files, or maybe it is in one of the many other figures. Too bad it is not clearly shown, this is a great case study I could use to teach my plant breeding class. Maybe it can be its own paper someday?

We are glad that the reviewer finds this case study here potentially useful for a plant breeding class.

The colour coding in 3d does show the unique haplotypes (dark blue) and the low frequency (light blue). We do share the reviewer's frustration that we could not expand on this a little. This is an important locus from our "Breeders Toolkit" and we do intend to use some more space in follow up publication to describe those results in detail.

Line 300 / Figure 4 – Figure 4C is not the right callout as shown in the figure. Based on legend, I

think the figure label for B and C should be swapped? Except that they are currently correct in lines 292 and 293 – so Fig 4B and 4C should be swapped in those line numbers if swapped in figure.

Thanks to the reviewer for detecting this. Fig 4B and 4C should be swapped..
We have now update Figure 4b/c accordingly.

Line 653 – "Conducted" is misspelled

Corrected.

Line 694& 695 – something odd is going on with the brackets here “[vs “)”

Corrected.

Line 710 – “was conducted” should replace “were conducted”

Corrected.

Line 770 – what is (dataset 2)?

In this study, we have defined three SNP datasets from the sequencing data using different filtering parameters:

- 1) SNP dataset 1: the high-quality SNPs after inbreeding coefficient filtering, which means it includes the rare alleles which are important for NAM GWAS (and for NAM Imputation).
- 2) SNP dataset 2: based on SNP dataset 1, but with additional filtering to remove low allele frequency (MAF <0.01) and high missing rates (>20%);
- 3) SNP dataset3: based on SNP dataset 2, with additional two-step LD filtering to reduce the dataset size for use in GWAS. This set only includes common alleles and excludes rare alleles or group or accessions-specific alleles. This set was also used for the construction of the HapMap. This is described in the online methods and corresponds to Supplementary Table 2.

Line 774 – needs a space between “potential” and “to”

Corrected.

Line 797 – I think this should be sup. Fig. 7 now

Corrected. Thanks!

Line 1030 – I think the formula is missing?

Thanks for pointing this out. It is now corrected. @Fengcong

Lines 1206/ 1218 - WWWG2B is defined a couple of sentences after the acronym was introduced which is awkward.

Thanks for pointing this out. We improve the description.

Line 1264, Extended Data 4 – I believe this should be e.ii at end of sentence? Those were the RILS?

Corrected

Sup. Figures Line 147 – “...continue to grow...” instead of “...continuously be growing...”

Corrected.

Sup. Figures Line 189 – “...effort...” instead of “...efforts...”

Corrected.